# Assouad, Fano, and Le Cam with Interaction: A Unifying Lower Bound Framework and Characterization for Bandit Learnability

**Fan Chen**
MIT
fanchen@mit.edu

**Dylan J. Foster**
Microsoft Research
dylanfoster@microsoft.com

**Yanjun Han**
New York University
yanjunhan@nyu.edu

**Jian Qian**
MIT
jianqian@mit.edu

**Alexander Rakhlin**
MIT
rakhlin@mit.edu

**Yunbei Xu**
National University of Singapore
yunbei@nus.edu.sg

## Abstract

We develop a unifying framework for information-theoretic lower bound in statistical estimation and interactive decision making. Classical lower bound techniques—such as Fano's method, Le Cam's method, and Assouad's lemma—are central to the study of minimax risk in statistical estimation, yet are insufficient to provide tight lower bounds for *interactive decision making* algorithms that collect data interactively (e.g., algorithms for bandits and reinforcement learning). Recent work of Foster et al. [40, 42] provides minimax lower bounds for interactive decision making using seemingly different analysis techniques from the classical methods. These results—which are proven using a complexity measure known as the *Decision-Estimation Coefficient* (DEC)—capture difficulties unique to interactive learning, yet do not recover the tightest known lower bounds for passive estimation. We propose a unified view of these distinct methodologies through a new lower bound approach called *interactive Fano method*. As an application, we introduce a novel complexity measure, the *Fractional Covering Number*, which facilitates the new lower bounds for interactive decision making that extend the DEC methodology by incorporating the complexity of estimation. Using the fractional covering number, we (i) provide a unified characterization of learnability for *any* stochastic bandit problem, (ii) close the remaining gap between the upper and lower bounds in Foster et al. [40, 42] (up to polynomial factors) for any interactive decision making problem in which the underlying model class is convex.

## 1 Introduction

The minimax criterion is a standard approach to studying the intrinsic difficulty of problems in statistics and machine learning. For an algorithm ALG that collects data (either passively or interactively) from the model $M$, the minimax criterion (stated somewhat informally here) is

$$\min_{\text{ALG}} \max_{M \in \mathcal{M}} \text{Cost}(\text{ALG}, M). \tag{1}$$

The expression reflects the best cost that can be achieved by an algorithm ALG for a worst-case problem instance in a collection $\mathcal{M}$, measured according to an appropriate cost function Cost. In statistics, the minimax approach was pioneered by A. Wald [90], who made the connection to von Neumann's theory of games [84] and unified statistical estimation and hypothesis testing under the umbrella of *statistical decision theory*. Minimax optimality and minimax rates of convergence of

38th Conference on Neural Information Processing Systems (NeurIPS 2024).

estimators have since become a central object in the modern non-asymptotic statistics [82, 89]; here, for instance, ALG is an estimator of an unknown parameter based on noisy observations.

Upper bounds on the minimax value (1) are typically achieved by choosing a particular algorithm, while lower bounds often require specialized techniques. In statistics, three such techniques are widely used: Le Cam's two-point method, Fano's method, and Assouad's method. These techniques entail constructing "difficult" subsets of the class $\mathcal{M}$. Le Cam's method focuses on two hypotheses, while Assouad's method and Fano's method involve multiple hypotheses. The relationships between these methods are explored in Yu [96].

Classical statistical estimation is a purely passive task. A parallel line of research [57] considers the task of *interactive decision making*, where ALG is a multi-round procedure that directly interacts with the data generating process and iteratively makes decisions with the (often contradictory) aims of minimizing cost and collecting information. Proving minimax lower bounds for interactive decision making problems presents unique challenges. The aforementioned lower bound techniques for estimation require quantifying the amount of information that can be gained from passively acquired data from a hard problem instance, but the amount information acquired by an *interactive* algorithm is harder to quantify [4, 68, 69], since it depends on the decisions made by the algorithm itself over multiple rounds.

In spite of the challenges, recent work of Foster et al. [40, 42] shows that a complexity measure known as the *Decision-Estimation Coefficient* (DEC) leads to both lower and upper bounds on the minimax rates for a general class of interactive decision making problems. Interestingly, the lower bound techniques in Foster et al. [40] proceed in a seemingly different fashion from classical lower bounds for statistical estimation; most notably, their techniques involve an *algorithm-dependent* (as opposed to oblivious) choice of a hard-to-distinguish alternative problem instance.

Given the differences between the classical Assouad, Fano, and Le Cam methods, and the even larger disparity between these methods and the interactive decision making techniques of Foster et al. [40, 42], it is natural to ask whether there is a hope of unifying these lower bounds techniques. Beyond the fundamental nature of this question, there is hope that a unified understanding might lead to tighter lower bounds, or even inspire new algorithms and upper bounds; of particular interest is to close the remaining (estimation-based) gaps between the upper and lower bounds on the minimax rates for interactive decision making left open by Foster et al. [42].

**Contributions.** We present a new framework for information-theoretic lower bounds which allows for a unifying presentation of classical lower bounds in statistical estimation (Assouad, Fano, and Le Cam) and recent DEC-based lower bounds for interactive decision making [40, 42].

- **Interactive lower bound framework (Section 3).** Our main result is to introduce a new lower bound technique, the *interactive Fano method*. The interactive Fano method generalizes the stringent separation condition in the classical Fano inequality to a novel algorithm-dependent condition by introducing the concept of "ghost data" generated from a reference distribution. This technique recovers the Le Cam two-point method (and convex hull method), Assouad method, and Fano method as special cases. By virtue of being algorithm-dependent in nature, the interactive Fano method seamlessly recovers DEC-based lower bounds for interactive decision making as a special case, and leads to refined quantile-based variants.

- **Fractional covering number and bandit learnability (Section 4).** As an application of the interactive Fano method, we derive lower bounds for interactive decision making based on a new complexity measure, the *fractional covering number*, which quantifies the difficulty of *estimating* a near-optimal policy/decision, and complements the original DEC lower bounds (which reflect difficulty of exploration as opposed to difficulty of estimation). As an application, the fractional covering number provides both lower and upper bound for learning any structured bandit problem, up to an exponential gap. In particular, finiteness of the fractional covering number is the first necessary and sufficient condition for finite-time learnability of any structured bandit problem. As a secondary result, we use the fractional covering number to close the remaining gap between the upper and lower bounds in Foster et al. [40, 42] (up to polynomial factors) for any interactive decision making problem in which the underlying model class is convex.

**Related work.** Due to space limitations, we discuss the related work in Appendix A.

## 1.1 Preliminaries

Let $P$ and $Q$ be two distributions over a space $\Omega$ such that $P$ is absolutely continuous with respect to $Q$. Then, for a convex function $f : [0, +\infty) \to (-\infty, +\infty]$ such that $f(x)$ is finite for all $x > 0$, $f(1) = 0$, and $f(0) = \lim_{x \to 0^+} f(x)$, the $f$-divergence of between $P$ and $Q$ is defined as

$$D_f(P, Q) := \int_\Omega f\left(\frac{dP}{dQ}\right) dQ.$$

Concretely, we make use of three well-known $f$-divergences: the KL-divergence $D_{\mathrm{KL}}$, the squared Hellinger distance $D_{\mathrm{H}}^2$, and the total variation distance $D_{\mathrm{TV}}$, for which the function $f(x)$ is chosen to be $x \log x$, $\frac{1}{2}(\sqrt{x} - 1)^2$, and $\frac{1}{2}|x - 1|$ respectively. For a pair of random variables $(X, Y)$ with joint distribution $P_{X,Y}$, the mutual information is defined as

$$I(X; Y) = \mathbb{E}_X\left[D_{\mathrm{KL}}\left(P_{Y|X} \,\|\, P_Y\right)\right],$$

where $P_{Y|X}$ is the conditional distribution of $Y|X$ and $P_Y$ is the marginal distribution of $Y$.

## 2 Statistical Estimation and Interactive Decision Making

We work in a general framework we refer to as *Interactive Statistical Decision Making* (ISDM). We adopt this framework as a convenient formalism which encompasses statistical estimation and interactive decision making in a unified fashion. We first introduce the framework and show how it subsumes statistical estimation (Section 2.1) and interactive decision making (Section 2.2), then give brief background on existing lower bound techniques and gaps in understanding (Appendix A).

**Interactive Statistical Decision Making.** An ISDM problem is specified by $(\mathcal{X}, \mathcal{M}, \mathcal{D}, L)$, where $\mathcal{X}$ is the space of outcomes, $\mathcal{M}$ is a model class (parameter space), $\mathcal{D}$ is the space of algorithms, and $L$ is a non-negative risk function. For an algorithm $\mathsf{ALG} \in \mathcal{D}$ chosen by the learner and a model $M \in \mathcal{M}$ specified by the environment, an observation $X$ is generated from a distribution induced by $M$ and $\mathsf{ALG}$: $X \sim \mathbb{P}^{M,\mathsf{ALG}}$. The performance of the algorithm $\mathsf{ALG}$ on the model $M$ is then measured by the risk function $L(M, X)$. The learner's goal is to minimize the risk by choosing the algorithm $\mathsf{ALG}$. As described in the Introduction, the best possible expected risk the learner may achieve is the following *minimax risk*:

$$\inf_{\mathsf{ALG} \in \mathcal{D}} \sup_{M \in \mathcal{M}} \mathbb{E}^{M,\mathsf{ALG}}[L(M, X)]. \tag{2}$$

While our main results concern the general problem formulation in Eq. (2), we focus on applications to statistical estimation and interactive decision making throughout. Below, we give additional background on these settings and show how to view them as special cases.

## 2.1 Statistical estimation

In statistical decision theory [90, 13, 12], the learner is given the parameter space $\Theta$, observation space $\mathcal{Y}$, decision space $\mathcal{A}$, and a loss function $L$. For an underlying parameter $\theta^\star \in \Theta$, $n$ i.i.d. samples $Y_1, ..., Y_n \sim P_{\theta^\star}$ are drawn and observed by the learner. The learner then chooses a decision $A = A(Y_1, \cdots, Y_n) \in \mathcal{A}$ based on the observations, and incurs the loss $L(\theta^\star, A)$.

Any general statistical estimation problem can be trivially viewed as a ISDM instance, by choosing the model class as $\mathcal{M} = \{P_\theta : \theta \in \Theta\}$ and the algorithm space as $\mathcal{D} = \{\mathsf{ALG} : \mathcal{Y}^{\otimes n} \to \mathcal{A}\}$. For model $M = P_\theta$ and algorithm $\mathsf{ALG}$, the distribution of the whole observation $X \sim \mathbb{P}^{M,\mathsf{ALG}}$ is given by

$$X = (Y_1, \cdots, Y_n, A), \qquad Y_1, \cdots, Y_n \overset{\text{i.i.d}}{\sim} P_\theta, \ A = \mathsf{ALG}(Y_1, \cdots, Y_n).$$

The loss under model $M$ is then measured by the loss of the decision $A$, i.e., $L(M, X) := L(\theta, A)$.

## 2.2 Interactive decision making

For interactive decision making, we consider the following variant of the Decision Making with Structured Observations (DMSO) framework [40], which subsumes bandits and reinforcement learning. The learner interacts with the environment (described by an underlying model $M^\star : \Pi \to \Delta(\mathcal{O})$, unknown to the learner) for $T$ rounds. For each round $t = 1, ..., T$:

- The learner selects a decision $\pi^t \in \Pi$, where $\Pi$ is the decision space.
- The learner receives an observation $o^t \in \mathcal{O}$ via $o^t \sim M^\star(\pi^t)$, where $\mathcal{O}$ is the observation space.

The underlying model $M^\star$ is formally a conditional distribution, and the learner is assumed to have access to a known model class $\mathcal{M} \subseteq (\Pi \to \Delta(\mathcal{O}))$ with the following property.

**Assumption 1** (Realizability). *The model class $\mathcal{M}$ contains $M^\star$.*

The model class $\mathcal{M}$ represents the learner's prior knowledge of the structure of the underlying environment. For example, for structured bandit problems, the models specify the reward distributions and hence encode the structural assumptions on the mean reward function (e.g. linearity, smoothness, or concavity). For a more detailed discussion, see Appendix B.

To each model $M \in \mathcal{M}$, we associate a *risk* function $L(M, \cdot) : \Pi \to \mathbb{R}_{\geq 0}$, which measures the performance of a decision in $\Pi$ under $M$. We consider two types of learning goals under the DMSO framework:

- Generalized no-regret learning: The goal of the agent is to minimize the *cumulative* sub-optimality during the course of the interaction, given by

$$\mathbf{Reg}_{\mathsf{DM}}(T) := \sum_{t=1}^T L(M^\star, \pi^t), \tag{3}$$

  where $\pi^t$ can be randomly drawn from a distribution $p^t \in \Delta(\Pi)$ chosen by the learner at step $t$.

- Generalized PAC (Probably Approximately Correct) learning: the goal of the agent is to minimize the sub-optimality of a final output decision $\widehat{\pi}$ (possibly randomized), which is selected by the learner once all $T$ rounds of interaction conclude. We measure performance via

$$\mathbf{Risk}_{\mathsf{DM}}(T) := L(M^\star, \widehat{\pi}). \tag{4}$$

With an appropriate choice for $L$, the setting captures reward maximization (regret minimization) [40, 42], model estimation and preference-based learning [23], multi-agent decision making and partial monitoring [37], and various other tasks. In the main text, we focus on reward maximization and defer the results for more general choices $L$ to the appendices (cf. Appendix B).

**Example 1** (Reward maximization). In the reward-maximization task, $R : \mathcal{O} \to [0, 1]$ is a known reward function.[1] For a model $M \in \mathcal{M}$, $\mathbb{E}^{M,\pi}[\cdot]$ denotes expectation under the process $o \sim M(\pi)$, and $f^M(\pi) := \mathbb{E}^{M,\pi}[R(o)]$ denotes the expected value function. For any $M \in \mathcal{M}$, we let $\pi_M \in \arg\max_{\pi \in \Pi} f^M(\pi)$ be an optimal decision under $M$, and the risk function is defined by $L(M, \pi) = f^M(\pi_M) - f^M(\pi)$, measuring the sub-optimality of the decision $\pi$ under model $M$.

**DMSO as an instance of ISDM.** Any DMSO class $(\mathcal{M}, \Pi)$ induces an ISDM as follows. For any $t \in [T]$, denote the full history of decisions and observations up to time $t$ by $\mathcal{H}^{t-1} = (\pi^s, o^s)_{s=1}^{t-1}$. The space of observations $\mathcal{X}$ consists of all $X$ of the form $X = (\mathcal{H}^T, \widehat{\pi})$, where $\widehat{\pi}$ is a final decision. An algorithm $\mathsf{ALG} = \{q^t\}_{t \in [T]} \cup \{p\}$ is specified by a sequence of mappings, where the $t$-th mapping $q^t(\cdot \mid \mathcal{H}^{t-1})$ specifies the distribution of $\pi^t$ based on $\mathcal{H}^{t-1}$, and the final map $p(\cdot \mid \mathcal{H}^T)$ specifies the distribution of the *output decision* $\widehat{\pi}$ based on $\mathcal{H}^T$. The algorithm space $\mathcal{D}$ consists of all such algorithms. The loss function is chosen to be $L(M^\star, X) = L(M^\star, \widehat{\pi})$ for PAC learning (4), and $L(M^\star, X) = \sum_{t=1}^T L(M^\star, \pi^t)$ for no-regret learning (3). For any algorithm $\mathsf{ALG}$ and model $M$, $\mathbb{P}^{M,\mathsf{ALG}}(\cdot)$ is the distribution of $X = (\mathcal{H}^T, \widehat{\pi})$ generated by the algorithm $\mathsf{ALG}$ under the model $M$, and we let $\mathbb{E}^{M,\mathsf{ALG}}[\cdot]$ to be the corresponding expectation.

## 3 A General Lower Bound

In this section, we introduce our general lower bound technique, the interactive Fano method, and use it to provide minimax lower bounds for the ISDM framework.

**Theorem 1** (Interactive Fano method). *Fix a $f$-divergence $D_f$. Let $\mathsf{ALG}$ be a given algorithm, $\mu \in \Delta(\mathcal{M})$ be a given prior distribution over models, and $\Delta > 0$ be a given risk level. For any reference distribution $\mathbb{Q} \in \Delta(\mathcal{X})$, we define*

$$\rho_{\Delta,\mathbb{Q}} = \mathbb{P}_{M \sim \mu, X \sim \mathbb{Q}}(L(M, X) < \Delta). \tag{5}$$

*Then, the following lower bound holds:*

$$\sup_{M \in \mathcal{M}} \mathbb{E}_{X \sim \mathbb{P}^{M,\mathsf{ALG}}}[L(M, X)] \geq \Delta \cdot \sup_{\mathbb{Q} \in \Delta(\mathcal{X}), \delta \in [0,1]} \{\delta : \mathbb{E}_{M \sim \mu}[D_f(\mathbb{P}^{M,\mathsf{ALG}}, \mathbb{Q})] < \mathsf{d}_{f,\delta}(\rho_{\Delta,\mathbb{Q}})\},$$

---

[1] We assume the reward function $R$ is known without loss of generality, since the observation $o$ may have a component containing the random reward.

*where we denote* $\mathsf{d}_{f,\delta}(p) = D_f(\mathrm{Bern}(1-\delta), \mathrm{Bern}(p))$ *if* $p \leq 1 - \delta$, *and* $\mathsf{d}_{f,\delta}(p) = 0$ *otherwise.*

This result generalizes the classical Fano method in the prequel (as well as more sophisticated variants [97, 36, 27]) in multiple ways:

- It encompasses general interactive learning/estimation problems in the ISDM framework, as opposed to purely passive estimation. This is reflected in the fact that the distribution over the outcome $X$ is allowed to depend on ALG itself.

- The most important and novel change is that Theorem 1 generalizes the "hard" separation condition required in the classical Fano method to a "soft" notion of separation captured by the quantile $\rho_{\Delta,\mathbb{Q}}$ in Eq. (5). The quantile $\rho_{\Delta,\mathbb{Q}}$ reflects the average separation under "ghost data" $X$ generated from an arbitrary reference distribution $\mathbb{Q}$, which is independent of the true model $M \sim \mu$.

- In addition, instead of relying on mutual information, which is can difficult to quantify for interactive problems, we use divergence with respect to the reference distribution $\mathbb{Q}$, generalizing a central idea in Foster et al. [40, 42].

In what follows, we will show that these generalizations allow the Interactive Fano method to achieve two important desiderata: (1) unifying the methods of Fano, Le Cam, and Assouad (Section 3.1), and (2) integrating these traditional lower bound techniques with the DEC approach [40, 42] to derive new lower bounds (see Section 3.2).

### 3.1 Recovering non-interactive lower bounds

We begin by applying Theorem 1 to recover classical non-interactive lower bounds for statistical estimation. Since a goal of our paper is to integrate the Fano and Assouad methods with the DEC framework, this serves as an important sanity check to demonstrate that our framework can recover the non-interactive versions of these methods.

**Fano method.** We specialize Theorem 1 to the KL divergence. Observe that for any reference distribution $\mathbb{Q}$,

$$\mathbb{P}_{M\sim\mu, X\sim\mathbb{Q}}(L(M,X) < \Delta) \leq \sup_x \mu(M : L(M,x) < \Delta).$$

By choosing $\mathbb{Q} = \mathbb{E}_{M\sim\mu}\mathbb{P}^{M,\mathrm{ALG}}$ in Theorem 1, we obtain the following proposition, which encompasses prior generalizations of Fano's inequality [97, 36, 27] developed in statistical estimation.

**Proposition 2** (Recovering the generalized Fano method). *Fix an algorithm* ALG *and prior distribution* $\mu \in \Delta(\mathcal{M})$, *and let* $I_{\mu,\mathrm{ALG}}(M;X)$ *be the mutual information between* $M$ *and* $X$ *under* $M \sim \mu$ *and* $X \sim \mathbb{P}^{M,\mathrm{ALG}}$. *The following Bayes risk lower bound holds for all* $\Delta \geq 0$:

$$\mathbb{E}_{M\sim\mu}\mathbb{E}_{X\sim\mathbb{P}^{M,\mathrm{ALG}}}[L(M,X)] \geq \Delta\left(1 + \frac{I_{\mu,\mathrm{ALG}}(M;X) + \log 2}{\log \sup_x \mu(M:L(M,x)<\Delta)}\right). \tag{6}$$

When applied to the statistical estimation setting (Section 2.1), the classical Fano method corresponds to the special case of Proposition 2 where $\Theta = \mathcal{A} = \{1, 2, \ldots, m\}$, $L(\theta, a) = \mathbb{1}(\theta \neq a)$ is the indicator loss, $\mu = \mathrm{Unif}(\Theta)$ is the uniform prior, and $\Delta = 1$.

Note that in Proposition 2, the term $\log \sup_x \mu(M \in \mathcal{M} : L(M,x) < \Delta)$ in the denominator of Eq. (6) takes the supremum over the outcome $x$, resulting in a simplified expression that removes the role of the algorithm ALG. This simplification is often sufficient to derive tight guarantees for estimation, but is insufficient for interactive decision making in general. The DEC, which we define in Section 3.2, more precisely accounts for the role of decisions selected by the algorithm.

**Le Cam's method and Assouad's method.** To recover Le Cam's two-point method and Assouad's method from Theorem 1, we appeal to the following result, which recovers a lower bound known as the Le Cam convex hull method [61, 96] which generalizes both approaches.

**Proposition 3** (Recovering Le Cam's convex hull method). *For a parameter space* $\Theta$ *and observation space* $\mathcal{Y}$, *consider a class of distributions* $\mathcal{P} = \{P_\theta \mid \theta \in \Theta\} \subseteq \Delta(\mathcal{Y})$ *indexed by* $\Theta$. *Let* $L : \Theta \times \mathcal{A} \to \mathbb{R}_+$ *be a loss function. Suppose* $\Theta_0 \subseteq \Theta$ *and* $\Theta_1 \subseteq \Theta$ *satisfy the* separation condition

$$L(\theta_0, a) + L(\theta_1, a) \geq 2\Delta, \quad \forall a \in \mathcal{A}, \theta_0 \in \Theta_0, \theta_1 \in \Theta_1. \tag{7}$$

*Then*

$$\inf_{\mathrm{ALG}} \sup_{\theta \in \Theta} \mathbb{E}_{Y\sim P_\theta} L(\theta, \mathrm{ALG}(Y)) \geq \frac{\Delta}{2} \max_{\nu_0 \in \Delta(\Theta_0), \nu_1 \in \Delta(\Theta_1)} \left(1 - D_{\mathrm{TV}}\left(P_{\nu_0}^{\otimes n}, P_{\nu_1}^{\otimes n}\right)\right),$$

where the infimum is taken over all algorithms $\mathsf{ALG} : \mathcal{Y}^{\otimes n} \to \mathcal{A}$, and $P_{\nu_i}^{\otimes n}$ is the distribution on $\mathcal{Y}^{\otimes n}$ induced by $\theta \sim \nu_i, Y = (Y_1, \ldots, Y_n) \overset{\mathrm{i.i.d}}{\sim} P_\theta$ for $i \in \{0, 1\}$.

Le Cam's convex hull method is the most general formulation of the Le Cam two-point method, which—in its most basic form—corresponds to the case in which $\nu_0$ and $\nu_1$ are singletons. The convex hull method is also capable of recovering Assouad's method [96]. It is important to note that the classical Fano's method, e.g. in the form of Proposition 2, cannot recover Proposition 3. This is because of fundamental differences between the divergences (KL versus TV) used in the traditional Fano method and the convex hull method.

## 3.2 Recovering DEC-based lower bounds for interactive decision making

Within the DMSO framework (Section 2.2), Foster et al. [40, 42] introduced the *Decision-Estimation Coefficient* (DEC) as a complexity measure, providing both upper and lower bounds for any model class $\mathcal{M}$. We now show how to recover the lower bounds of Foster et al. [40, 42] through Theorem 1. We focus on the lower bounds from Foster et al. [42], which are based on a variant of the DEC called the *constrained DEC*, and provide the tightest guarantees from prior work.

**Background on the Decision-Estimation Coefficient.** Consider the reward maximization setting (Example 1) under DMSO. For a model class $\mathcal{M}$ and a reference model $\overline{M} : \Pi \to \Delta(\mathcal{O})$ (not necessarily in $\mathcal{M}$), we define the constrained regret-DEC via

$$\mathsf{r\text{-}dec}_\varepsilon^\mathsf{c}(\mathcal{M}, \overline{M}) := \inf_{p \in \Delta(\Pi)} \sup_{M \in \mathcal{M}} \left\{ \mathbb{E}_{\pi \sim p}[L(M, \pi)] \mid \mathbb{E}_{\pi \sim p} D_\mathrm{H}^2\left(M(\pi), \overline{M}(\pi)\right) \leq \varepsilon^2 \right\}, \quad (8)$$

and define the constrained PAC-DEC via

$$\mathsf{p\text{-}dec}_\varepsilon^\mathsf{c}(\mathcal{M}, \overline{M}) := \inf_{p, q \in \Delta(\Pi)} \sup_{M \in \mathcal{M}} \left\{ \mathbb{E}_{\pi \sim p}[L(M, \pi)] \mid \mathbb{E}_{\pi \sim q} D_\mathrm{H}^2\left(M(\pi), \overline{M}(\pi)\right) \leq \varepsilon^2 \right\}. \quad (9)$$

Here, the superscript "c" indicates "constrained", and the superscript "r" (resp. "p") indicates "regret" (resp. "PAC"). We further define

$$\mathsf{p\text{-}dec}_\varepsilon^\mathsf{c}(\mathcal{M}) = \sup_{\overline{M} \in \mathrm{co}(\mathcal{M})} \mathsf{p\text{-}dec}_\varepsilon^\mathsf{c}(\mathcal{M}, \overline{M}), \quad \mathsf{r\text{-}dec}_\varepsilon^\mathsf{c}(\mathcal{M}) = \sup_{\overline{M} \in \mathrm{co}(\mathcal{M})} \mathsf{r\text{-}dec}_\varepsilon^\mathsf{c}(\mathcal{M} \cup \{\overline{M}\}, \overline{M}),$$

where $\mathrm{co}(\mathcal{M})$ denotes the convex hull of the model class $\mathcal{M}$.

Based on these complexity measures, Foster et al. [42] (see also Glasgow and Rakhlin [44]) provide the following lower and upper bounds on optimal risk and regret, under mild growth conditions on the DECs.

**Theorem 4** (Informal; Foster et al. [42])**.** *Consider the reward maximization variant of the DMSO setting (Example 1). For any model class $\mathcal{M}$ and $T \in \mathbb{N}$, the following lower and upper bounds hold:*
*(1) For PAC learning,*

$$\mathsf{p\text{-}dec}_{\underline{\varepsilon}(T)}^\mathsf{c}(\mathcal{M}) \lesssim \inf_{\mathsf{ALG}} \sup_{M \in \mathcal{M}} \mathbb{E}^{M,\mathsf{ALG}}[\mathbf{Risk}_\mathsf{DM}(T)] \lesssim \mathsf{p\text{-}dec}_{\bar{\varepsilon}(T)}^\mathsf{c}(\mathcal{M}),$$

*where $\underline{\varepsilon}(T) \asymp \sqrt{1/T}$ and $\bar{\varepsilon}(T) \asymp \sqrt{\log|\mathcal{M}|/T}$ (up to logarithmic factors).*
*(2) For no-regret learning,*

$$\mathsf{r\text{-}dec}_{\underline{\varepsilon}(T)}^\mathsf{c}(\mathcal{M}) \cdot T \lesssim \inf_{\mathsf{ALG}} \sup_{M \in \mathcal{M}} \mathbb{E}^{M,\mathsf{ALG}}[\mathbf{Reg}_\mathsf{DM}(T)] \lesssim \mathsf{r\text{-}dec}_{\bar{\varepsilon}(T)}^\mathsf{c}(\mathcal{M}) \cdot T + T \cdot \bar{\varepsilon}(T).$$

Therefore, up to the $\log|\mathcal{M}|$-gap between the parameters $\underline{\varepsilon}(T)$ and $\bar{\varepsilon}(T)$ appearing in the lower and upper bounds, the constrained PAC-DEC tightly captures the minimax risk of PAC learning, and the constrained regret-DEC captures the minimax regret of no-regret learning.

**A new complexity measure: The quantile Decision-Estimation Coefficient.** We recover the DEC-based lower bounds from Foster et al. [42] through a new variant we refer to as the *quantile DEC*. To do so, we briefly recount the proof technique used by Foster et al. [42].

Given an algorithm $\mathsf{ALG}$, the proof strategy is to first fix an arbitrary *reference model* $\overline{M}$, then adversarially choose a hard *alternative model* $M \in \mathcal{M}$ (in a way that is guided by the DEC and the algorithm $\mathsf{ALG}$ itself) such that $D_\mathrm{TV}(\mathbb{P}^{M,\mathsf{ALG}}, \mathbb{P}^{\overline{M},\mathsf{ALG}})$ is small, yet $\mathsf{ALG}$ cannot achieve low risk on model $M$. This lower bound technique does not explicitly require a separation condition between

$M$ and $\overline{M}$, which is a departure from the classical Fano and two-point methods. Thus to recover it, the lack of an explicit separation condition in Theorem 1 will be critical. More precisely, for any model $M$, we consider the following distributions over decisions:

$$q_{M,\text{ALG}} = \mathbb{E}^{M,\text{ALG}}\left[\tfrac{1}{T}\sum_{t=1}^{T} q^t(\cdot \mid \mathcal{H}^{t-1})\right] \in \Delta(\Pi), \quad p_{M,\text{ALG}} = \mathbb{E}^{M,\text{ALG}}\left[p(\mathcal{H}^T)\right] \in \Delta(\Pi). \tag{10}$$

That is, $q_{M,\text{ALG}}$ is the expected empirical distribution over the decisions $(\pi^1, \cdots, \pi^T)$ played by the algorithm under $M$, and $p_{M,\text{ALG}}$ is the expected distribution of the final decision $\widehat{\pi}$.

With these definitions, we instantiate Theorem 1 with the Hellinger distance. We will use the sub-additivity of Hellinger distance (Lemma C.1), which allows us to bound

$$D_{\mathrm{H}}^2\left(\mathbb{P}^{M,\text{ALG}}, \mathbb{P}^{\overline{M},\text{ALG}}\right) \leq 7T \cdot \mathbb{E}_{\pi \sim p_{\overline{M},\text{ALG}}}\left[D_{\mathrm{H}}^2\left(M(\pi), \overline{M}(\pi)\right)\right]. \tag{11}$$

Theorem 1 then yields the following intermediate result.

**Lemma 5** (Recovering interactive two-point method). *Let $\delta \in [0,1]$ be given, and consider an algorithm* ALG. *Define*

$$\Delta_{\text{ALG},\delta}^{\star} := \sup_{\overline{M} \in \text{co}(\mathcal{M})} \sup_{M \in \mathcal{M}} \sup_{\Delta \geq 0}\left\{\Delta : \sqrt{p_{\overline{M},\text{ALG}}(\pi : L(M,\pi) \geq \Delta)} > \sqrt{\delta} + \sqrt{14T\,\mathbb{E}_{\pi \sim q_{\overline{M},\text{ALG}}}D_{\mathrm{H}}^2\left(M(\pi), \overline{M}(\pi)\right)}\right\}.$$

*Then there exists $M \in \mathcal{M}$ such that $\mathbb{P}^{M,\text{ALG}}\left(L(M, \widehat{\pi}) \geq \Delta_{\text{ALG},\delta}^{\star}\right) \geq \delta$.*

Using Lemma 5, as a starting point, we derive a new quantile-based variant of the DEC, which we will show can be viewed as a slight generalization of the original PAC DEC of Foster et al. [42].

For any model $M \in \mathcal{M}$ and any parameter $\delta \in [0,1]$, we define the $\delta$-quantile risk as follows:

$$\widehat{L}_\delta(M, p) = \sup_{\Delta \geq 0}\{\Delta : \mathbb{P}_{\pi \sim p}(L(M,\pi) \geq \Delta) \geq \delta\};$$

this serves as a measure of the sub-optimality of the distribution $p \in \Delta(\Pi)$ in terms of $\delta$-quantile. We now define the quantile PAC DEC as follows:

$$\mathsf{p\text{-}dec}_{\varepsilon,\delta}^{\mathrm{q}}(\mathcal{M}, \overline{M}) := \inf_{p,q \in \Delta(\Pi)} \sup_{M \in \mathcal{M}}\left\{\widehat{L}_\delta(M, p) \,\middle|\, \mathbb{E}_{\pi \sim q}D_{\mathrm{H}}^2\left(M(\pi), \overline{M}(\pi)\right) \leq \varepsilon^2\right\}, \tag{12}$$

and define $\mathsf{p\text{-}dec}_{\varepsilon,\delta}^{\mathrm{q}}(\mathcal{M}) := \sup_{\overline{M} \in \text{co}(\mathcal{M})} \mathsf{p\text{-}dec}_{\varepsilon,\delta}^{\mathrm{q}}(\mathcal{M}, \overline{M})$. Applying Lemma 5, it is immediate to see that the quantile PAC-DEC provides a lower bound on the PAC risk.

**Theorem 6** (Quantile DEC lower bound). *Let any $T \geq 1$ and $\delta \in [0,1)$ be given, and define $\varepsilon_\delta(T) := \frac{1}{14}\sqrt{\frac{\delta}{T}}$. Then, for any algorithm* ALG, *there exists $M^\star \in \mathcal{M}$ such that*

$$\mathbb{P}^{M^\star,\text{ALG}}\left(\mathbf{Risk}_{\mathsf{DM}}(T) \geq \mathsf{p\text{-}dec}_{\varepsilon_\delta(T),\delta}^{\mathrm{q}}(\mathcal{M})\right) \geq \tfrac{\delta}{2}.$$

Unlike the original constrained DEC lower bounds (Theorem 4), which are restricted to the reward maximization variant of the DMSO setting (Example 1), the quantile DEC lower bound in Theorem 6 holds for *any risk function $L$*. As a result, the lower bound applies to a broader range of generalized PAC learning tasks, including model estimation [23] and multi-agent decision making [37], where DEC-based lower bounds from prior work are loose in general; as a concrete application, we derive a new lower bound for *interactive estimation* (Example 3) in Appendix E.2.

**Recovering DEC-based lower bounds using the quantile DEC.** At first glance, Theorem 6 might appear to be weaker than the constrained PAC-DEC lower bound in Theorem 4 due to the loose conversion from quantile risk to expected risk. However, by specializing to reward maximization (Example 3) and leveraging the structure of the risk function $L$, we show that quantile PAC-DEC is equivalent to its constrained counterpart for this setting, leading to a tight lower bound.

**Proposition 7** (Recovering the PAC DEC lower bound). *Under the reward maximization setting (Example 1), for any $\varepsilon > 0$ and $\delta \in [0,1)$ it holds that*

$$\mathsf{p\text{-}dec}_{\varepsilon}^{\mathrm{c}}(\mathcal{M}) \leq \mathsf{p\text{-}dec}_{\sqrt{2}\varepsilon,\delta}^{\mathrm{q}}(\mathcal{M}) + \tfrac{4\varepsilon}{1-\delta}.$$

As a corollary, we may choose $\delta = \frac{1}{2}$ and $\underline{\varepsilon}(T) = \frac{1}{20\sqrt{T}}$ in Theorem 6, so that

$$\sup_{M \in \mathcal{M}} \mathbb{E}^{M,\text{ALG}}[\mathbf{Risk}_{\mathsf{DM}}(T)] \geq \tfrac{1}{4}\mathsf{p\text{-}dec}_{\sqrt{2}\underline{\varepsilon}(T),1/2}^{\mathrm{q}}(\mathcal{M}) \geq \tfrac{1}{4}\left(\mathsf{p\text{-}dec}_{\underline{\varepsilon}(T)}^{\mathrm{c}}(\mathcal{M}) - 8\underline{\varepsilon}(T)\right).$$

Thus, the quantile PAC-DEC lower bound indeed recovers the constrained PAC-DEC lower bound in Theorem 4.

Our quantile DEC lower bound extends to regret with minor modifications, allowing us to recover the regret lower bounds in Theorem 4. We defer the details to the Appendix E.1 (Theorem E.1).

## 3.3 Recovering mutual information-based lower bounds for interactive decision making

The following result uses Theorem 1 to extend classical Fano method to interactive decision making and achieves tight dependence on the problem dimension that is not recovered by the standard DEC lower bound in Foster et al. [40, 42].

**Proposition 8** (Mutual information-based lower bound). *Consider the DMSO setting. For any $T \geq 1$ and prior $\mu \in \Delta(\mathcal{M})$, we define the maximum $T$-round mutual information as*

$$I_\mu(T) := \sup_{\mathsf{ALG}} I_{\mu,\mathsf{ALG}}(M; \mathcal{H}^T),$$

*where we recall that $I_{\mu,\mathsf{ALG}}(M; \mathcal{H}^T)$ is the mutual information between $M$ and $\mathcal{H}^T$ under $M \sim \mu$ and $\mathcal{H}^T \sim \mathbb{P}^{M,\mathsf{ALG}}$, and the supremum is taken over all possible DMSO algorithms $\mathsf{ALG}$. Then for any algorithm $\mathsf{ALG}$,*

$$\sup_{M \in \mathcal{M}} \mathbb{E}^{M,\mathsf{ALG}}[L(M, \widehat{\pi})] \geq \frac{1}{2} \sup_{\mu \in \Delta(\mathcal{M})} \sup_{\Delta > 0} \left\{ \Delta \mid \sup_\pi \mu(M : L(M, \pi) \leq \Delta) \leq \frac{1}{4} \exp(-2I_\mu(T)) \right\}.$$

Using Proposition 8, along with mutual information bounds from Rajaraman et al. [70], we recover a $\Omega(d/\sqrt{T})$ PAC lower bound for $d$-dimensional linear bandits, which in turn recovers the $\Omega(d\sqrt{T})$ regret lower bound [30, 73, 57, etc.].

**Corollary 9.** *For $d \geq 2$, consider the $d$-dimensional linear bandit problem with decision space $\Pi = \{\pi \in \mathbb{R}^d : \|\pi\|_2 \leq 1\}$, parameter space $\Theta = \{\theta \in \mathbb{R}^d : \|\theta\|_2 \leq 1\}$, and Gaussian rewards. The model class is $\mathcal{M} = \{M_\theta\}_{\theta \in \Theta}$, where for each $\theta \in \Theta$, the model $M_\theta$ is given by $M_\theta(\pi) = \mathcal{N}(\langle \pi, \theta \rangle, 1)$. Then Proposition 8 implies a minimax risk lower bound:*

$$\inf_{\mathsf{ALG}} \sup_{M \in \mathcal{M}} \mathbb{E}^{M,\mathsf{ALG}}[\mathbf{Risk}_{\mathsf{DM}}(T)] \geq \Omega\left(\min\{d/\sqrt{T}, 1\}\right). \tag{13}$$

In Section 4, we also instantiate Proposition 8 to derive a new complexity measure for DMSO.

# 4 Application to Interactive Decision Making: Bandit Learnability and Beyond

In this section, we focus on the DMSO setting and apply our general results (Theorem 1) to derive new lower and upper bounds for interactive decision making that go beyond the previous results based on the Decision-Estimation Coefficient [40, 42] by incorporating hardness of estimation.

**Background: Gaps between DEC-based and upper and lower bounds.** A fundamental open question of the DEC framework is whether the $\log|\mathcal{M}|$-gap between DEC lower and upper bounds in Theorem 4 can be closed. To highlight this gap in a more interpretable fashion, we re-state Theorem 4 in terms of a quantity we refer to as the *minimax sample complexity*. Let us focus on regret. Recall that for a fixed model class $\mathcal{M}$, the following notion of minimax regret (2) is the central objective of interest:

$$\mathbf{Reg}^\star(\mathcal{M}, T) := \inf_{\mathsf{ALG}} \sup_{M \in \mathcal{M}} \mathbb{E}^{M,\mathsf{ALG}}[\mathbf{Reg}_{\mathsf{DM}}(T)].$$

Given a parameter $\Delta > 0$, we define the *minimax sample complexity*

$$T^\star(\mathcal{M}, \Delta) := \inf_{T \geq 1} \{T : \mathbf{Reg}^\star(\mathcal{M}, T) \leq T\Delta\} \tag{14}$$

as the least value $T$ for which there exists an algorithm that achieves $\Delta T$ regret. Clearly, characterizing $T^\star(\mathcal{M}, \Delta)$ is equivalent to characterizing the minimax regret $\mathbf{Reg}^\star(\mathcal{M}, T)$.

Consider the following quantity induced by DEC for a class $\mathcal{M}$ and parameter $\Delta > 0$:

$$T^{\mathsf{DEC}}(\mathcal{M}, \Delta) = \inf_{\varepsilon \in (0,1)} \{\varepsilon^{-2} : \mathsf{r\text{-}dec}^{\mathsf{c}}_\varepsilon(\mathcal{M}) \leq \Delta\}. \tag{15}$$

With this definition, Theorem 4 is equivalent to the following characterization of the minimax sample complexity $T^\star(\mathcal{M}, \Delta)$:

$$T^{\mathsf{DEC}}(\mathcal{M}, \Delta) \lesssim T^\star(\mathcal{M}, \Delta) \lesssim T^{\mathsf{DEC}}(\mathcal{M}, \Delta) \cdot \log|\mathcal{M}|. \tag{16}$$

That is, Theorem 4 characterizes the minimax sample complexity up to a multiplicative $\log|\mathcal{M}|$ factor. Our main result in this section will be to use the fractional covering number and interactive Fano method (Theorem 1), to (i) tighten the above characterization (16) for various special cases of interest, and (ii) give a new characterization for $T^\star(\mathcal{M}, \Delta)$ in structured bandit problems which avoids spurious parameters such as $\log|\mathcal{M}|$ altogether.

## 4.1 New upper and lower bounds through the fractional covering number

For the a model class $\mathcal{M}$ and parameter $\Delta > 0$, we define the *fractional covering number* as follows:

$$\mathsf{N}_{\mathrm{frac}}(\mathcal{M}, \Delta) := \inf_{p \in \Delta(\Pi)} \sup_{M \in \mathcal{M}} \frac{1}{p(\pi : L(M, \pi) \leq \Delta)}. \tag{17}$$

Informally, the fractional covering number $\mathsf{N}_{\mathrm{frac}}(\mathcal{M}, \Delta)$ represents the best possible coverage over $\Delta$-optimal decisions that can be achieved through a single exploratory distribution in the face of an unknown model $M \in \mathcal{M}$. As we will now show, this quantity naturally arises as a lower bound on optimal risk through the interactive Fano method. We begin with the following assumption.

**Assumption 2** (Regular model class). *There exists a constant $C_{\mathrm{KL}} > 0$ and a reference model $\overline{M}$ such that $D_{\mathrm{KL}}(M(\pi) \| \overline{M}(\pi)) \leq C_{\mathrm{KL}}$ for all $M \in \mathcal{M}$ and $\pi \in \Pi$.*

Assumption 2 is a mild assumption on the boundedness of KL divergence. As an example, for structured bandits with means in $[0, 1]$ and Gaussian rewards, Assumption 2 holds with $C_{\mathrm{KL}} = \frac{1}{2}$. Details and more examples are provided in Appendix B.2. Our main lower bound based on the fractional covering number follows by specializing Theorem 1 to KL divergence.

**Theorem 10** (Fractional covering number lower bound). *Suppose that $\mathcal{M}$ satisfies Assumption 2 with parameter $C_{\mathrm{KL}} > 0$. Then for any algorithm $\mathtt{ALG}$ and $\Delta > 0$, unless $T \geq \frac{\log \mathsf{N}_{\mathrm{frac}}(\mathcal{M}, \Delta) - 2}{2 C_{\mathrm{KL}}}$, there exists $M^\star \in \mathcal{M}$ such that $\mathbb{P}^{M^\star, \mathtt{ALG}}[L(M^\star, \widehat{\pi}) \geq \Delta] \geq \frac{1}{2}$.*

In particular, for (generalized) no-regret learning, fractional covering number also implies a regret lower bound through Theorem 10. That is, for any algorithm to achieve $\Delta T$-regret, it is necessary to have $T = \Omega(\log \mathsf{N}_{\mathrm{frac}}(\mathcal{M}, 2\Delta))$. Combining this with Theorem 4, we conclude that boundedness of both the DEC and the fractional covering number is necessary for learning with any model class $\mathcal{M}$.

**Upper bounds based on the fractional covering number.** We now complement Theorem 10 by showing that for any reward maximization instance of the DMSO setting (Example 1), boundedness of the fractional covering number alone is also sufficient to derive *upper bounds* on the sample complexity of learning. The caveat is that while the lower bound is logarithmic in $\mathsf{N}_{\mathrm{frac}}(\mathcal{M}, \Delta)$, the upper bound will be polynomial.

**Theorem 11** (Fractional covering number upper bound). *Consider the reward maximization task (Example 1). There exists an algorithm that for any class $\mathcal{M}$ and $\Delta > 0$, ensures that with probability at least $1 - \delta$,*

$$\mathbf{Reg}_{\mathsf{DM}}(T) \leq T \cdot \Delta + O(\log(T/\delta)) \cdot \sqrt{T \cdot \mathsf{N}_{\mathrm{frac}}(\mathcal{M}, \Delta)}.$$

Combining Theorem 10 and Theorem 11 yields the following bounds on $T^\star(\mathcal{M}, \Delta)$ (omitting poly-logarithmic factors):

$$\frac{\log \mathsf{N}_{\mathrm{frac}}(\mathcal{M}, 2\Delta)}{C_{\mathrm{KL}}} \lesssim T^\star(\mathcal{M}, \Delta) \lesssim \frac{\mathsf{N}_{\mathrm{frac}}(\mathcal{M}, \Delta/2)}{\Delta^2}. \tag{18}$$

The gap between the lower and upper bounds of (18) is exponential; However, for model classes with $C_{\mathrm{KL}} = O(1)$, (18) suffices to characterize *finite-time learnability*. As a special case, we now show that fractional covering number characterizes the learnability of any structured bandit problem.

## 4.2 Application: Bandit learnability

We consider a structured bandit setting given by a reward function class $\mathcal{H} \subseteq (\Pi \to [0, 1])$. The protocol is as follows: For each round $t \in [T]$, the learner chooses a decision $\pi^t \in \Pi$, then receives a reward $r^t \sim \mathcal{N}(h_\star(\pi^t), 1)$ in response, where the mean reward function $h_\star \in \mathcal{H}$. This corresponds to an instance of the DMSO framework with induced model class $\mathcal{M}_{\mathcal{H}} = \{\pi \mapsto \mathcal{N}(h(\pi), 1) \mid h \in \mathcal{H}\}$. We define the fractional covering number for $\mathcal{H}$ via

$$\mathsf{N}_{\mathrm{frac}}(\mathcal{H}, \Delta) := \mathsf{N}_{\mathrm{frac}}(\mathcal{M}_{\mathcal{H}}, \Delta) = \inf_{p \in \Delta(\Pi)} \sup_{h \in \mathcal{H}} \frac{1}{p(\pi : h(\pi_h) - h(\pi) \leq \Delta)}, \tag{19}$$

where we denote $\pi_h := \arg\max_{\pi \in \Pi} h(\pi)$. This exactly coincides with the notion of *maximin volume* of Hanneke and Yang [45], which was shown to give a tight characterization of learnability for the special case of *noiseless binary-valued* structured bandits. We discuss the connection to Hanneke and Yang [45] in more detail in Appendix G.2.

It is straightforward to show that for any structured bandit problem, the induced class $\mathcal{M}_{\mathcal{H}}$ satisfies Assumption 2 with $C_{\mathrm{KL}} = \frac{1}{2}$ (Example 7). This leads to the following lower bound.

**Corollary 12** (Lower bound for stochastic bandits). *For the bandit model class $\mathcal{M}_{\mathcal{H}}$ defined as above, it holds that $T^{\star}(\mathcal{M}_{\mathcal{H}}, \Delta) \geq 2 \log \mathsf{N}_{\mathrm{frac}}(\mathcal{H}, \Delta) - 2$.*

Combining this result with the upper bound in Theorem 11, we obtain the following bounds on the minimax-optimal sample complexity for the structure bandit problem with class $\mathcal{H}$:

$$\log \mathsf{N}_{\mathrm{frac}}(\mathcal{H}, 2\Delta) \lesssim T^{\star}(\mathcal{M}_{\mathcal{H}}, \Delta) \lesssim \frac{\mathsf{N}_{\mathrm{frac}}(\mathcal{H}, \Delta/2)}{\Delta^2}. \tag{20}$$

This implies that $\mathsf{N}_{\mathrm{frac}}(\mathcal{H}, \Delta)$ characterizes learnability for structured bandits.

**Theorem 13** (Structured bandit learnability). *For a given reward function class $\mathcal{H}$, the bandit problem class $\mathcal{M}_{\mathcal{H}}$ is learnable for finite $T$ if and only if $\mathsf{N}_{\mathrm{frac}}(\mathcal{H}, \Delta) < +\infty$ for all $\Delta > 0$.*

We remark that the lower and upper bound in Eq. (20) cannot be improved in terms of the fractional covering number alone: (1) For $K$-armed bandits, we have $\mathsf{N}_{\mathrm{frac}}(\mathcal{H}, \Delta) \leq K$, meaning the upper bound is tight. (2) For $d$-dimensional linear bandits, we have $\log \mathsf{N}_{\mathrm{frac}}(\mathcal{H}, \Delta) = \Omega(d)$, meaning the lower bound is nearly tight. Nevertheless, the exponential gap in Eq. (20) can be partly mitigated by combining the fractional covering number with the DEC, as we will show in Section 4.3.

Our characterization bypasses the impossibility results of Hanneke and Yang [45]. Specifically, Hanneke and Yang [45] show that for *noiseless* structured bandit problems, there exist classes $\mathcal{H}$ for which bandit learnability is independent of the axioms of ZFC. Therefore, their results rule out the possibility of a characterization of noiseless bandit learnability through any *combinatorial dimension* [11] for the problem class. Our characterization is compatible with this result because the argument of Hanneke and Yang [45] relies on the noiseless nature of the bandit problem, and hence does not preclude a characterization for the noisy setting. Additional discussion is deferred to Appendix G.2.

### 4.3 Improved upper bounds for general decision making

To close this section, we derive tighter upper bounds that scale with $\log \mathsf{N}_{\mathrm{frac}}(\mathcal{M}, \Delta)$ by combining the fractional covering number with the Decision-Estimation Coefficient. For simplicity of presentation, we focus on regret minimization under the setting of Example 1, and we assume the following condition to simplify our bounds (the fully general upper bound is detailed in Appendix G.3).

**Assumption 3** (Regularity of constrained DEC). *A function $\mathsf{d} : [0, 1] \to \mathbb{R}$ is said to have moderate decay if $\mathsf{d}(\varepsilon) \geq 10\varepsilon \ \forall \varepsilon \in [0, 1]$, and there exists a constant $c \geq 1$ such that $c \frac{\mathsf{d}(\varepsilon)}{\varepsilon} \geq \frac{\mathsf{d}(\varepsilon')}{\varepsilon'}$ for all $\varepsilon' \geq \varepsilon$. We assume the function $\varepsilon \mapsto \mathsf{r\text{-}dec}_{\varepsilon}^{\mathsf{c}}(\mathrm{co}(\mathcal{M}))$, as a function of $\varepsilon$, satisfies moderate decay for a constant $c_{\mathrm{reg}} \geq 1$.*

This condition essentially requires that the DEC for $\mathrm{co}(\mathcal{M})$ exhibits moderate growth, which means that learning with $\mathrm{co}(\mathcal{M})$ is not "too easy". We now state our upper bound, which tightens Theorem 4 by replacing the $\log|\mathcal{M}|$ dependence in the upper bound with $\log \mathsf{N}_{\mathrm{frac}}(\mathcal{M}, \Delta)$ (with the caveat that the upper bound scales with the DEC for the *convexified* model class $\mathrm{co}(\mathcal{M})$).

**Theorem 14** (Upper bound with DEC and fractional covering number). *Consider the reward maximization variant of the DMSO setting. Let $\mathcal{M}$ be any class for which Assumption 3 holds, and assume that $\Pi$ is finite. Let $\bar{\varepsilon}(T) \asymp \sqrt{\log \mathsf{N}_{\mathrm{frac}}(\mathcal{M}, \Delta)/T}$. Then for any $\Delta > 0$, Algorithm 1 (see Appendix G.3) ensures that with high probability,*

$$\mathbf{Reg}_{\mathsf{DM}} \leq T \cdot \Delta + O\left(c_{\mathrm{reg}} T \sqrt{\log T}\right) \cdot \mathsf{r\text{-}dec}_{\bar{\varepsilon}(T)}^{\mathsf{c}}(\mathrm{co}(\mathcal{M})).$$

Restating this upper bound in terms of minimax sample complexity and combining it with the preceding lower bounds yields the following result.

**Theorem 15.** *For any class $\mathcal{M}$ that satisfies Assumption 2 and 3, we have*

$$\max\left\{T^{\mathsf{DEC}}(\mathcal{M}, \Delta), \frac{\log \mathsf{N}_{\mathrm{frac}}(\mathcal{M}, 2\Delta)}{C_{\mathrm{KL}}}\right\} \lesssim T^{\star}(\mathcal{M}, \Delta) \lesssim T^{\mathsf{DEC}}(\mathrm{co}(\mathcal{M}), \Delta) \cdot \log \mathsf{N}_{\mathrm{frac}}(\mathcal{M}, \Delta/2), \tag{21}$$

*up to dependence on $c_{\mathrm{reg}}$ and logarithmic factors.*

In particular, when the model class $\mathcal{M}$ is convex (i.e. $\mathrm{co}(\mathcal{M}) = \mathcal{M}$) and $C_{\mathrm{KL}} = O(1)$, Theorem 15 provides lower and upper bounds for learning with $\mathcal{M}$ that match up to a quadratic factor. Indeed, for convex model classes, the upper bound of (21) is always tighter than (16) (and also tighter than the result in Foster et al. [40, 41]), as by definition we have

$$\log \mathsf{N}_{\mathrm{frac}}(\mathcal{M}, \Delta) \leq \log \mathsf{N}_{\mathrm{frac}}(\mathcal{M}, 0) \leq \min\{\log|\mathcal{M}|, \log|\Pi|\}, \qquad \forall \Delta > 0.$$

As applications, we apply Theorem 15 to structured bandits and contextual bandits (Appendix G).

## Acknowledgements

We acknowledge support from ARO through award W911NF-21-1-0328, the Simons Foundation and the NSF through award DMS-2031883, as well as NSF PHY-2019786.

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

## Contents of Appendix

## A Related Work

To motivate our results, we briefly survey the most relevant lower bound techniques for estimation and decision making.

**Minimax bounds for statistical estimation.** There is a vast body of literature on minimax risk bounds for statistical estimation, including Hasminskii and Ibragimov [47], Bretagnolle and Huber [17], Birgé [15], Donoho and Liu [33], Cover and Thomas [29], Ibragimov and Has'Minskii [48], Tsybakov [81] as well as references therein. For minimax lower bounds, the most widely applied techniques are Le Cam's two-point method and convex-hull method [61], Assouad's lemma [8], and Fano's method [29]. Variants and applications of these three methods abound [2, 27, 67, 36]; Fano's method in particular has perhaps the largest number of variants, of which the most general version

we are aware of is due to Chen et al. [27], which is recovered by our interactive Fano method (cf. Proposition 2). Another celebrated thread, starting from the seminal work of Donoho and Liu [32], provides upper and lower bounds for a large class of non-parametric estimation problems based on Le Cam's two-point method through the study of a complexity measure known as the modulus of continuity [33, 34, 60, 67].

**Lower bounds for interactive learning.** There is a long line of work studying the fundamental limits of online learning and reinforcement learning (RL), including lower bounds for structured bandits [30, 73, 79, 57, 52, etc.], contextual bandits [72, 39, etc.], Markov Decision Processes (MDPs) [66, 31, 98, 92, 91, etc.], partially observable RL [53, 64, 25, 26, etc.], dynamical systems and control [80, 50, 78, 85, 99, etc.], and offline RL [71, 93, 88, 51, 24, 62, 86, etc.]. Most of these lower bounds are proven in a case-by-case basis, as the constructions of hard instances are specialized to the specific settings.

**Decision-Estimation Coefficient.** Toward a unifying understanding of the minimax complexity for interactive decision making problems, Foster et al. [40, 42] introduce Decision-Estimation Coefficient (DEC) as a complexity measure and show that it characterizes the minimax-optimal regret up to a $\log|\mathcal{M}|$ factor. The DEC can be viewed as an interactive counterpart of the modulus of continuity [32], and captures hardness of interactive decision making related to exploration, but not necessarily estimation. An active line of research has built on the DEC to encompass a variety of more general decision making settings [40, 41, 23, 42, 37, 44], including adversarial decision making [41], PAC decision making [23, 42], reward-free learning and preference-based learning [23], and multi-agent decision making and partial monitoring [37].

However, there is a remaining gap between the DEC lower and upper bounds [40, 42], which closely relates to the complexity of *estimation*. Specifically, through the DEC framework, the sample complexity (number of rounds required to achieve $\varepsilon$-risk) is characterized as

$$T^{\mathsf{DEC}}(\mathcal{M}, \varepsilon) \lesssim \text{\# sample complexity} \lesssim T^{\mathsf{DEC}}(\mathcal{M}, \varepsilon) \times \mathsf{Est}(\mathcal{M}),$$

where $T^{\mathsf{DEC}}(\mathcal{M}, \varepsilon)$ is a quantity measuring the complexity of *exploration*,[2] and $\mathsf{Est}(\mathcal{M})$ is a measure of the complexity of *online estimation* over $\mathcal{M}$. The dependency on $\mathsf{Est}(\mathcal{M})$ can be necessary: For example, in linear bandits, the optimal sample complexity scales as $d^2/\varepsilon^2$, while $T^{\mathsf{DEC}}(\mathcal{M}, \varepsilon) \asymp d/\varepsilon^2$, and $\mathsf{Est}(\mathcal{M}) \asymp d$. However, the complexity of estimation $\mathsf{Est}(\mathcal{M})$ is missing from the DEC lower bound. This gap remains one of the main open questions in the DEC approach.

One potential reason is that the DEC lower bound does *not* recover Fano's method or Assouad's lemma, as it essentially generalizes Le Cam's two-point method. More specifically, while the statistical estimation task is subsumed by the DMSO framework, the DEC lower bound specialized to this setting at best recovers Le Cam's two-point method. On the other hand, the $\Omega(d^2/\varepsilon^2)$ lower bound for linear bandits is typically proven through Assouad's lemma [18] or Fano's method [70], similar to its statistical estimation analog. Therefore, to close the remaining gap in the DEC approach, it is necessary to have a deeper understanding of the latter two methods in the interactive setting.

**Additional related work.** A large portion of the aforementioned lower bounds for interactive learning are proven using (variants of) the two-point method and can be recovered by the DEC lower bound approach [40, 42]. Beyond the two-point method, comparatively fewer lower bounds for interactive learning have been established using Assouad's lemma or Fano's method [21, 72, 3, 69, 39, 78, 70, etc.]. The approaches in these papers are specialized to the specific settings under consideration, and there is not a general principle through which Fano's method or Assouad's lemma can be lifted to handle interactivity. Indeed, the challenge of applying Fano's method in interactive contexts has been highlighted in various prior works, e.g., Arias-Castro et al. [7, Section 1.3] and Rajaraman et al. [70, Section 1.5.4].

The DEC is also closely related to a Bayesian complexity measure known as the information ratio [75, 76, 56, 55, etc.], which was originally introduced to analyze Bayesian algorithms such as posterior sampling. It is also related to a more recent generalization known as the *algorithmic information ratio* (AIR) [94], developed for frequentist algorithms. Additionally, the DEC is connected to asymptotic instance-dependent complexity, as explored by [87].

---

[2]Formally, the quantity $T^{\mathsf{DEC}}(\mathcal{M}, \varepsilon)$ here is the sample complexity implied by DEC (cf. Section 4).

# B   Additional Background on DMSO Framework

The DMSO framework (Section 1.1) encompasses a wide range of learning goals beyond the reward maximization setting [40, 42], including reward-free learning, model estimation, and preference-based learning [23], and also multi-agent decision making and partial monitoring [37]. We provide two examples below for illustration.

**Example 2** (Preference-based learning). In preference-based learning, each model $M \in \mathcal{M}$ is assigned with a comparison function $\mathbb{C}^M : \Pi \times \Pi \to \mathbb{R}$ (where $\mathbb{C}^M(\pi_1, \pi_2)$ typically the probability of $\tau_1 \succ \tau_2$ for $\tau_1 \sim (M, \pi_1)$, $\tau_2 \sim (M, \pi_2)$), and the risk function is specified by $L(M, \pi) = \max_{\pi^\star} \mathbb{C}^M(\pi^\star, \pi)$. Chen et al. [23] provide lower and upper bounds for this setting in terms of Preference-based DEC (PBDEC).

**Example 3** (Interactive estimation). In the setting of interactive estimation (a generalized PAC learning goal), each model $M \in \mathcal{M}$ is assigned with a parameter $\theta_M \in \Theta$, which is the parameter that the agent aims to estimate. The decision space $\Pi = \Pi_0 \times \Theta$, where each decision $\pi \in \Pi$ consists of $\pi = (\pi_0, \theta)$, where $\pi_0$ is the *explorative* policy to interact with the model[3], and $\theta$ is the estimator of the model parameter. In this setting, we define $L(M, \pi) = \mathrm{Dist}(\theta_M, \theta)$ for certain distance $\mathrm{Dist}(\cdot, \cdot)$.

This setting is an interactive version of the statistical estimation task, and it is also a generalization of the model estimation task studied in Chen et al. [23]. Natural examples include estimating some coordinates of the parameter $\theta$ in linear bandits. We provide nearly tight guarantee for this setting in Appendix E.2.

**Applicability of our results**   Our general interactive Fano method Lemma 5 applies to any generalized no-regret / PAC learning goal (Section 1.1). Therefore, our risk lower bound in terms of quantile PAC-DEC Theorem 6 and fractional covering number lower bound Theorem 10 both apply to any generalized learning goal. For a concrete example, see Appendix E.2 for the application to interactive estimation.

## B.1   Examples for statistical estimation (Section 2.1)

**Example 4** (Mean estimation). For the mean estimation task, the parameter space $\Theta \subseteq \mathbb{R}^d$, and for each $\theta \in \Theta$, $P_\theta = \mathcal{N}(\theta, I_d)$. The goal is to estimate the ground truth parameter $\theta^\star$, i.e., the decision space is $\mathcal{A} = \mathbb{R}^d$, and the loss is given by $L(\theta, A) = \|\theta - A\|$, where $\|\cdot\|$ is a norm over $\mathbb{R}^d$.

**Example 5** (Functional estimation). In the functional estimation task, a function $T : \Theta \to \mathbb{R}$ is given, and the goal is to estimate the value of $T(\theta^\star)$, i.e., the decision space is $\mathcal{A} = \mathbb{R}$, and the loss is $L(\theta, A) = |T(\theta) - A|$.

**Example 6** (Density estimation). In the density estimation task, the goal is to estimate $P_{\theta^\star}$, i.e., the decision space $\mathcal{A} \subseteq \Delta(\mathcal{Y})$, and the loss is given by $L(\theta, A) = D(P_\theta, A)$, where $D$ is a certain divergence (e.g., TV distance or KL divergence).

## B.2   Examples for Assumption 2

In this section, we provide three general types of model classes where Assumption 2 holds with mild $C_{\mathrm{KL}}$. It is worth noting that in Assumption 2, the reference model $\overline{M}$ does *not* necessarily belong to $\mathrm{co}(\mathcal{M})$.

**Example 7** (Gaussian bandits). Suppose that $\mathcal{H} \subseteq (\mathcal{A} \to [0, 1])$ is a class of mean value function, and $\mathcal{M}_{\mathcal{H}, \mathbb{V}}$ is the class of the model $M$ associated with a $h^M \in \mathcal{H}$:

$$M(\pi) = \mathcal{N}(h^M(\pi), 1), \qquad \pi \in \mathcal{A}.$$

Then, consider the reference model $\overline{M}$ given by $\overline{M}(\pi) = \mathcal{N}(0, 1) \forall \pi \in \mathcal{A}$. It is clear that for any $\pi$, and model $M \in \mathcal{M}_{\mathcal{H}, \mathbb{V}}$,

$$D_{\mathrm{KL}}(M(\pi) \parallel \overline{M}(\pi)) = \frac{1}{2} h^M(\pi)^2 \leq \frac{1}{2},$$

and hence Assumption 2 holds with $C_{\mathrm{KL}} = \frac{1}{2}$.

---

[3]In other words, $M(\pi)$ only depends on $\pi$ through $\pi_0$.

**Example 8** (Problems with finite observations). Suppose that the observation space $\mathcal{O}$ is finite. Then, consider the reference model $\overline{M}$ given by $\overline{M}(\pi) = \mathrm{Unif}(\mathcal{O}) \forall \pi \in \Pi$. It holds that

$$D_{\mathrm{KL}}(M(\pi) \parallel \overline{M}(\pi)) \leq \log |\mathcal{O}|, \qquad \forall \pi \in \Pi,$$

and hence Assumption 2 holds with $C_{\mathrm{KL}} = \log |\mathcal{O}|$.

Example 8 can further be generalized to infinite observation space, as long as every model in $\mathcal{M}$ admits a bounded density function with respect to the same base measure.

**Example 9** (Contextual bandits). Suppose that $\mathcal{H} \subseteq (\mathcal{C} \times \mathcal{A} \to [0,1])$ is a class of mean value function, and $\mathcal{M}_{\mathcal{H},\mathbb{V}}$ is the class of the model $M$ specified by a value function $h^M \in \mathcal{H}$ and a context distribution $\nu_M \in \Delta(\mathcal{C})$. More specifically, for any $\pi \in \Pi = (\mathcal{C} \to \mathcal{A})$, $M(\pi)$ is the distribution of $(c, a, r)$, generated by $c \sim \nu_M$, $a = \pi(c)$, and $r \sim \mathcal{N}(h^M(c,a), 1)$.

Then, consider the reference model $\overline{M}$ specified by $\nu_{\overline{M}} = \mathrm{Unif}(\mathcal{C})$ and $h^{\overline{M}} \equiv 0$. It is clear that for any $\pi$, and model $M \in \mathcal{M}_{\mathcal{H},\mathbb{V}}$,

$$D_{\mathrm{KL}}(M(\pi) \parallel \overline{M}(\pi)) \leq \log |\mathcal{C}| + 1$$

and hence Assumption 2 holds with $C_{\mathrm{KL}} = \log |\mathcal{C}| + 1$.

The factor of $\log |\mathcal{C}|$ in Example 9 is due to the definition (45) of $\log \mathsf{N}_{\mathrm{frac}}(\mathcal{H}, \Delta)$, where we take supremum over all context distribution $\mu$. This factor can be removed if we instead restrict the model class to have a common context distribution (i.e., the setting where context distribution is known or can be estimated from samples).

## C Technical Tools

The following lemma is the "chain rule" of Hellinger distance [49] (see also Duchi [35, Lemma 11.5.3] and Foster et al. [43, Lemma D.2]).

**Lemma C.1** (Sub-additivity for squared Hellinger distance). *Let $(\mathcal{X}_1, \mathcal{F}_1), \ldots, (\mathcal{X}_T, \mathcal{F}_T)$ be a sequence of measurable spaces, and let $\mathcal{X}^t = \prod_{i=1}^{t} \mathcal{X}_i$ and $\mathcal{F}^t = \bigotimes_{i=1}^{t} \mathcal{F}_i$. For each $t$, let $\mathbb{P}^t(\cdot \mid \cdot)$ and $\mathbb{Q}^t(\cdot \mid \cdot)$ be probability kernels from $(\mathcal{X}^{t-1}, \mathcal{F}^{t-1})$ to $(\mathcal{X}_t, \mathcal{F}_t)$.*

*Let $\mathbb{P}$ and $\mathbb{Q}$ be the laws of $X_1, \ldots, X_T$ under $X_t \sim \mathbb{P}^t(\cdot \mid X_{1:t-1})$ and $X_t \sim \mathbb{Q}^t(\cdot \mid X_{1:t-1})$ respectively. Then it holds that*

$$D_{\mathrm{H}}^2(\mathbb{P}, \mathbb{Q}) \leq 7 \, \mathbb{E}_{\mathbb{P}} \left[ \sum_{t=1}^{T} D_{\mathrm{H}}^2 \big( \mathbb{P}^t(\cdot \mid X_{1:t-1}), \mathbb{Q}^t(\cdot \mid X_{1:t-1}) \big) \right]. \tag{22}$$

In particular, given a $T$-round algorithm ALG and a model $M$, we can consider random variables $X_1 = (\pi^1, o^1), \cdots, X_T = (\pi^T, o^T)$. Then, $\mathbb{P}^{M,\mathrm{ALG}}(X_t = \cdot \mid X_{1:t-1})$ is the distribution of $(\pi^t, o^t)$, where $\pi^t \sim p^t(\cdot \mid \pi^1, o^1, \cdots, \pi^{t-1}, o^{t-1})$, and $o^t \sim M(\pi^t)$. Therefore, applying Lemma C.1 to $D_{\mathrm{H}}^2(\mathbb{P}^{M,\mathrm{ALG}}, \mathbb{P}^{\overline{M},\mathrm{ALG}})$ gives the following corollary.

**Corollary C.2.** *For any $T$-round algorithm ALG, it holds that*

$$\frac{1}{2} D_{\mathrm{TV}} \big( \mathbb{P}^{M,\mathrm{ALG}}, \mathbb{P}^{\overline{M},\mathrm{ALG}} \big)^2 \leq D_{\mathrm{H}}^2 \big( \mathbb{P}^{M,\mathrm{ALG}}, \mathbb{P}^{\overline{M},\mathrm{ALG}} \big) \leq 7T \cdot \mathbb{E}_{\pi \sim p_{\overline{M},\mathrm{ALG}}} \big[ D_{\mathrm{H}}^2 \big( M(\pi), \overline{M}(\pi) \big) \big].$$

**Lemma C.3** (Foster et al. [40, Lemma A.4]). *For any sequence of real-valued random variables $(X_t)_{t \leq T}$ adapted to a filtration $(\mathcal{F}_t)_{t \leq T}$, it holds that with probability at least $1 - \delta$, for all $t \leq T$,*

$$\sum_{s=1}^{t} -\log \mathbb{E} \left[ \exp(-X_s) \mid \mathcal{F}_{s-1} \right] \leq \sum_{s=1}^{t} X_s + \log(1/\delta).$$

**Lemma C.4.** *For any pair of random variable $(X, Y)$, it holds that*

$$\mathbb{E}_{X \sim \mathbb{P}_X} \big[ D_{\mathrm{H}}^2 \big( \mathbb{P}_{Y|X}, \mathbb{Q}_{Y|X} \big) \big] \leq 2 D_{\mathrm{H}}^2 \big( \mathbb{P}_{X,Y}, \mathbb{Q}_{X,Y} \big).$$

**Lemma C.5.** *Suppose that for a random variable $X$, its mean and variance under $\mathbb{P}$ is $\mu_{\mathbb{P}}$ and $\sigma_{\mathbb{P}}^2$, and its mean and variance under $\mathbb{Q}$ is $\mu_{\mathbb{Q}}$ and $\sigma_{\mathbb{Q}}^2$. Then it holds that*

$$|\mu_{\mathbb{P}} - \mu_{\mathbb{Q}}|^2 \leq 4\left(\sigma_{\mathbb{P}}^2 + \sigma_{\mathbb{Q}}^2 + \frac{1}{2}|\mu_{\mathbb{P}} - \mu_{\mathbb{Q}}|^2\right) D_{\mathrm{H}}^2\left(\mathbb{P}, \mathbb{Q}\right).$$

*In particular, when $\mu_{\mathbb{P}}, \mu_{\mathbb{Q}}, \sigma_{\mathbb{P}}, \sigma_{\mathbb{Q}} \in [0, 1]$, we have $D_{\mathrm{H}}^2\left(\mathbb{P}, \mathbb{Q}\right) \geq \frac{1}{10}|\mu_{\mathbb{P}} - \mu_{\mathbb{Q}}|^2$.*

*On the other hand, when $\mathbb{P} = \mathcal{N}(\mu_{\mathbb{P}}, 1), \mathbb{Q} = \mathcal{N}(\mu_{\mathbb{Q}}, 1)$, then $D_{\mathrm{H}}^2\left(\mathbb{P}, \mathbb{Q}\right) \leq \frac{1}{8}|\mu_{\mathbb{P}} - \mu_{\mathbb{Q}}|^2$.*

*Proof.* Let $\nu = \frac{\mathbb{P} + \mathbb{Q}}{2}$ be the common base measure and set $\mu = \frac{\mu_{\mathbb{P}} + \mu_{\mathbb{Q}}}{2}$. Then

$$\begin{aligned}
|\mu_{\mathbb{P}} - \mu_{\mathbb{Q}}|^2 &= |\mathbb{E}_{\mathbb{P}}[X - \mu] - \mathbb{E}_{\mathbb{Q}}[X - \mu]|^2 \\
&= \left|\mathbb{E}_{\nu}\left[\left(\frac{d\mathbb{P}}{d\nu} - \frac{d\mathbb{P}}{d\nu}\right)(X - \mu)\right]\right|^2 \\
&\leq \mathbb{E}_{\nu}\left[\left(\sqrt{\frac{d\mathbb{P}}{d\nu}} - \sqrt{\frac{d\mathbb{P}}{d\nu}}\right)^2\right] \mathbb{E}_{\nu}\left[\left(\sqrt{\frac{d\mathbb{P}}{d\nu}} + \sqrt{\frac{d\mathbb{P}}{d\nu}}\right)^2 (X - \mu)^2\right] \\
&\leq 2D_{\mathrm{H}}^2\left(\mathbb{P}, \mathbb{Q}\right) \cdot 2\left(\mathbb{E}_{\mathbb{P}}(X - \mu)^2 + \mathbb{E}_{\mathbb{Q}}(X - \mu)^2\right) \\
&= 4\left(\sigma_{\mathbb{P}}^2 + \sigma_{\mathbb{Q}}^2 + \frac{1}{2}|\mu_{\mathbb{P}} - \mu_{\mathbb{Q}}|^2\right) D_{\mathrm{H}}^2\left(\mathbb{P}, \mathbb{Q}\right).
\end{aligned}$$

$\square$

# D  Proofs from Section 3

In this section, we present proofs for the results in Section 3, except Section 3.2.

Before proceeding to proofs, we first discuss the classical Fano's method to motivate our approach.

## D.1  Additional background on classical Fano's method

To motivate our approach, which can be viewed as a generalization of the classical Fano method [29], let us briefly recall the classical approach and highlight some shortcomings. The classical Fano method applies to the statistical estimation setting Section 2.1 (a special case of ISDM), and takes the following form.

**Proposition D.1** (Classical Fano method). *Consider the statistical estimation setting (Section 2.1) with parameter space $\Theta$. Suppose that there exist $\theta_1, \ldots, \theta_m \in \Theta$ such that the following separation condition holds:*

$$L(\theta_i, a) + L(\theta_j, a) \geq 2\Delta, \quad \forall i \neq j \in [m], \forall a \in \mathcal{A}. \tag{23}$$

*Let $\mu$ be the uniform distribution over $\{\theta_1, \cdots, \theta_m\}$, and let $I_{\mu}(\theta; Y)$ denote the mutual information of $(\theta, Y) \sim \mathbb{P}_{\mu}$ generated by $\theta \sim \mu$ and $Y = (Y_1, \ldots, Y_n) \overset{\text{i.i.d}}{\sim} P_{\theta}$. Then for any algorithm $\mathsf{ALG}$, we have*

$$\mathbb{E}_{\theta \sim \mu, Y \sim P_{\theta}}\left[L(\theta, \mathsf{ALG}(Y))\right] \geq \Delta \cdot \sup_{\delta > 0}\{\delta : I_{\mu}(\theta; Y) < \mathrm{kl}(1 - \delta \parallel 1/m)\}, \tag{24}$$

*where the binary KL divergence is defined as $\mathrm{kl}(p \parallel q) = D_{\mathrm{KL}}(\mathrm{Bern}(p) \parallel \mathrm{Bern}(q))$. This implies the minimax lower bound*

$$\inf_{\mathsf{ALG}} \sup_{\theta \in \Theta} \mathbb{E}_{Y \sim P_{\theta}}\left[L(\theta, \mathsf{ALG}(Y))\right] \geq \Delta\left(1 - \frac{I_{\mu}(\theta; Y) + \log 2}{\log m}\right). \tag{25}$$

The $\log m$ factor in Eq. (25) reflects the complexity of estimation in the parameter space, which is a key concept we aim to incorporate into interactive decision making. Looking deeper, the "estimation complexity" term $\log m$ in Eq. (25) arises from the "quantile" parameter $1/m$ appearing in the Eq. (24). This parameter reflects the fact that under the separation condition (23), the following *quantile probability* is at most $1/m$ for any distribution $Y \sim \mathbb{Q}$:

$$\mathbb{P}_{\theta \sim \mu, Y \sim \mathbb{Q}}(L(\theta, \mathsf{ALG}(Y)) < \Delta) \leq \sup_a \mathbb{P}_{\theta \sim \mu}(L(\theta, a) < \Delta) \leq \frac{1}{m}. \tag{26}$$

Note that in this expression, $\theta$ is drawn from the uniform prior $\mu$ and $Y$ is drawn *independently* of $\theta$. To deduce Eq. (24), it suffices to choose $\mathbb{Q} = \mathbb{E}_{\theta \sim \mu} P_\theta$ and apply data-processing inequality:

$$I_\mu(\theta; Y) \geq \mathrm{kl}(\mathbb{P}_\mu(L(\theta, \mathsf{ALG}(Y)) < \Delta) \parallel \mathbb{P}_{\theta \sim \mu, Y \sim \mathbb{Q}}(L(\theta, \mathsf{ALG}(Y)) < \Delta))$$
$$\geq \mathrm{kl}(\mathbb{P}_\mu(L(\theta, \mathsf{ALG}(Y)) < \Delta) \parallel 1/m).$$

In particular, for any $\delta \in (0,1)$ such that $I_\mu(\theta; Y) \leq \mathrm{kl}(1 - \delta \parallel 1/m)$, we have $\mathbb{P}_\mu(L(\theta, \mathsf{ALG}(Y)) < \Delta) \leq 1 - \delta$, using the monotonicity of the KL divergence. This argument gives Eq. (24) immediately, and by choosing $\delta_\star = 1 - \frac{I_\mu(\theta;Y) + \log 2}{\log m}$ in Eq. (24), we arrive in the canonical statement in Eq. (25).

To summarize, the structure of the classical Fano lower bound involves (i) a **prior** $\mu$, (ii) a **reference distribution** $\mathbb{Q} = \mathbb{E}_{\theta \sim \mu} P_\theta$, and (iii) a **quantile parameter** $\delta$ determined by the (iv) **separation condition** (23) in the argument above. Crucially, we understand that the complexity of estimation $\log m$ arises from the *quantile probability* $\mathbb{P}_{\theta \sim \mu, Y \sim \mathbb{Q}}(L(\theta, \mathsf{ALG}(Y)) < \Delta)$, and the only use of the traditional separation condition (23) is to further bound this quantile by $1/m$ as in Eq. (26).

Having gained these insights into classical Fano method, we would like to point out several limitations:

- First, in the form above, it is only applicable to statistical estimation rather than general interactive settings.

- Second, it relies on mutual information due to the choice of the reference distribution, which depends on the evolution of the algorithm over all $T$ rounds in interactive settings, making it difficult to analyze in many interactive problems.

- Third, and perhaps most importantly, the separation condition (23) must hold for an arbitrary decision $a$. This "hard" separation condition is unlikely to hold for general model classes, as noted throughout the DEC approach [42, Remark 2.3] line of work.

To address these shortcomings, we make use of core concepts (prior, reference distribution, quantile parameter, separation condition) above, but adopt a new perspective that emphasizes and generalizes the role of the quantile probability $\mathbb{P}_{\theta \sim \mu, Y \sim \mathbb{Q}}(L(\theta, \mathsf{ALG}(Y)) < \Delta)$.

### D.2 Proof of Theorem 1

In the following, we fix a prior $\mu \in \Delta(\mathcal{M})$, parameter $\Delta > 0$, $f$-divergence $D_f$, and an algorithm $\mathsf{ALG}$. For simplicity, we denote $D_f(x, y) = D_f(\mathrm{Bern}(x), \mathrm{Bern}(y))$ for $x, y \in [0, 1]$.

We first prove the following quantile lower bound:

$$\mathbb{P}_{M \sim \mu, X \sim \mathbb{P}^{M, \mathsf{ALG}}}(L(M, X) \geq \Delta) \geq \sup_{\mathbb{Q} \in \Delta(\mathcal{X}), \delta \in [0,1]} \left\{ \delta : \mathbb{E}_{M \sim \mu}[D_f(\mathbb{P}^{M, \mathsf{ALG}}, \mathbb{Q})] < \mathsf{d}_{f, \delta}(\rho_{\Delta, \mathbb{Q}}) \right\}.$$

(27)

We only need to prove the following claim.

**Claim.** Suppose that there exists a reference distribution $\mathbb{Q}$ such that

$$\mathsf{d}_{f, \delta}(\rho_{\Delta, \mathbb{Q}}) > \mathbb{E}_{M \sim \mu} D_f(\mathbb{P}^{M, \mathsf{ALG}}, \mathbb{Q}),$$

then $\mathbb{P}_{M \sim \mu, X \sim \mathbb{P}^{M, \mathsf{ALG}}}(L(M, X) \geq \Delta) \geq \delta$.

We denote $\bar{\rho}_\Delta = \mathbb{P}_{M \sim \mu, X \sim \mathbb{P}^{M, \mathsf{ALG}}}(L(M, X) < \Delta)$, and recall that we define $\rho_{\Delta, \mathbb{Q}} = \mathbb{P}_{M \sim \mu, X \sim \mathbb{Q}}(L(M, X) < \Delta)$. We then consider the following two distributions over $\mathcal{M} \times \mathcal{X}$:

$$P_0 : M \sim \mu, X \sim \mathbb{P}^{M, \mathsf{ALG}}, \qquad P_1 : M \sim \mu, X \sim \mathbb{Q}.$$

By the data processing inequality of $f$-divergence, we have

$$D_f(\bar{\rho}_\Delta, \rho_{\Delta, \mathbb{Q}}) \leq D_f(P_0, P_1) = \mathbb{E}_{M \sim \mu} D_f(\mathbb{P}^{M, \mathsf{ALG}}, \mathbb{Q}).$$

Therefore, using $\mathsf{d}_{f, \delta}(\rho_{\Delta, \mathbb{Q}}) > \mathbb{E}_{M \sim \mu} D_f(\mathbb{P}^{M, \mathsf{ALG}}, \mathbb{Q})$, we know that $\mathsf{d}_{f, \delta}(\rho_{\Delta, \mathbb{Q}}) > D_f(\bar{\rho}_\Delta, \rho_{\Delta, \mathbb{Q}})$. In particular, this implies $\rho_{\Delta, \mathbb{Q}} < 1 - \delta$, and

$$D_f(\bar{\rho}_\Delta, \rho_{\Delta, \mathbb{Q}}) < D_f(1 - \delta, \rho_{\Delta, \mathbb{Q}})$$

Hence, we consider two cases: (1) $\bar{\rho}_\Delta \leq \rho_{\Delta,\mathbb{Q}}$, and (2) $\bar{\rho}_\Delta > \rho_{\Delta,\mathbb{Q}}$. For case (1), we have $\bar{\rho}_\Delta \leq \rho_{\Delta,\mathbb{Q}} < 1 - \delta$. For case (2), we can use the monotone property of $D_f$ (Lemma D.2), which also implies $\bar{\rho}_\Delta < 1 - \delta$.

Therefore, it holds that $\bar{\rho}_\Delta < 1 - \delta$, and

$$\mathbb{P}_{M\sim\mu,X\sim\mathbb{P}^{M,\text{ALG}}}(L(M,X) \geq \Delta) = 1 - \bar{\rho}_\Delta > \delta.$$

Hence, the proof of Eq. (27) is completed. The in-expectation lower bounds then follows from the fact that

$$\mathbb{E}_{M\sim\mu}\mathbb{E}_{X\sim\mathbb{P}^{M,\text{ALG}}}[L(M,X)] \geq \Delta \cdot \mathbb{P}_{M\sim\mu,X\sim\mathbb{P}^{M,\text{ALG}}}(L(M,X) \geq \Delta).$$

$\square$

**Lemma D.2.** *For $x, y \in [0,1]$, the quantity $D_f(x,y)$ is increasing with respect to $x$ when $x \geq y$.*

**Proof of Lemma D.2** Fix any $1 \geq x > z \geq y \geq 0$, we define

$$p = y \cdot \frac{x-z}{x-y} \in [0,1], \qquad q = 1 - (1-y) \cdot \frac{x-z}{x-y} \in [0,1].$$

Then, by definition,

$$p(1-x) + qx = z, \qquad p(1-y) + qy = y,$$

and hence for the channel $P$ from $\{0,1\}$ to itself given by $P(\cdot|0) = \text{Bern}(p), P(\cdot|1) = \text{Bern}(q)$, it holds that

$$P \circ \text{Bern}(x) = \text{Bern}(z), \qquad P \circ \text{Bern}(y) = \text{Bern}(y).$$

Therefore, by data-processing inequality, we have

$$D_f(\text{Bern}(z), \text{Bern}(y)) \leq D_f(\text{Bern}(x), \text{Bern}(y)).$$

This is the desired result. $\square$

### D.3   Proof of Proposition 2

Fix the parameter $\Delta > 0$ and let $\mu \in \Delta(\mathcal{M})$ be given. To apply Theorem 1, we consider KL divergence (corresponding to $f(x) = x \log x$) and choose the reference distribution

$$\mathbb{Q} = \mathbb{E}_{M\sim\mu}\mathbb{P}^{M,\text{ALG}}.$$

Then, by the choice of $\mathbb{Q}$ and definition of KL-divergence, we have

$$\mathbb{E}_{M\sim\mu}D_{\text{KL}}(\mathbb{P}^{M,\text{ALG}} \| \mathbb{Q}) = I_{\mu,\text{ALG}}(M;X),$$

and by definition, we have

$$\rho_{\Delta,\mathbb{Q}} = \mathbb{P}_{M\sim\mu,X'\sim\mathbb{Q}}(L(M,X') < \Delta) \leq \sup_x \mu(M : L(M,x) < \Delta), \tag{28}$$

By Theorem 1, for any $\delta \in (0,1)$ such that $I_{\mu,\text{ALG}}(M;X) < \text{kl}(1-\delta \| \rho_{\Delta,\mathbb{Q}})$, we have

$$\mathbb{E}_{M\sim\mu}\mathbb{E}_{X\sim\mathbb{P}^{M,\text{ALG}}}[L(M,X)] \geq \delta\Delta.$$

In particular, we may choose

$$\delta_\star := 1 + \frac{I_{\mu,\text{ALG}}(M;X) + \log 2}{\log \sup_x \mu(M : L(M,x) < \Delta)}.$$

As long as $\delta_\star > 0$, we have $I_{\mu,\text{ALG}}(M;X) < \text{kl}(1 - \delta_\star \| \rho_{\Delta,\mathbb{Q}})$, and hence $\mathbb{E}_{M\sim\mu}\mathbb{E}_{X\sim\mathbb{P}^{M,\text{ALG}}}[L(M,X)] \geq \delta_\star\Delta$. This gives the desired lower bound (note that if $\delta_\star \leq 0$, there is nothing to prove). $\square$

**D.4 Proof of Proposition 3**

We recover this result by applying Theorem 1 with TV distance. We first frame the problem in the ISDM framework. Consider the "enlarged" model class $\mathcal{M} = \{M_\nu : \nu \in \Delta(\Theta)\}$, where for any algorithm $\mathtt{ALG} : \mathcal{Y}^{\otimes n} \to \mathcal{A}$, the distribution $\mathbb{P}^{M_\nu, \mathtt{ALG}}$ is given by

$$X = (Y, \mathtt{ALG}(Y)) \sim \mathbb{P}^{M_\nu, \mathtt{ALG}} : \theta \sim \nu, Y = (Y_1, \cdots, Y_n) \overset{\mathrm{i.i.d}}{\sim} P_\theta.$$

In other words, $\mathbb{P}^{M_\nu, \mathtt{ALG}}$ is the distribution induced by the prior $\nu$ and the algorithm $\mathtt{ALG}$. We then extend the loss function to $L : \mathcal{M} \times \mathcal{X} \to \mathbb{R}_+$ as

$$L(M_\nu, X) := \inf_{\theta \in \mathrm{supp}(\nu)} L(\theta, \mathtt{ALG}(Y)), \qquad \forall X = (Y, \mathtt{ALG}(Y)), \ \nu \in \Delta(\Theta).$$

By the separation condition (7), we have $L(M_{\nu_0}, X) + L(M_{\nu_1}, X) \geq 2\Delta$ for any $\nu_0 \in \Delta(\Theta_0)$, $\nu_1 \in \Delta(\Theta_1)$. Therefore, choosing the prior $\mu = \mathrm{Unif}(\{M_{\nu_0}, M_{\nu_1}\})$ and the reference distribution $\mathbb{Q} = \mathbb{E}_{M \sim \mu} \mathbb{P}^{M, \mathtt{ALG}}$ gives

$$\rho_{\Delta, \mathbb{Q}} = \mathbb{P}_{M \sim \mu, X \sim \mathbb{Q}}(L(M, X) < \Delta) \leq 1/2,$$

and by the data-processing inequality,

$$\mathbb{E}_{M \sim \mu}[D_{\mathrm{TV}}\left(\mathbb{P}^{M, \mathtt{ALG}}, \mathbb{Q}\right)] = \frac{1}{2}\left(D_{\mathrm{TV}}\left(\mathbb{P}^{M_{\nu_0}, \mathtt{ALG}}, \mathbb{Q}\right) + D_{\mathrm{TV}}\left(\mathbb{P}^{M_{\nu_1}, \mathtt{ALG}}, \mathbb{Q}\right)\right)$$

$$\leq \frac{1}{2} D_{\mathrm{TV}}\left(\mathbb{P}^{M_{\nu_0}, \mathtt{ALG}}, \mathbb{P}^{M_{\nu_1}, \mathtt{ALG}}\right) \leq \frac{1}{2} D_{\mathrm{TV}}\left(P_{\nu_0}^{\otimes n}, P_{\nu_1}^{\otimes n}\right).$$

Therefore, for any $0 \leq \delta < \frac{1}{2} - \frac{1}{2} D_{\mathrm{TV}}\left(P_{\nu_0}^{\otimes n}, P_{\nu_1}^{\otimes n}\right)$, we have

$$\mathbb{E}_{M \sim \mu}[D_{\mathrm{TV}}(\mathbb{P}^{M, \mathtt{ALG}}, \mathbb{Q})] \leq \mathsf{d}_{\mathrm{TV}, \delta}(\rho_{\Delta, \mathbb{Q}}),$$

and hence applying Theorem 1 gives

$$\mathbb{E}_{\theta \sim \frac{\nu_0 + \nu_1}{2}} \mathbb{E}_{Y \sim P_\theta}[L(\theta, \mathtt{ALG}(Y))] \geq \mathbb{E}_{M \sim \mu} \mathbb{E}_{X \sim \mathbb{P}^{M, \mathtt{ALG}}}[L(M, X)] \geq \frac{\Delta}{2}\left(1 - D_{\mathrm{TV}}\left(P_{\nu_0}^{\otimes n}, P_{\nu_1}^{\otimes n}\right)\right).$$

Taking the supremum over $\nu_0 \in \Delta(\Theta_0)$ and $\nu_1 \in \Delta(\Theta_1)$ gives the desired result. $\qquad\square$

We can use the Le Cam convex hull method to recover the classic two-point method and Assouad's method, as follows.

**Example 10** (Le Cam's two-point method)**.** In Proposition 3, we can take the distribution $\nu_0$ ($\nu_1$) to be supported on a single point in $\Theta_0$ ($\Theta_1$), to recover the classical two-point method. Concretely, under the setting and assumption of Proposition 3, we have the following two-point lower bound:

$$\inf_{\mathtt{ALG}} \sup_{\theta \in \Theta} \mathbb{E}_{Y \sim P_\theta} L(\theta, \mathtt{ALG}(Y)) \geq \frac{\Delta}{2} \max_{\theta_0 \in \Theta_0, \theta_1 \in \Theta_1} \left(1 - D_{\mathrm{TV}}\left(P_{\theta_0}^{\otimes n}, P_{\theta_1}^{\otimes n}\right)\right),$$

where $P_\theta^{\otimes n}$ is the distribution of $Y = (Y_1, \cdots, Y_n) \overset{\mathrm{i.i.d}}{\sim} P_\theta$. $\qquad\triangleleft$

**Example 11** (Assouad's method)**.** Suppose that $\Theta = \{-1, 1\}^d$ for some $d \geq 1$, and that the loss function has the following coordinate-wise structure:

$$L(\theta, a) = \sum_{j=1}^d L_j(\theta, a), \qquad \forall \theta \in \Theta, a \in \mathcal{A}.$$

We write $\theta \sim_j \theta'$ if $\theta$ and $\theta'$ differ only in the $j$-th coordinate. Assume that the following separation condition holds for all $j \in [d]$:

$$L_j(\theta, a) + L_j(\theta', a) \geq 2\Delta, \quad \forall a \in \mathcal{A}, \theta \sim_j \theta'.$$

To apply Proposition 3, we consider $\Theta_i^j = \{\theta : \theta_j = i\}$ for $i \in \{0, 1\}$ and $j \in [d]$. Then, for each $j \in [d]$, the separation condition (7) holds for the loss $L_j$ and the subsets $\Theta_0^j, \Theta_1^j$. Therefore, applying Proposition 3 for each $j \in [d]$ with $\nu_0^j = \mathrm{Unif}(\Theta_0^j)$ and $\nu_1^j = \mathrm{Unif}(\Theta_1^j)$ gives the following Assouad-type lower bound:

$$\inf_{\mathtt{ALG}} \sup_{\theta \in \Theta} \mathbb{E}_{Y \sim P_\theta} L(\theta, \mathtt{ALG}(Y)) \geq \frac{d\Delta}{2} \min_{\exists j : \theta \sim_j \theta'} \left(1 - D_{\mathrm{TV}}\left(P_\theta^{\otimes n}, P_{\theta'}^{\otimes n}\right)\right).$$

$\triangleleft$

### D.5 Proof of Corollary 9

Consider the following setup of linear bandits: let $\theta^\star \in \mathbb{R}^d$ be an unknown parameter. At time $t$, the learner chooses an action $\pi^t \in \{\pi \in \mathbb{R}^d : \|\pi\|_2 \le 1\}$ and receives a Gaussian reward $r^t \sim \mathcal{N}(\langle \theta^\star, \pi^t \rangle, 1)$. For $T \in \mathbb{N}$, let $\mathcal{H}^T = (\pi^1, r^1, \cdots, \pi^T, r^T)$ be the observed history up to time $T$. The central claim of this section is the following upper bound on the mutual information.

**Theorem D.3.** *For any $r > 0$, we define the prior $\mu_r$ over $\mathbb{B}^d(r)$ by*

$$\mu_r : \theta^\star \sim \mathcal{N}\left(0, \frac{r^2}{4d}I_d\right) \mid \|\theta^\star\| \le r.$$

*Then for any algorithm* ALG, *we have*

$$I_{\mu_r, \mathsf{ALG}}(\theta^\star; \mathcal{H}^T) \le d \log\left(1 + \frac{r^2 T}{4d^2}\right).$$

*Proof.* Denote $\lambda = \frac{r^2}{4}$. We first prove that if $\theta^\star \sim \mu = \mathcal{N}(0, \lambda I_d/d)$, then

$$I_{\mu, \mathsf{ALG}}(\theta^\star; \mathcal{H}^T) \le \frac{d}{2} \log\left(1 + \frac{\lambda T}{d^2}\right). \tag{29}$$

By the Bayes rule, the posterior distribution of $\theta^\star$ conditioned on $(\mathcal{H}^{t-1}, \pi^t)$ is

$$p(\theta^\star \mid \mathcal{H}^{t-1}, \pi^t) \propto \exp\left(-\frac{d\|\theta^\star\|_2^2}{2\lambda} - \frac{1}{2}\sum_{s<t}(r^s - \langle \theta^\star, \pi^s \rangle)^2\right),$$

which is a Gaussian distribution with covariance $(\Sigma^{t-1})^{-1}$, where

$$\Sigma^{t-1} = \frac{d}{\lambda}I_d + \sum_{s<t} \pi^s (\pi^s)^\top.$$

Therefore, by the chain rule of mutual information, we have

$$\begin{aligned}
I_{\mu, \mathsf{ALG}}(\theta^\star; \mathcal{H}^T) &= \sum_{t=1}^T I_{\mu, \mathsf{ALG}}(\theta^\star; r^t \mid H^{t-1}, \pi^t) \\
&= \sum_{t=1}^T \mathbb{E}^{\mu, \mathsf{ALG}}\left[\frac{1}{2}\log\left(1 + (\pi^t)^\top (\Sigma^{t-1})^{-1} \pi^t\right)\right] \\
&= \mathbb{E}^{\mu, \mathsf{ALG}}\left[\frac{1}{2}\sum_{t=1}^T \log \frac{\det(\Sigma^t)}{\det(\Sigma^{t-1})}\right] \\
&= \mathbb{E}^{\mu, \mathsf{ALG}}\left[\frac{1}{2}\log \frac{\det(\Sigma^T)}{(d/\lambda)^d}\right] \\
&\le \mathbb{E}^{\mu, \mathsf{ALG}}\left[\frac{d}{2}\log \frac{\mathrm{Tr}(\Sigma^T)/d}{d/\lambda}\right] \\
&\le \frac{d}{2}\log\left(1 + \frac{\lambda T}{d^2}\right),
\end{aligned}$$

which is exactly Eq. (29).

Next we deduce the claimed result from Eq. (29). Consider the random variable $Z = \mathbf{1}\{\|\theta^\star\|_2 \le r\} \in \{0, 1\}$, and then

$$\begin{aligned}
\frac{d}{2}\log\left(1 + \frac{\lambda T}{d^2}\right) &\ge I_{\mu, \mathsf{ALG}}(\theta^\star; \mathcal{H}^T) \\
&\ge I_{\mu, \mathsf{ALG}}(\theta^\star; \mathcal{H}^T \mid Z) \\
&\ge \mathbb{P}(Z = 1) \cdot I_{\mu_r, \mathsf{ALG}}(\theta^\star; \mathcal{H}^T \mid Z = 1)
\end{aligned}$$

$$= \mathbb{P}_\mu(\|\theta^\star\|_2 \leq r) \cdot I_{\mu_r, \text{ALG}}(\theta^\star; \mathcal{H}^T).$$

Here the first inequality is Eq. (29), the second inequality follows from $I(X;Y) - I(X;Y \mid f(X)) = I(f(X);Y) - I(f(X);Y \mid X) \geq 0$, the third identity follows from the definition of conditional mutual information. Finally, noticing that $\mathbb{P}_\mu(\|\theta^\star\|_2 \leq r) \geq \frac{1}{2}$ by concentration of $\chi_d^2$ random variable, we arrive at the desired statement. $\square$

Next we show how to translate the mutual information upper bound in Theorem D.3 to lower bounds of estimation and regret.

**Theorem D.4.** *Let $T \geq 1$, $r = \min\left\{\frac{c_0 d}{\sqrt{T}}, 1\right\}$ for a small absolute constant $c_0$, and consider the prior $\mu = \mu_r$. For any $T$-round algorithm with output $\widehat{\pi}$, Proposition 8 implies that*

$$\mathbb{E}^{\mu, \text{ALG}}\left[\left\|\widehat{\pi} - \frac{\theta^\star}{\|\theta^\star\|}\right\|^2\right] \geq \frac{1}{4}.$$

*Therefore, we may deduce that*

$$\sup_{M^\star \in \mathcal{M}} \mathbb{E}^{M^\star, \text{ALG}}[\mathbf{Risk}_{\text{DM}}(T)] \gtrsim \min\left\{\frac{d}{\sqrt{T}}, 1\right\}.$$

*Proof.* We first prove the first inequality by applying Proposition 8 to the following risk function

$$\tilde{L}(M_\theta, \pi) = \|\pi - \text{normalize}(\theta)\|_2^2,$$

where we denote $\text{normalize}(\theta) = \frac{\theta}{\|\theta\|} \in \mathbb{B}^d(1)$. Notice that for $\theta \in \Theta$, we have

$$L(M_\theta, \pi) = \|\theta\| - \langle \theta, \pi \rangle \geq \|\theta\| \cdot \left\|\pi - \frac{\theta}{\|\theta\|}\right\|^2 = \|\theta\| \cdot \tilde{L}(M_\theta, \pi).$$

For $\Delta \in (0, 1)$, we first claim that

$$\rho_\Delta := \sup_\pi \mu\left(\theta : \tilde{L}(M_\theta, \pi) \leq \Delta\right) = O\left(\sqrt{d}\Delta^{(d-1)/2}\right). \tag{30}$$

To see so, by symmetry of Gaussian distribution, we know for fixed any $\pi \in \mathbb{R}^d$,

$$\mu\left(\theta : \tilde{L}(M_\theta, \pi) \leq \Delta\right) = \mathbb{P}_{\theta \sim \text{Unif}(\mathbb{S}^{d-1})}\left(\theta : \|\theta - \pi\|^2 \leq \Delta\right),$$

and hence we can instead consider the uniform distribution over the sphere $\mathbb{S}^{d-1}$. By rotational invariance, we may assume that $\pi = (x, 0, \cdots, 0)$, with $x \geq 0$. Then

$$\left\{\theta \in \mathbb{S}^{d-1} : \|\theta - \pi\|_2^2 \leq \Delta\right\} = \left\{\theta \in \mathbb{S}^{d-1} : \theta_1 \geq \frac{x^2 + 1 - \Delta}{2x}\right\} \subseteq \left\{\theta \in \mathbb{S}^{d-1} : \theta_1 \geq \sqrt{1 - \Delta}\right\}.$$

By Bubeck et al. [19, Section 2], for $\theta \sim \text{Unif}(\mathbb{S}^{d-1})$, the density of $\theta_1 \in [-1, 1]$ is given by

$$f(\theta_1) = \frac{\Gamma(d/2)}{\Gamma((d-1)/2)\sqrt{\pi}}(1 - \theta_1^2)^{(d-3)/2}.$$

Therefore,

$$\rho_\Delta \leq \int_{\sqrt{1-\Delta}}^1 f(\theta_1)d\theta_1 = O(\sqrt{d}) \cdot (1 - \sqrt{1-\Delta})\Delta^{(d-3)/2} = O\left(\sqrt{d}\Delta^{(d-1)/2}\right).$$

With the upper bound (30) of $\rho_\Delta$, we know that for $\Delta = \frac{1}{2}$, it holds

$$\log(1/\rho_\Delta) \geq 2I_\mu(T),$$

as long as $c_0$ is a sufficiently small constant. Therefore, Proposition 8 gives that

$$\mathbb{E}^{\mu, \text{ALG}}\left[\|\widehat{\pi} - \text{normalize}(\theta)\|^2\right] = \mathbb{E}^{\mu, \text{ALG}}\left[\tilde{L}(M_\theta, \widehat{\pi})\right] \geq \frac{1}{4}.$$

This completes the proof of the first inequality.

Finally, using the fact that $\mathbb{P}_{\theta^\star \sim \mu}(\|\theta^\star\| \leq c_1 r) \leq \frac{1}{100}$ for a small absolute constant $c_1$, we can conclude that

$$\sup_{M^\star \in \mathcal{M}} \mathbb{E}^{M^\star, \text{ALG}}[\mathbf{Risk}_{\text{DM}}(T)] \geq \mathbb{E}^{\mu, \text{ALG}}[L(M_\theta, \pi)] \geq \frac{c_1 r}{8} = \Omega\left(\min\left\{\frac{d}{\sqrt{T}}, 1\right\}\right).$$

This is the desired result. $\square$

# E  Additional Results from Section 3.2

In addition to the reward-maximization setting (Example 1), we also introduce a slightly more general setting. In this setting, we assume that for each model $M \in \mathcal{M}$, the risk function is $L(M, \pi) = f^M(\pi_M) - f^M(\pi)$, but $f^M$ is not assumed to be the expected reward function (Example 1). Instead, we only require $f^M$ satisfying the following assumption, where $\mathcal{M}^+ \subseteq (\Pi \to \Delta(\mathcal{O}))$ is a pre-specified model class of reference models that contains $\mathrm{co}(\mathcal{M})$ (following Foster et al. [42]). The lower bound we prove can be stronger by allowing $\mathcal{M}^+$ to be a larger model class.

**Assumption 4.** *Let $\mathcal{M}^+ \subseteq (\Pi \to \Delta(\mathcal{O}))$ be a given class of reference models, such that $\mathrm{co}(\mathcal{M}) \subseteq \mathcal{M}^+$. For any $M \in \mathcal{M}$, the risk function takes form $L(M, \pi) = f^M(\pi_M) - f^M(\pi)$ for some functional $f^M : \Pi \to \mathbb{R}$, so that $f^M$ can be extended to $\mathcal{M}^+$, such that for any model $M \in \mathcal{M}$ and reference model $\overline{M} \in \mathcal{M}^+$ we have*

$$|f^M(\pi) - f^{\overline{M}}(\pi)| \leq C_{\mathrm{r}} D_{\mathrm{H}}\big(M(\pi), \overline{M}(\pi)\big), \qquad \forall \pi \in \Pi. \tag{31}$$

In some cases, considering a larger reference model class $\mathcal{M}^+$ can be convenient for proving lower bounds, see e.g., Appendix B.2 and Appendix H.7.

## E.1  Recovering DEC-based regret lower bounds

In this section, we demonstrate how our general lower bound approach recovers the regret lower bounds of Foster et al. [42], Glasgow and Rakhlin [44]. We first state our lower bound in terms of constrained DEC in the following theorem.

**Theorem E.1.** *Under the reward maximization setting (Example 1), for any $T$-round algorithm* ALG, *there exists $M^\star \in \mathcal{M}$ such that*

$$\mathbf{Reg}_{\mathsf{DM}} \geq \frac{T}{2} \cdot \Big(\mathsf{r\text{-}dec}^{\mathrm{c}}_{\underline{\varepsilon}(T)}(\mathcal{M}) - 6\underline{\varepsilon}(T)\Big) - 1$$

*with probability at least $0.01$ under $\mathbb{P}^{M^\star, \mathsf{ALG}}$, where $\underline{\varepsilon}(T) = \frac{1}{100\sqrt{T}}$.*

Theorem E.1 immediately yields an in-expectation regret lower bound in terms of constrained DEC. It also shaves off the unnecessary logarithmic factors in the lower bound of Foster et al. [42, Theorem 2.2].

For the remainder of this section, we sketch how we prove Theorem E.1 in a slightly more general setting (Assumption 4), following Appendix F.3. Before providing our regret lower bounds, we first present several important definitions.

**Definition of quantile regret-DEC**   We note that it is possible to directly modify the definition of quantile PAC-DEC (12), and then apply Theorem 6 to obtain an analogous regret lower bound immediately. However, as Foster et al. [42] noted, the "correct" notion of regret-DEC (cf. Eq. (8)) turns out to be more sophisticated. Therefore, we define the quantile version of regret-DEC similarly, as follows.

Throughout the remainder of this section, we fix the integer $T$. Define

$$\Pi_T = \left\{\widehat{\pi} : \widehat{\pi} = \frac{1}{T}\sum_{t=1}^{T} \delta_{\pi_t}, \text{ where } \pi_1, \cdots, \pi_T \in \Pi\right\} \subseteq \Delta(\Pi),$$

i.e., $\Pi_T$ is the class of all $T$-round mixture decision. We introduce the mixture decision space $\Pi_T$ here to handle the average of $T$-round profile $(\pi_1, \cdots, \pi_T)$ of the algorithm. In particular, when $\Pi$ is convex, we may regard $\Pi_T = \Pi$.

Next, we define the quantile regret-DEC as

$$\mathsf{r\text{-}dec}^{\mathrm{q}}_{\varepsilon,\delta}(\mathcal{M}, \overline{M}) := \inf_{p \in \Delta(\Pi_T)} \sup_{M \in \mathcal{M}} \left\{ \widehat{L}_\delta(M, p) \vee \mathbb{E}_{\pi \sim p}[L(\overline{M}, \pi)] \,\Big|\, \mathbb{E}_{\pi \sim p} D_{\mathrm{H}}^2\big(M(\pi), \overline{M}(\pi)\big) \leq \varepsilon^2 \right\}, \tag{32}$$

and define $\mathsf{r\text{-}dec}^{\mathrm{q}}_{\varepsilon,\delta}(\mathcal{M}) := \sup_{\overline{M} \in \mathcal{M}^+} \mathsf{r\text{-}dec}^{\mathrm{q}}_{\varepsilon,\delta}(\mathcal{M}, \overline{M})$.

The following proposition relates our quantile regret-DEC to the constrained regret-DEC (proof in Appendix F.6).

**Proposition E.2.** *Suppose that Assumption 4 holds for $\mathcal{M}$. Then, for any $\overline{M} \in \mathcal{M}^+$, it holds that*

$$\text{r-dec}_\varepsilon^c(\mathcal{M} \cup \{\overline{M}\}, \overline{M}) \leq 2 \cdot \text{r-dec}_{\varepsilon,\delta}^q(\mathcal{M}, \overline{M}) + c_\delta C_r \varepsilon,$$

*where we denote $c_\delta = \max\left\{\frac{\delta}{1-\delta}, 1\right\}$. In particular, it holds that*

$$\text{r-dec}_{\varepsilon,1/2}^q(\mathcal{M}) \geq \frac{1}{2}\left(\max_{\overline{M} \in \mathcal{M}^+} \text{r-dec}_\varepsilon^c(\mathcal{M} \cup \{\overline{M}\}, \overline{M}) - C_r \varepsilon\right).$$

**Lower bound with quantile regret-DEC**  Now, we prove the following lower bound for the regret of any $T$-round algorithm, via our general interactive Fano method (Lemma 5). The proof is presented in Appendix F.5.

**Theorem E.3.** *Suppose that Assumption 4 holds for $\mathcal{M}$. Then, for any $T$-round algorithm* ALG, *parameters $\varepsilon, \delta, C > 0$, there exists $M \in \mathcal{M}$ such that*

$$\mathbb{P}^{M,\text{ALG}}\left(\mathbf{Reg}_{\text{DM}}(T) \geq T \cdot (\text{r-dec}_{\varepsilon,\delta}^q(\mathcal{M}) - CC_r\varepsilon) - 1\right) \geq \delta - \frac{1}{C^2} - \sqrt{14T\varepsilon^2}.$$

*As a corollary, there exists $M^\star \in \mathcal{M}$ such that*

$$\mathbf{Reg}_{\text{DM}}(T) \geq \frac{T}{2} \cdot \left(\max_{\overline{M} \in \mathcal{M}^+} \text{r-dec}_{\underline{\varepsilon}(T)}^c(\mathcal{M} \cup \{\overline{M}\}, \overline{M}) - 4C_r\underline{\varepsilon}(T)\right) - 1$$

$$\geq \frac{T}{2} \cdot \left(\text{r-dec}_{\underline{\varepsilon}(T)}^c(\mathcal{M}) - 4C_r\underline{\varepsilon}(T)\right) - 1$$

*with probability at least $0.01$ under $\mathbb{P}^{M^\star,\text{ALG}}$, where $\underline{\varepsilon}(T) = \frac{1}{100\sqrt{T}}$.*

Theorem E.1 is now an immediate corollary, because for reward-maximization setting, we always have $C_r = \sqrt{2}$ in Assumption 4.

## E.2  Results for interactive estimation

More generally, we show that for a fairly different task of interactive estimation (Example 3), we also have an equivalence between quantile PAC-DEC with constrained PAC-DEC.

Recall that in this setting, each model $M \in \mathcal{M}$ is assigned with a parameter $\theta_M \in \Theta$, which is the parameter that the agent want to estimate. The decision space $\Pi = \Pi_0 \times \Theta$, where each decision $\pi \in \Pi$ consists of $\pi = (\pi_0, \theta)$, where $\pi_0$ is the *explorative* decision to interact with the model, and $\theta$ is the estimator of the model parameter. The risk function is then defined as $L(M, \pi) = \rho(\theta_M, \theta)$, for certain distance $\rho(\cdot, \cdot)$.

In interactive estimation, we can show that the quantile DEC is in fact lower bounded the constrained DEC, as follows (proof in Appendix F.7).

**Proposition E.4.** *Consider the setting of Example 3. Then as long as $\delta < \frac{1}{2}$, it holds that*

$$\text{p-dec}_\varepsilon^c(\mathcal{M}) \leq 2 \cdot \text{p-dec}_{\varepsilon,\delta}^q(\mathcal{M}).$$

In particular, for such a setting (which encompasses the model estimation task considered in Chen et al. [23]), Theorem 6 provides a lower bound of estimation error in terms of constrained PAC-DEC. This is significant because the constrained PAC-DEC upper bound in Theorem 4 is actually not restricted to Example 1, and we have hence shown that

$$\text{p-dec}_{\underline{\varepsilon}(T)}^c(\mathcal{M}) \lesssim \inf_{\text{ALG}} \sup_{M^\star \in \mathcal{M}} \mathbb{E}^{M^\star,\text{ALG}}[\mathbf{Risk}_{\text{DM}}(T)] \lesssim \text{p-dec}_{\bar{\varepsilon}(T)}^c(\mathcal{M}),$$

where $\underline{\varepsilon}(T) \asymp \sqrt{1/T}$ and $\bar{\varepsilon}(T) \asymp \sqrt{\log|\mathcal{M}|/T}$. Therefore, for interactive estimation, constrained PAC-DEC is also a *nearly tight* complexity measure.

**Remark E.5.** The $\log|\mathcal{M}|$-gap between the lower and upper bound can further be closed for convex model class, utilizing the upper bounds in Appendix G.3. More specifically, we consider a convex model class $\mathcal{M}$, where $M \mapsto \theta_M$ is a convex function on $\mathcal{M}$. Then, a suitable instantiation of $\text{ExO}^+$ (Algorithm 1) achieves

$$\mathbf{Risk}_{\text{DM}}(T) \lesssim \Delta + \inf_{\gamma > 0}\left(\text{p-dec}_\gamma^o(\mathcal{M}) + \frac{\log N(\Theta, \Delta) + \log(1/\delta)}{T}\right),$$

where $N(\Theta, \Delta)$ is the $\Delta$-covering number of $\Theta$, because we have $\log \mathsf{N}_{\mathrm{frac}}(\mathcal{M}, \Delta) \leq \log N(\Theta, \Delta)$ by considering the prior $q = \mathrm{Unif}(\Theta_0)$ for a minimal $\Delta$-covering of $\Theta$. Similar to Theorem H.5, we can upper bound $\mathsf{p\text{-}dec}_\gamma^{\mathrm{o}}(\mathcal{M})$ by $\mathsf{p\text{-}dec}_\varepsilon^{\mathrm{c}}(\mathcal{M})$. Taking these pieces together, we can show that under the assumption that $\mathsf{p\text{-}dec}_\varepsilon^{\mathrm{c}}(\mathcal{M})$ is of moderate decay, $\mathsf{ExO}^+$ achieves
$$\mathbf{Risk}_{\mathrm{DM}}(T) \lesssim \mathsf{p\text{-}dec}_{\varepsilon(T)}^{\mathrm{c}}(\mathcal{M}),$$
where $\varepsilon(T) \asymp \sqrt{\log N(\Theta, 1/T)/T}$.

In particular, for the (non-interactive) *functional estimation* problem (see e.g. Polyanskiy and Wu [67]), the parameter space $\Theta \subset \mathbb{R}$, and hence by considering covering number, we have $\log |\Theta| = \widetilde{O}(1)$. Therefore, for convex $\mathcal{M}$, under mild assumption that the DEC is of moderate decaying (Assumption 3), the minimax risk is then characterized by (up to logarithmic factors)
$$\inf_{\mathsf{ALG}} \sup_{M^\star \in \mathcal{M}} \mathbb{E}^{M^\star, \mathsf{ALG}}[\mathbf{Risk}_{\mathrm{DM}}(T)] \asymp \mathsf{p\text{-}dec}_{\sqrt{1/T}}^{\mathrm{c}}(\mathcal{M}).$$
This result can be regarded as a generalization of Polyanskiy and Wu [67] to the interactive estimation setting.

# F  Proofs from Section 3.2 and Appendix E

**Additional notations**  For notational simplicity, for any distribution $q \in \Delta(\Pi)$ and reference model $\overline{M}$, we denote the localized model class around $\overline{M}$ as
$$\mathcal{M}_{q,\varepsilon}(\overline{M}) := \left\{ M \in \mathcal{M} : \mathbb{E}_{\pi \sim q} D_{\mathrm{H}}^2\left(M(\pi), \overline{M}(\pi)\right) \leq \varepsilon^2 \right\}.$$

## F.1  Proof of Lemma 5

To apply Theorem 1, we consider the squared Hellinger distance (which we recall is a $f$-divergence corresponding to $f(x) = \frac{1}{2}(\sqrt{x} - 1)^2$). Consider a fixed tuple $(\overline{M}, M, \Delta)$ with $M \in \mathcal{M}$, $\overline{M} \in \mathrm{co}(\mathcal{M})$, and $\Delta \geq 0$ that satisfies
$$\sqrt{p_{\overline{M}, \mathsf{ALG}}(\pi : L(M, \pi) \geq \Delta)} > \sqrt{\delta} + \sqrt{14T\, \mathbb{E}_{\pi \sim q_{\overline{M}, \mathsf{ALG}}} D_{\mathrm{H}}^2\left(M(\pi), \overline{M}(\pi)\right)}. \tag{33}$$
We choose the reference distribution to be $\mathbb{Q} = \mathbb{P}^{\overline{M}, \mathsf{ALG}}$, and take $\mu$ to be the singleton distribution supported on $M$. Recall that for the DMSO framework, the observation is $X = (\mathcal{H}^T, \widehat{\pi})$, and the loss function is $L(M, X) = L(M, \widehat{\pi})$ (Section 2.2). Then, by definition, we have
$$\rho_{\Delta, \mathbb{Q}} = \mathbb{P}_{X \sim \mathbb{Q}}(L(M, X) < \Delta) = p_{\overline{M}, \mathsf{ALG}}(\pi : L(M, \pi) < \Delta).$$
Further, using the sub-additivity of Hellinger distance (11), we have
$$D_{\mathrm{H}}^2\left(\mathbb{P}^{M, \mathsf{ALG}}, \mathbb{P}^{\overline{M}, \mathsf{ALG}}\right) \leq 7T \cdot \mathbb{E}_{\pi \sim q_{\overline{M}, \mathsf{ALG}}} D_{\mathrm{H}}^2\left(M(\pi), \overline{M}(\pi)\right).$$
Therefore, using the condition (33), we have
$$\frac{1}{2}\left(\sqrt{\delta} - \sqrt{1 - \rho_{\Delta, \mathbb{Q}}}\right)^2 > D_{\mathrm{H}}^2\left(\mathbb{P}^{M, \mathsf{ALG}}, \mathbb{P}^{\overline{M}, \mathsf{ALG}}\right).$$
Hence, it holds that $D_{\mathrm{H}}^2\left(\mathbb{P}^{M, \mathsf{ALG}}, \mathbb{Q}\right) < D_{\mathrm{H}}^2\left(1 - \delta, \rho_{\Delta, \mathbb{Q}}\right)$, and applying Theorem 1 gives
$$\mathbb{P}^{M, \mathsf{ALG}}(L(M, \widehat{\pi}) \geq \Delta) \geq \delta.$$
Taking supremum over all pairs $(\overline{M}, M, \Delta)$ satisfying Eq. (33) gives the desired lower bound.  $\square$

## F.2  Proof of Theorem 6

Fix any algorithm $\mathsf{ALG}$ and abbreviate $\underline{\varepsilon} = \underline{\varepsilon}_\delta(T)$. Take an arbitrary parameter $\Delta_0 < \mathsf{p\text{-}dec}_{\underline{\varepsilon}, \delta}^{\mathrm{q}}(\mathcal{M})$. Then there exists $\overline{M}$ such that $\Delta_0 < \mathsf{p\text{-}dec}_{\underline{\varepsilon}, \delta}^{\mathrm{q}}(\mathcal{M}, \overline{M})$. Hence, by the definition (12), we know that
$$\Delta_0 < \sup_{M \in \mathcal{M}} \left\{ \widehat{L}_\delta(M, p_{\overline{M}, \mathsf{ALG}}) \,\Big|\, \mathbb{E}_{\pi \sim q_{\overline{M}, \mathsf{ALG}}} D_{\mathrm{H}}^2\left(M(\pi), \overline{M}(\pi)\right) \leq \underline{\varepsilon}^2 \right\}.$$
Therefore, there exists $M \in \mathcal{M}$ such that
$$\mathbb{E}_{\pi \sim q_{\overline{M}, \mathsf{ALG}}} D_{\mathrm{H}}^2\left(M(\pi), \overline{M}(\pi)\right) \leq \underline{\varepsilon}^2, \qquad \mathbb{P}_{\pi \sim p_{\overline{M}, \mathsf{ALG}}}(L(M, \pi) \geq \Delta_0) \geq \delta.$$
This immediately implies
$$\sqrt{p_{\overline{M}, \mathsf{ALG}}(\pi : L(M, \pi) \geq \Delta)} > \sqrt{\delta_1} + \sqrt{14T\mathbb{E}_{\pi \sim q_{\overline{M}, \mathsf{ALG}}} D_{\mathrm{H}}^2\left(M(\pi), \overline{M}(\pi)\right)},$$
where $\sqrt{\delta_1} = \sqrt{\delta} - \sqrt{14T\underline{\varepsilon}^2}$. Notice that $\delta_1 > \frac{\delta}{2}$ by the choice of $\underline{\varepsilon} = \frac{1}{14}\sqrt{\frac{\delta}{T}}$, and hence applying Lemma 5 shows that there exists $M \in \mathcal{M}$ such that $\mathbb{P}^{M, \mathsf{ALG}}(L(M, \widehat{\pi}) \geq \Delta_0) \geq \frac{\delta}{2}$. Letting $\Delta_0 \to \mathsf{p\text{-}dec}_{\underline{\varepsilon}, \delta}^{\mathrm{q}}(\mathcal{M})$ completes the proof.  $\square$

### F.3  Proof of Proposition 7

In this section, we prove Proposition 7 under the slightly more general setting of Assumption 4.

**Proposition F.1.** *Under Assumption 4, for any reference model $\overline{M} \in \mathcal{M}^+$ and $\varepsilon > 0, \delta \in [0,1)$, it holds that*

$$\mathsf{p\text{-}dec}^{\mathsf{c}}_{\varepsilon/\sqrt{2}}(\mathcal{M}, \overline{M}) \leq \mathsf{p\text{-}dec}^{\mathsf{q}}_{\varepsilon,\delta}(\mathcal{M}, \overline{M}) + \frac{2\varepsilon C_{\mathrm{r}}}{1-\delta}.$$

For Example 1, we always have $C_{\mathrm{r}} \leq \sqrt{2}$, and hence Proposition 7 follows immediately from Proposition F.1.

**Proof of Proposition F.1.**  Fix a reference model $\overline{M}$ and a $\Delta_0 > \mathsf{p\text{-}dec}^{\mathsf{q}}_{\varepsilon,\delta}(\mathcal{M}, \overline{M})$. Then, we pick a pair $(\bar{p}, \bar{q})$ such that

$$\Delta_0 > \sup_{M \in \mathcal{M}} \left\{ \widehat{L}_\delta(M, \bar{p}) \,\middle|\, \mathbb{E}_{\pi \sim \bar{q}} D_{\mathrm{H}}^2 \left( M(\pi), \overline{M}(\pi) \right) \leq \varepsilon^2 \right\},$$

whose existence is guaranteed by the definition of $\mathsf{p\text{-}dec}^{\mathsf{q}}_{\varepsilon,\delta}(\mathcal{M}, \overline{M})$ in (12). In other words, we have

$$\mathbb{P}_{\pi \sim \bar{p}}(L(M, \pi) \leq \Delta_0) \geq 1 - \delta, \qquad \forall M \in \mathcal{M}_{\bar{q}, \varepsilon}(\overline{M})$$

Consider $q = \frac{\bar{p}+\bar{q}}{2}$ and $\varepsilon' = \frac{\varepsilon}{\sqrt{2}}$. Also let

$$\tilde{M} := \arg\max_{M \in \mathcal{M}_{q, \varepsilon'}(\overline{M})} f^M(\pi_M).$$

Now, consider $p \in \Delta(\Pi)$ given by

$$p(\cdot) = \bar{p}\Big(\cdot \,\big|\, L(\tilde{M}, \pi) \leq \Delta_0\Big).$$

By definition, for $\pi \sim p$ we have $f^{\tilde{M}}(\pi) \geq f^{\tilde{M}}(\pi_{\tilde{M}}) - \Delta_0$, and hence

$$\begin{aligned}
\mathbb{E}_{\pi \sim p}[L(M, \pi)] &= f^M(\pi_M) - \mathbb{E}_{\pi \sim p}[f^M(\pi)] \\
&\leq f^M(\pi_M) - \mathbb{E}_{\pi \sim p}\Big[f^{\tilde{M}}(\pi)\Big] + C_{\mathrm{r}} \cdot \mathbb{E}_{\pi \sim p} D_{\mathrm{H}}\Big(M(\pi), \tilde{M}(\pi)\Big) \\
&\leq f^M(\pi_M) - f^{\tilde{M}}(\pi_{\tilde{M}}) + \Delta_0 + C_{\mathrm{r}} \cdot \mathbb{E}_{\pi \sim p} D_{\mathrm{H}}\Big(M(\pi), \tilde{M}(\pi)\Big).
\end{aligned}$$

Notice that for any $M \in \mathcal{M}_{q, \varepsilon'}(\overline{M})$, we have $f^M(\pi_M) \leq f^{\tilde{M}}(\pi_{\tilde{M}})$ and also

$$\begin{aligned}
\mathbb{E}_{\pi \sim p} D_{\mathrm{H}}\Big(M(\pi), \tilde{M}(\pi)\Big) &\leq \frac{1}{\bar{p}\Big(L(\tilde{M}, \pi) \leq \Delta_0\Big)} \mathbb{E}_{\pi \sim \bar{p}} D_{\mathrm{H}}\Big(M(\pi), \tilde{M}(\pi)\Big) \\
&\leq \frac{1}{1-\delta}\Big(\mathbb{E}_{\pi \sim \bar{p}} D_{\mathrm{H}}\big(M(\pi), \overline{M}(\pi)\big) + \mathbb{E}_{\pi \sim \bar{p}} D_{\mathrm{H}}\Big(\tilde{M}(\pi), \overline{M}(\pi)\Big)\Big) \\
&\leq \frac{2\varepsilon}{1-\delta}.
\end{aligned}$$

Combining these inequalities gives

$$\mathsf{p\text{-}dec}^{\mathsf{c}}_{\varepsilon'}(\mathcal{M}, \overline{M}) \leq \sup_{M \in \mathcal{M}} \left\{ \mathbb{E}_{\pi \sim p}[L(M, \pi)] \,\big|\, \mathbb{E}_{\pi \sim q} D_{\mathrm{H}}^2\left(M(\pi), \overline{M}(\pi)\right) \leq \frac{\varepsilon^2}{2} \right\} \leq \Delta_0 + \frac{2\varepsilon C_{\mathrm{r}}}{1-\delta}.$$

Letting $\Delta_0 \to \mathsf{p\text{-}dec}^{\mathsf{q}}_{\varepsilon,\delta}(\mathcal{M}, \overline{M})$ completes the proof. $\qquad\square$

### F.4  Proof of Proposition 8

Recall that we frame the DMSO setting as an instance of ISDM in Section 2.2, where the observation is given by $X = (\mathcal{H}^T, \widehat{\pi})$, and the loss is $L(M, X) = L(M, \widehat{\pi})$. In particular, for any prior $\mu \in \Delta(\mathcal{M})$, we have

$$\sup_X \mu(M : L(M, X) < \Delta) = \sup_{\pi \in \Pi} \mu(M : L(M, \pi) < \Delta),$$

and by definition, $I_{\mu,\text{ALG}}(M; X) \leq I_\mu(T)$. In particular, for any pair $(\Delta, \mu)$ such that

$$\sup_\pi \mu(M : L(M, \pi) \leq \Delta) \leq \frac{1}{4} \exp(-2I_\mu(T)), \tag{34}$$

we have $\frac{I_{\mu,\text{ALG}}(M;X) + \log 2}{\log \sup_x \mu(M:L(M,x) < \Delta)} \geq -\frac{1}{2}$, and hence applying Proposition 2 gives $\sup_{M \in \mathcal{M}} \mathbb{E}^{M,\text{ALG}}[L(M, \widehat{\pi})] \geq \frac{\Delta}{2}$. Taking supremum over all pairs $(\Delta, \mu)$ satisfying (34) gives the desired lower bound. $\qquad\square$

## F.5 Proof of Theorem E.3

Our proof adopts the analysis strategy originally proposed by Glasgow and Rakhlin [44].

Fix a $0 < \Delta < \text{r-dec}^{\text{q}}_{\varepsilon,\delta}(\mathcal{M})$ and a parameter $c \in (0, 1)$. Then there exists $\overline{M} \in \mathcal{M}^+$ such that $\text{r-dec}^{\text{q}}_{\varepsilon,\delta}(\mathcal{M}, \overline{M}) > \Delta$.

Fix a $T$-round algorithm ALG with rules $p_1, \cdots, p_T$, we consider a modified algorithm ALG' : for $t = 1, \cdots, T$, and history $\mathcal{H}^{(t-1)}$, we set $p'_t(\cdot | \mathcal{H}^{(t-1)}) = p_t(\cdot | \mathcal{H}^{(t-1)})$ if $\sum_{s=1}^{t-1} L(\overline{M}, \pi^s) < T\Delta - 1$, and set $p'_t(\cdot | \mathcal{H}^{(t-1)}) = 1_{\pi_{\overline{M}}}$ if otherwise. By our construction, it holds that under ALG', we have $\sum_{t=1}^T L(\overline{M}, \pi^t) < T\Delta$ almost surely. Furthermore, we can define the stopping time

$$\tau = \inf \left\{ t : \sum_{s=1}^t L(\overline{M}, \pi^s) \geq T\Delta - 1 \text{ or } t = T+1 \right\}.$$

If $\tau \leq T$, then it holds that $\sum_{t=1}^\tau L(\overline{M}, \pi^t) \geq T\Delta - 1$.

Now, we consider $p = \mathbb{P}^{\overline{M},\text{ALG}'}(\frac{1}{T} \sum_{t=1}^T \pi^t = \cdot) \in \Delta(\Pi_T)$. Using our definition of r-dec$^{\text{q}}$, we know that $\mathbb{E}_{\pi \sim p} L(\overline{M}, \pi) < \Delta$ by our construction, and hence there exists $M \in \mathcal{M}$ such that

$$\mathbb{P}_{\overline{\pi} \sim p}(L(M, \overline{\pi}) \geq \Delta) > \delta, \qquad \mathbb{E}_{\overline{\pi} \sim p} D^2_{\text{H}}\left(M(\overline{\pi}), \overline{M}(\overline{\pi})\right) \leq \varepsilon^2.$$

By definition of $p$ and Lemma C.1, we have

$$\mathbb{P}^{\overline{M},\text{ALG}'}\left(\sum_{t=1}^T L(M, \pi^t) \geq T\Delta\right) > \delta, \qquad D^2_{\text{H}}\left(\mathbb{P}^{M,\text{ALG}'}, \mathbb{P}^{\overline{M},\text{ALG}'}\right) \leq 7T\varepsilon^2. \tag{35}$$

We also know

$$\mathbb{E}^{\overline{M},\text{ALG}'}\left[\frac{1}{T} \sum_{t=1}^T |f^M(\pi^t) - f^{\overline{M}}(\pi^t)|^2\right] \leq \mathbb{E}^{\overline{M},\text{ALG}'}\left[\frac{1}{T} \sum_{t=1}^T C^2_{\text{r}} D^2_{\text{H}}\left(M(\pi^t), \overline{M}(\pi^t)\right)\right]$$
$$= C^2_{\text{r}} \mathbb{E}_{\overline{\pi} \sim p} D^2_{\text{H}}\left(M(\overline{\pi}), \overline{M}(\overline{\pi})\right) \leq C^2_{\text{r}} \varepsilon^2,$$

and hence by Markov inequality,

$$\mathbb{P}^{\overline{M},\text{ALG}'}\left(\frac{1}{T} \sum_{t=1}^T |f^M(\pi^t) - f^{\overline{M}}(\pi^t)| \geq CC_{\text{r}}\varepsilon\right) \leq \frac{1}{C^2}.$$

In the following, we consider events

$$\mathcal{E}_1 := \left\{\sum_{t=1}^T L(M, \pi^t) \geq T\Delta\right\},$$

and the random variable $X := \sum_{t=1}^T |f^M(\pi^t) - f^{\overline{M}}(\pi^t)|$. By definition, $\mathbb{P}^{\overline{M},\text{ALG}'}(\mathcal{E}_1) > \delta$, $\mathbb{P}^{\overline{M},\text{ALG}'}(X \geq CTC_{\text{r}}\varepsilon) \leq \frac{1}{C^2}$. We have the following claim.

**Claim:** Under the event $\mathcal{E}_1 \cap \{\tau \leq T\}$, we have

$$\sum_{t=1}^\tau L(M, \pi^t) \geq T\Delta - X - 1.$$

To prove the claim, we bound

$$\sum_{t=1}^{\tau} L(M, \pi^t) = \sum_{t=1}^{T} L(M, \pi^t) - \sum_{t=\tau+1}^{T} L(M, \pi^t)$$

$$\geq T\Delta - \sum_{t=\tau+1}^{T} [f^M(\pi_M) - f^M(\pi^t)]$$

$$\geq T\Delta - (T-\tau)f^M(\pi_M) + \sum_{t=\tau+1}^{T} f^{\overline{M}}(\pi^t) - X$$

$$= T\Delta - (T-\tau) \cdot \left( f^M(\pi_M) - f^{\overline{M}}(\pi_{\overline{M}}) \right) - X,$$

where the first inequality follows from $\mathcal{E}_1$, and the second inequality follows from $\sum_{t=\tau+1}^{T} |f^M(\pi^t) - f^{\overline{M}}(\pi^t)| \leq X$. On the other hand, we can also bound

$$\sum_{t=1}^{\tau} L(M, \pi^t) = \sum_{t=1}^{\tau} [f^M(\pi_M) - f^M(\pi^t)]$$

$$\geq \tau f^M(\pi_M) - \sum_{t=1}^{\tau} f^{\overline{M}}(\pi^t) - X$$

$$= \tau \cdot \left( f^M(\pi_M) - f^{\overline{M}}(\pi_{\overline{M}}) \right) + \sum_{t=1}^{\tau} L(\overline{M}, \pi^t) - X$$

$$\geq \tau \cdot \left( f^M(\pi_M) - f^{\overline{M}}(\pi_{\overline{M}}) \right) + T\Delta - 1 - X,$$

where the first inequality follows from $\sum_{t=1}^{\tau} |f^M(\pi^t) - f^{\overline{M}}(\pi^t)| \leq X$, and the second inequality is because $\sum_{t=1}^{\tau} L(\overline{M}, \pi^t) \geq T\Delta - 1$ given $\tau \leq T$, which follows from the definition of the stopping time $\tau$. Therefore, taking maximum over the above two inequalities proves our claim.

Now, using the claim, we know

$$\mathbb{P}^{\overline{M}, \text{ALG}'} \left( \sum_{t=1}^{\tau \wedge T} L(M, \pi^t) \geq T(\Delta - C\varepsilon) - 1 \right) \geq \mathbb{P}^{\overline{M}, \text{ALG}'}(\mathcal{E}_1 \cap \{X \leq CT\varepsilon\}) \geq \delta - \frac{1}{C^2}.$$

Notice that $D_{\text{H}}^2 \left( \mathbb{P}^{M, \text{ALG}'}, \mathbb{P}^{\overline{M}, \text{ALG}'} \right) \leq 7T\varepsilon^2$, and hence for any event $\mathcal{E}$, it holds $\mathbb{P}^{M, \text{ALG}'}(\mathcal{E}) \geq \mathbb{P}^{\overline{M}, \text{ALG}'}(\mathcal{E}) - \sqrt{14T\varepsilon^2}$. In particular, we have

$$\mathbb{P}^{M, \text{ALG}'} \left( \sum_{t=1}^{\tau \wedge T} L(M, \pi^t) \geq T(\Delta - CC_{\text{r}}\varepsilon) - 1 \right) \geq \delta - \frac{1}{C^2} - \sqrt{14T\varepsilon^2}.$$

Finally, we note that ALG and ALG$'$ agree on the first $\tau \wedge T$ rounds (formally, ALG and ALG$'$ induce the same distribution of $(\pi^1, \cdots, \pi^{\tau \wedge T})$), and hence

$$\mathbb{P}^{M, \text{ALG}} \left( \sum_{t=1}^{\tau \wedge T} L(M, \pi^t) \geq T(\Delta - CC_{\text{r}}\varepsilon) - 1 \right) \geq \delta - \frac{1}{C^2} - \sqrt{14T\varepsilon^2}.$$

The proof is hence complete by noticing that $\sum_{t=1}^{\tau \wedge T} L(M, \pi^t) \leq \sum_{t=1}^{T} L(M, \pi^t) = \mathbf{Reg}_{\text{DM}}(T)$ and taking $\Delta \to \text{r-dec}_{\varepsilon, \delta}^{\text{q}}(\mathcal{M})$.

### F.6 Proof of Proposition E.2

Fix a $\overline{M} \in \mathcal{M}^+$, and $\Delta > \text{r-dec}_{\varepsilon, \delta}^{\text{q}}(\mathcal{M}, \overline{M})$. Choose $p \in \Delta(\Pi_T)$ such that

$$\widehat{L}_\delta(M, p) \vee \mathbb{E}_{\pi \sim p}[L(\overline{M}, \pi)] \leq \Delta, \qquad \forall M \in \mathcal{M}_{p, \varepsilon}(\overline{M}).$$

The existence of $p$ is guaranteed by the definition (32). In other words, we have $\mathbb{E}_{\pi \sim p}[L(\overline{M}, \pi)] \leq \Delta$ and

$$\mathbb{P}_{\pi \sim p}(L(M, \pi) \geq \Delta) \leq \delta, \qquad \forall M \in \mathcal{M}_{p, \varepsilon}(\overline{M}).$$

We then has the following claim.

**Claim.** For any model $M \in \mathcal{M}$, it holds that

$$\mathbb{E}_{\pi \sim p}[L(M, \pi)] \leq \mathbb{E}_{\pi \sim p}[L(\overline{M}, \pi)] + \Delta + c_\delta C_\mathrm{r} \mathbb{E}_{\pi \sim p} D_\mathrm{H}\big(M(\pi), \overline{M}(\pi)\big). \tag{36}$$

Fix any $M \in \mathcal{M}$, we prove (36) as follows. Consider the event $\mathcal{E} = \{\pi : L(M, \pi) \leq \Delta\}$. Then,

$$p(\mathcal{E})\big(f^M(\pi_M) - f^{\overline{M}}(\pi_{\overline{M}})\big) = \mathbb{E}_{\pi \sim p}\mathbf{1}\left\{\mathcal{E}\right\}\big(L(M, \pi) - L(\overline{M}, \pi) + f^{\overline{M}}(\pi) - f^M(\pi)\big)$$
$$\leq p(\mathcal{E})\Delta + C_\mathrm{r}\mathbb{E}_{\pi \sim p}\mathbf{1}\left\{\mathcal{E}\right\}D_\mathrm{H}\big(M(\pi), \overline{M}(\pi)\big),$$

where the inequality uses $L(M, \pi) \leq \Delta$ for $\pi \in \mathcal{E}$ and Assumption 4. Therefore,

$$\mathbb{E}_{\pi \sim p}L(M, \pi) = \mathbb{E}_{\pi \sim p}\mathbf{1}\left\{\mathcal{E}\right\}L(M, \pi) + \mathbb{E}_{\pi \sim p}\mathbf{1}\left\{\mathcal{E}^c\right\}L(M, \pi)$$
$$\leq p(\mathcal{E})\Delta + \mathbb{E}_{\pi \sim p}\mathbf{1}\left\{\mathcal{E}^c\right\}\big(f^M(\pi_M) - f^{\overline{M}}(\pi_{\overline{M}}) + f^{\overline{M}}(\pi) - f^M(\pi) + L(\overline{M}, \pi)\big)$$
$$\leq 2\Delta + \frac{p(\mathcal{E}^c)C_\mathrm{r}}{p(\mathcal{E})}\mathbb{E}_{\pi \sim p}\mathbf{1}\left\{\mathcal{E}\right\}D_\mathrm{H}\big(M(\pi), \overline{M}(\pi)\big) + C_\mathrm{r}\mathbb{E}_{\pi \sim p}\mathbf{1}\left\{\mathcal{E}^c\right\}D_\mathrm{H}\big(M(\pi), \overline{M}(\pi)\big)$$
$$\leq 2\Delta + \max\left\{\frac{p(\mathcal{E}^c)}{p(\mathcal{E})}, 1\right\}C_\mathrm{r}\mathbb{E}_{\pi \sim p}D_\mathrm{H}\big(M(\pi), \overline{M}(\pi)\big).$$

This completes the proof of our claim.

Therefore, using (36) with $\mathbb{E}_{\pi \sim p}[L(\overline{M}, \pi)] \leq \Delta$ yields

$$\mathbb{E}_{\pi \sim p}[L(M, \pi)] \leq 2\Delta + c_\delta C_\mathrm{r}\varepsilon, \qquad \forall M \in \mathcal{M}_{p, \varepsilon}(\overline{M}).$$

This immediately implies

$$\mathsf{r\text{-}dec}^\mathsf{c}_\varepsilon(\mathcal{M} \cup \{\overline{M}\}, \overline{M}) \leq 2\Delta + c_\delta C_\mathrm{r}\varepsilon.$$

Finally, taking $\Delta \to \mathsf{r\text{-}dec}^\mathsf{q}_{\varepsilon, \delta}(\mathcal{M}, \overline{M})$ completes the proof. $\qquad\square$

## F.7 Proof of Proposition E.4

Fix a reference model $\overline{M}$ and let $\Delta_0 > \mathsf{p\text{-}dec}^\mathsf{q}_{\varepsilon, \delta}(\mathcal{M}, \overline{M})$. Then there exists $p, q \in \Delta(\Pi)$ such that

$$\sup_{M \in \mathcal{M}}\left\{\widehat{L}_\delta(M, p) \,\Big|\, \mathbb{E}_{\pi \sim q}D_\mathrm{H}^2\big(M(\pi), \overline{M}(\pi)\big) \leq \varepsilon^2\right\} < \Delta_0.$$

Therefore, it holds that

$$\mathbb{P}_{\pi \sim p}(L(M, \pi) \leq \Delta_0) \geq 1 - \delta, \qquad \forall M \in \mathcal{M}_{q, \varepsilon}(\overline{M}).$$

If the constrained set $\mathcal{M}_{q, \varepsilon}(\overline{M})$ is empty, then we immediately have $\mathsf{p\text{-}dec}^\mathsf{c}_\varepsilon(\mathcal{M}, \overline{M}) = 0$, and the proof is completed. Therefore, in the following we may assume $\mathcal{M}_{q, \varepsilon}(\overline{M})$ is non-empty, and $\widehat{M} \in \mathcal{M}_{q, \varepsilon}(\overline{M})$.

**Claim.** Let $\widehat{\theta} = \theta_{\widehat{M}}$ and $\widehat{\pi} = (\pi_0, \widehat{\theta})$ for an arbitrary $\pi_0$, it holds that

$$L(M, \widehat{\pi}) \leq \Delta_0, \qquad \forall M \in \mathcal{M}_{q, \varepsilon}(\overline{M}).$$

This is because for any $M \in \mathcal{M}_{q, \varepsilon}(\overline{M})$, it holds that

$$\mathbb{P}_{\pi \sim p}(L(M, \pi) \leq \Delta_0, L(\widehat{M}, \pi) \leq \Delta_0) \geq 1 - 2\delta > 0.$$

Hence, there exists $\theta \in \Theta$ such that $\rho(\theta_M, \theta) \leq \Delta_0$ and $\rho(\theta_{\widehat{M}}, \theta) \leq \Delta_0$ holds. Therefore, it must hold that $\rho(\theta_M, \widehat{\theta}) \leq 2\Delta_0$ for any $M \in \mathcal{M}_{q, \varepsilon}(\overline{M})$.

The above claim immediately implies that

$$\mathsf{p\text{-}dec}^\mathsf{c}_\varepsilon(\mathcal{M}, \overline{M}) \leq \sup_{M \in \mathcal{M}}\left\{L(M, \widehat{\pi}) \,\big|\, \mathbb{E}_{\pi \sim q}D_\mathrm{H}^2\big(M(\pi), \overline{M}(\pi)\big) \leq \varepsilon^2\right\} \leq 2\Delta_0.$$

Letting $\Delta_0 \to \mathsf{p\text{-}dec}^\mathsf{q}_{\varepsilon, \delta}(\mathcal{M}, \overline{M})$ yields $\mathsf{p\text{-}dec}^\mathsf{c}_\varepsilon(\mathcal{M}, \overline{M}) \leq 2\mathsf{p\text{-}dec}^\mathsf{q}_{\varepsilon, \delta}(\mathcal{M}, \overline{M})$, which is the desired result. $\qquad\square$

# G  Additional Discussion and Results from Section 4

## G.1  Properties of the fractional covering number

Before proceeding to applications, let us briefly discuss some connections between the fractional covering number and classical notions of covering number considered in the context of statistical estimation.

To start, we recall that for many standard statistical estimation tasks such as regression and non-parametric estimation, the risk function $L$ is given by a (pseudo) metric (e.g., $\ell_2$ distance between parameters or mean-squared error in predictions). The following lemma shows that in this case, the fractional covering number coincides with the classical covering number induced by the metric, e.g., in location estimation (Example 4), density estimation (Example 6), etc.

**Lemma G.1** (Connection to classical covering numbers). *Suppose the decision space $\Pi$ is equipped with a pseudo-metric $\rho : \Pi \times \Pi \to \mathbb{R}_+$ and there is a map $M \mapsto \pi_M \in \Pi$ such that the risk function is given by*

$$L(M, \pi) = \rho(\pi_M, \pi), \qquad \forall M \in \mathcal{M}, \pi \in \Pi. \tag{37}$$

*Let $\Pi_{\mathcal{M}} := \{\pi_M : M \in \mathcal{M}\} \subseteq \Pi$, and define $\mathsf{N}(\Pi_{\mathcal{M}}, \Delta)$ to be the $\Delta$-covering number of $\Pi_{\mathcal{M}}$ under $\rho$. Then*

$$\mathsf{N}(\Pi_{\mathcal{M}}, 2\Delta) \leq \mathsf{N}_{\mathrm{frac}}(\mathcal{M}, \Delta) \leq \mathsf{N}(\Pi_{\mathcal{M}}, \Delta).$$

**Duality between fractional covering and fractional packing**  For classical covering numbers with respect to a pseudo-metric (as in Lemma G.1), it is known that there is a duality between covering and packing. In spite of a lack of metric structure for the general setting we study, we can show that fractional covering number naturally admits a dual representation in terms of a *fractional packing number*. Specifically, it holds that

$$\inf_{\mu \in \Delta(\mathcal{M})} \sup_{\pi \in \Pi} \mu(M : L(M, \pi) \leq \Delta) = \sup_{p \in \Delta(\Pi)} \inf_{M \in \mathcal{M}} p(\pi : L(M, \pi) \leq \Delta),$$

as long as the minimax theorem can be applied (e.g. when $\Pi$ or $\mathcal{M}$ are finite or satisfy appropriate compactness conditions). Therefore, in this case, we have

$$\mathsf{N}_{\mathrm{frac}}(\mathcal{M}, \Delta) = \sup_{\mu \in \Delta(\mathcal{M})} \inf_{\pi \in \Pi} \frac{1}{\mu(M : L(M, \pi) \leq \Delta)} \tag{38}$$

which can be interpreted as a fractional packing number. We mention in passing that using this interpretation, it is possible to derive Theorem 10 directly from Proposition 8.

**Recovering the Yang-Barron method.**  As a simple example of the fractional covering number, we recover and further generalize the well-known Yang-Barron method [95] for statistical estimation problems (see also [89, Section 15.3.5]).

**Example 12** (Yang-Barron method).  For a statistical estimation problem with model class $\mathcal{M}$, we define the KL covering number of $\mathcal{M}$ as

$$\mathsf{N}_{\mathrm{KL}}(\mathcal{M}, \varepsilon) := \min \left\{ k : \exists M^1, \cdots, M^k \in \mathcal{M}, \text{such that } \forall M \in \mathcal{M}, \min_{i \in [k]} D_{\mathrm{KL}}(M \parallel M^i) \leq \varepsilon^2 \right\}.$$

For a fixed parameter $\varepsilon > 0$, we can pick $k = \mathsf{N}_{\mathrm{KL}}(\mathcal{M}, \varepsilon)$ and $M^1, \cdots, M^k \in \mathcal{M}$ such that $\min_{i \in [k]} D_{\mathrm{KL}}(M \parallel M^i) \leq \varepsilon^2$ for all $M \in \mathcal{M}$. Then, let us consider the localized sub-class $\mathcal{M}_i := \{M \in \mathcal{M} : D_{\mathrm{KL}}(M \parallel M^i) \leq \varepsilon^2\}$ for each $i \in [k]$. It is clear that Assumption 2 holds for each $\mathcal{M}_i$ with $C_{\mathrm{KL}} \leq \varepsilon^2$. Further, using $\mathcal{M} = \bigcup_{i=1}^n \mathcal{M}_i$, we have

$$\log \mathsf{N}_{\mathrm{frac}}(\mathcal{M}, \Delta) \leq \max_{i \in [k]} \log \mathsf{N}_{\mathrm{frac}}(\mathcal{M}_i, \Delta) + \log k. \tag{39}$$

For details, see Appendix H.4. Therefore, applying Theorem 10 to $\mathcal{M}_i$ and take supremum over $i \in [k]$ and $\varepsilon > 0$ gives the following result: *For any algorithm* ALG *to achieve* $\sup_{\theta \in \Theta} \mathbb{E}^{\theta, \text{ALG}} L(M, \pi) \leq \Delta$, *it is necessary that*

$$\log \mathsf{N}_{\mathrm{frac}}(\mathcal{M}, \Delta) \leq \inf_{\varepsilon > 0} \left( 2T\varepsilon^2 + \log \mathsf{N}_{\mathrm{KL}}(\mathcal{M}, \varepsilon) \right) + 2.$$

When the risk function $L$ is given by a metric (cf. Eq. (37)), this inequality coincides with the Yang-Barron method formulated in Wainwright [89, Section 15.3.5], as the fractional covering number $\mathsf{N}_{\mathrm{frac}}(\mathcal{M}, \Delta)$ can be lower bounded by the covering number (Lemma G.1).

**Recovering the local packing lower bound for statistical estimation.** As a simple example of the fractional covering number, we recover the well-known local packing-based lower bound [15] for the classical problem of location estimation [89].

**Example 13** (Local packing lower bound for location estimation)**.** In the location estimation task (Example 4), recall that the model class is given by $\mathcal{M} = \{M_\theta : \theta \in \Theta\}$, where $M_\theta = \mathcal{N}(\theta, I_d)$. Consider the local packing number of $\Theta$ around $\theta^\star \in \Theta$, which is given by

$$\mathsf{N}_{\mathrm{loc}}(\Theta, \Delta; \theta^\star) := \max \left\{ k : \exists \theta^1, \cdots, \theta^k \in \Theta, \left\| \theta^i - \theta^\star \right\| \le \Delta, \left\| \theta^i - \theta^j \right\| > \frac{\Delta}{2}, \forall i \ne j \right\}.$$

Then, for the localized sub-class $\mathcal{M}_{\varepsilon, \theta^\star} := \{M_\theta : \|\theta - \theta^\star\| \le \varepsilon\} \subseteq \mathcal{M}$, Assumption 2 holds with $C_{\mathrm{KL}} \le \frac{1}{2}\varepsilon^2$, and we also have $\mathsf{N}_{\mathrm{frac}}(\mathcal{M}_{\varepsilon, \theta^\star}, \varepsilon/4) \ge \mathsf{N}_{\mathrm{loc}}(\Theta, \varepsilon; \theta^\star)$. Therefore, we can apply Theorem 10 to $\mathcal{M}_{\varepsilon, \theta^\star}$ and take supremum over all $\theta^\star \in \Theta$ and $\varepsilon \ge 8\Delta$ to show the following result: *For any algorithm* ALG *to achieve* $\sup_{\theta \in \Theta} \mathbb{E}^{\theta, \text{ALG}} \|\widehat{\pi} - \theta\| \le \Delta$, *it is necessary that*

$$T \ge \sup_{\varepsilon \ge 8\Delta} \frac{\log \mathsf{N}_{\mathrm{loc}}(\Theta, \varepsilon) - 2}{\varepsilon^2},$$

where $\mathsf{N}_{\mathrm{loc}}(\Theta, \varepsilon) = \sup_{\theta^\star \in \Theta} \mathsf{N}_{\mathrm{loc}}(\Theta, \varepsilon; \theta^\star)$ is the local packing number of $\Theta$. This lower bound is known to be tight in general [14, 15, 59, etc.]. ◁

## G.2 Additional discussion from Section 4.2

In the literature on statistical learning, there is a long line of work which characterizes learnability of hypothesis classes in terms of abstract complexity measures. Examples include the Vapnik-Chervonenkis dimension for binary classification [83, 16], the Littlestone dimension [63] for online classification [10] and differentially private classification [20, 6], and their real-valued counterparts (e.g. scale-sensitive dimensions) for regression [9, 5].

Beyond the settings above—particularly for interactive settings—learnability is less well understood. The question of what complexity measure characterizes bandit learnability has been explored in Russo and Van Roy [74], Abernethy et al. [1], Simchowitz et al. [79], Hashimoto et al. [46, etc.], but a complete resolution has yet to be reached. Remarkably, Ben-David et al. [11] demonstrate that there exists a simple and natural learning task for which it is impossible to characterize learnability through any *combinatorial* dimension. More recently, Hanneke and Yang [45] provide similar impossibility results for characterizing the learnability of noiseless structured noiseless bandits with real-valued rewards.

Our characterization bypasses the impossibility results of Hanneke and Yang [45]. Specifically, Hanneke and Yang [45] show that for *noiseless* structured bandit problems, there exist classes $\mathcal{H}$ for which bandit learnability is independent of the axioms of ZFC. Therefore, their results rule out the possibility of a characterization of noiseless bandit learnability through any *combinatorial dimension* [11] for the problem class. Our characterization is compatible with this result because the argument of Hanneke and Yang [45] relies on the noiseless nature of the bandit problem, and hence does not preclude a characterization for the noisy setting.

**Connection to the maximin volume.** Hanneke and Yang [45] propose *maximin volume*, a complexity measure that tightly characterizes the complexity of learning *noiseless binary-valued* structured bandit problems. For such problem classes, the fractional covering number is exactly the inverse of the maximin volume. While the fractional covering number can be viewed as a generalization of the maximin volume in this sense, we emphasize that the fractional covering number directly arises from our general lower bound framework, and is applicable to general decision making problems in the DMSO framework.

**Noise distribution.** We note that the upper bound in (20) applies to any reward distribution with sub-Gaussian noise (cf. Appendix H.2). Meanwhile, since the lower bound in Corollary 12 is specialized to Gaussian noise, it acts as a lower bound for the broader class of sub-Gaussian noise distributions as well. We expect the lower bound to extend to other "reasonable" noise distributions.

## G.3 Exploration-by-Optimization Algorithm

In this section, we present a slightly modified version of the Exploration-by-Optimization Algorithm (ExO$^+$) developed by Foster et al. [41], built upon Lattimore and Szepesvári [58], Lattimore and

---

**Algorithm 1** Exploration-by-Optimization (ExO$^+$)

---

**Input:** Problem $(\mathcal{M}, \Pi)$, prior $q \in \Delta(\Pi)$, parameter $T \geq 1$, $\gamma > 0$.
1: Set $q^1 = q$.
2: **for** $t = 1, \cdots, T$ **do**
3:   Solve the *exploration-by-optimization* objective

$$(p^t, \ell^t) \leftarrow \underset{p \in \Delta(\Pi), \ell \in \mathcal{L}}{\arg\min} \Gamma_{q^t, \gamma}(p, \ell)$$

4:   Sample $\pi^t \sim p^t$, execute $\pi^t$ and observe $o^t$
5:   Update

$$q^{t+1}(\pi) \propto_\pi q^t(\pi) \exp(\ell^t(\pi; \pi^t, o^t))$$

---

Gyorgy [55]. The original ExO$^+$ algorithm has an *adversarial* regret guarantee for any model class $\mathcal{M}$, scaling with r-dec$_\gamma^o(\text{co}(\mathcal{M}))$, the offset DEC of the mode class $\text{co}(\mathcal{M})$, and $\log |\Pi|$, the log-cardinality of the decision space. For our purpose, we adapt the original ExO$^+$ algorithm by using a prior $q \in \Delta(\Pi)$ not necessarily the uniform prior, and with a suitably chosen prior $q$, ExO$^+$ then achieves a regret guarantee scaling with $\log \mathsf{N}_{\text{frac}}(\mathcal{M}, \Delta)$, instead of $\log |\Pi|$ (cf. Foster et al. [41]), which is always an upper bound of $\log \mathsf{N}_{\text{frac}}(\mathcal{M}, \Delta)$.

**Offset DEC for regret.**  We first recall the following (original) definition of DEC [40]:

$$\mathsf{r\text{-}dec}_\gamma^o(\mathcal{M}, \overline{M}) := \inf_{p \in \Delta(\Pi)} \sup_{M \in \mathcal{M}} \mathbb{E}_{\pi \sim p}[L(M, \pi)] - \gamma \mathbb{E}_{\pi \sim p} D_{\text{H}}^2\left(M(\pi), \overline{M}(\pi)\right), \quad (40)$$

and $\mathsf{r\text{-}dec}_\gamma^o(\mathcal{M}) := \sup_{\overline{M} \in \text{co}(\mathcal{M})} \mathsf{r\text{-}dec}_\gamma^o(\mathcal{M}, \overline{M})$.  Through the Estimation-to-Decision (E2D) algorithm [40], offset regret-DEC provides an upper bound of $\mathbf{Reg}_{\text{DM}}$ for any learning problem, and it is also closely related to the complexity of adversarial decision making.

As discussed in Foster et al. [42], in the reward maximization setting (Example 1), the constrained regret-DEC r-dec$^c$ can always be upper bounded in terms of the offset DEC r-dec$^o$. Conversely, in the same setting, we also show that the offset DEC can also be upper bounded in terms of the constrained DEC (Theorem H.5), and hence the two concepts can be regarded as equivalent under mild assumptions (e.g. moderate decaying, Assumption 3).

**Exploration-by-Optimization algorithm.**  The algorithm, ExO$^+$, is restated in Algorithm 1. At each round $t$, the algorithm maintains a reference distribution $q^t \in \Delta(\Pi)$, and use it to obtain a decision distribution $p^t \in \Delta(\Pi)$ and an estimation function $\ell^t \in \mathcal{L} := (\Pi \times \Pi \times \mathcal{O} \to \mathbb{R})$, by solving a joint minimax optimization problem based on the *exploration-by-optimization* objective: Defining

$$\begin{aligned}
\Gamma_{q, \gamma}(p, \ell; M, \pi^\star) = \mathbb{E}_{\pi \sim p}[f^M(\pi^\star) - f^M(\pi)] \\
- \gamma \mathbb{E}_{\pi \sim p} \mathbb{E}_{o \sim M(\pi)} \mathbb{E}_{\pi' \sim q}[1 - \exp\left(\ell(\pi'; \pi, o) - \ell(\pi^\star; \pi, o)\right)],
\end{aligned} \quad (41)$$

and

$$\Gamma_{q, \gamma}(p, \ell) = \sup_{M \in \mathcal{M}, \pi^\star \in \Pi} \Gamma_{q, \gamma}(p, \ell; M, \pi^\star), \quad (42)$$

the algorithm solve $(p^t, \ell^t) \leftarrow \arg\min_{p \in \Delta(\Pi), \ell \in \mathcal{L}} \Gamma_{q^t, \gamma}(p, \ell)$. The algorithm then samples $\pi^t \sim p^t$, executes $\pi^t$ and observes $o^t$ from the environment. Finally, the algorithm updates the reference distribution by performing the exponential weight update with weight function $\ell^t(\cdot; \pi^t, o^t)$.

**Guarantee of ExO$^+$.**  Following Foster et al. [41], we define

$$\mathsf{exo}_{1/\gamma}(\mathcal{M}, q) := \inf_{p \in \Delta(\Pi), \ell \in \mathcal{L}} \Gamma_{q, \gamma}(p, \ell), \quad (43)$$

and $\mathsf{exo}_{1/\gamma}(\mathcal{M}) = \sup_{q \in \Delta(\Pi)} \mathsf{exo}_{1/\gamma}(\mathcal{M}, q)$. The following theorem is deduced from Foster et al. [41, Theorem 3.1 and 3.2].

**Theorem G.2.** *Under the reward maximization setting*[4]*(Assumption 4), it holds that*

$$\mathsf{r\text{-}dec}^{\mathsf{o}}_{\gamma/4}(\mathrm{co}(\mathcal{M})) \le \mathsf{exo}_{1/\gamma}(\mathcal{M}) \le \mathsf{r\text{-}dec}^{\mathsf{o}}_{\gamma/8}(\mathrm{co}(\mathcal{M})), \qquad \forall \gamma > 0.$$

Now, we present the main guarantee of Algorithm 1, which has the desired dependence on the prior $q \in \Delta(\Pi)$.

**Theorem G.3.** *It holds that with probability at least* $1 - \delta$,

$$\mathbf{Reg}_{\mathsf{DM}} \le T\Big(\Delta + \mathsf{r\text{-}dec}^{\mathsf{o}}_{\gamma/8}(\mathrm{co}(\mathcal{M}))\Big) + \gamma \log\left(\frac{1}{\delta \cdot q(\pi : f^{M^\star}(\pi_{M^\star}) - f^{M^\star}(\pi) \le \Delta)}\right)$$

*Proof.* Consider the set $\Pi^\star := \{\pi : f^{M^\star}(\pi_{M^\star}) - f^{M^\star}(\pi) \le \Delta\}$ and the distribution $q^\star = q(\cdot|\Pi^\star)$. Following Proposition G.4, we consider

$$X_t(\pi^t, o^t) := \mathbb{E}_{\pi \sim q^\star}\big[\ell^t(\pi; \pi^t, o^t)\big] - \log \mathbb{E}_{\pi \sim q^t}\big[\exp\big(\ell^t(\pi; \pi^t, o^t)\big)\big],$$

and Proposition G.4 implies that

$$\sum_{t=1}^{T} X_t(\pi^t, o^t) \le \log(1/q(\Pi^\star)).$$

Applying Lemma C.3, we have with probability at least $1 - \delta$,

$$\sum_{t=1}^{T} - \log \mathbb{E}_{t-1}\big[\exp\big(-X_t(\pi^t, o^t)\big)\big] \le \sum_{t=1}^{T} X_t(\pi^t, o^t) + \log(1/\delta).$$

Notice that

$$\mathbb{E}_{t-1}\big[\exp\big(-X_t(\pi^t, o^t)\big)\big] = \mathbb{E}_{\pi \sim p^t}\mathbb{E}_{o \sim M^\star(\pi)}\mathbb{E}_{\pi' \sim q^t}\big[\exp\big(\ell^t(\pi'; \pi, o) - \mathbb{E}_{\pi^\star \sim q^\star}\ell^t(\pi^\star; \pi, o)\big)\big].$$

Using the fact that $1 - x \le -\log x$ and Jensen's inequality, we have

$$\sum_{t=1}^{T} \mathbb{E}_{\pi^\star \sim q^\star} \mathrm{Err}(p^t, \ell^t; q^t, M^\star, \pi^\star) \le \log(1/q(\Pi^\star)) + \log(1/\delta),$$

where we denote

$$\mathrm{Err}(p, \ell; q, M^\star, \pi^\star) := \mathbb{E}_{\pi \sim p}\mathbb{E}_{o \sim M^\star(\pi)}\mathbb{E}_{\pi' \sim q}[1 - \exp\big(\ell(\pi'; \pi, o) - \ell(\pi^\star; \pi, o)\big)].$$

Therefore, it holds that

$$\begin{aligned}
\mathbf{Reg}_{\mathsf{DM}} &= \sum_{t=1}^{T} \mathbb{E}_{\pi \sim p^t}\big[f^{M^\star}(\pi_{M^\star}) - f^{M^\star}(\pi)\big] \\
&\le \sum_{t=1}^{T} \Delta + \mathbb{E}_{\pi^\star \sim q^\star}\mathbb{E}_{\pi^t \sim p^t}\big[f^{M^\star}(\pi^\star) - f^{M^\star}(\pi^t)\big] \\
&= T\Delta + \gamma \sum_{t=1}^{T} \mathbb{E}_{\pi^\star \sim q^\star} \mathrm{Err}(p^t, \ell^t; q^t, M^\star, \pi^\star) \\
&\quad + \sum_{t=1}^{T} \mathbb{E}_{\pi^\star \sim q^\star} \underbrace{\Big[\mathbb{E}_{\pi^t \sim p^t}\big[f^{M^\star}(\pi^\star) - f^{M^\star}(\pi^t)\big] - \gamma \mathrm{Err}(p^t, \ell^t; q^t, M^\star, \pi^\star)\Big]}_{=\Gamma_{q^t, \gamma}(p^t, \ell^t; M^\star, \pi^\star)} \\
&\le T\Delta + \gamma(\log(1/q(\Pi^\star)) + \log(1/\delta)) + \sum_{t=1}^{T} \Gamma_{q^t, \gamma}(p^t, \ell^t) \\
&\le T\big(\Delta + \mathsf{exo}_{1/\gamma}(\mathcal{M})\big) + \gamma(\log(1/q(\Pi^\star)) + \log(1/\delta)).
\end{aligned}$$

Applying Theorem G.2 completes the proof. $\square$

---

[4]We remark that their proof applies as long as $f^M$ can be linearly extended to $\mathrm{co}(\mathcal{M})$.

**Proposition G.4.** *For any $q' \in \Delta(\Pi)$, it holds that*

$$\sum_{t=1}^{T} \mathbb{E}_{\pi \sim q'}[\ell^t(\pi; \pi^t, o^t)] - \log \mathbb{E}_{\pi \sim q^t}\left[\exp\left(\ell^t(\pi; \pi^t, o^t)\right)\right] \leq D_{\mathrm{KL}}(q' \| q).$$

*Proof.* This is essentially the standard guarantee of exponential weight updates. For simplicity, we assume $\Pi$ is discrete. Then, by definition,

$$q^t(\pi) = \frac{q(\pi) \exp\left(\sum_{s=1}^{t} \ell^s(\pi; \pi^s, o^s)\right)}{\sum_{\pi' \in \Pi} q(\pi') \exp\left(\sum_{s=1}^{t-1} \ell^s(\pi'; \pi^s, o^s)\right)},$$

and hence

$$\log \mathbb{E}_{\pi \sim q^t}\left[\exp\left(\ell^t(\pi; \pi^t, o^t)\right)\right] = \log \mathbb{E}_{\pi \sim q} \exp\left(\sum_{s=1}^{t} \ell^s(\pi; \pi^s, o^s)\right)$$

$$- \log \mathbb{E}_{\pi \sim q} \exp\left(\sum_{s=1}^{t-1} \ell^s(\pi; \pi^s, o^s)\right).$$

Therefore, taking summation over $t = 1, \cdots, T$, we have

$$-\sum_{t=1}^{T} \log \mathbb{E}_{\pi \sim q^t}\left[\exp\left(\ell^t(\pi; \pi^t, o^t)\right)\right] = -\log \mathbb{E}_{\pi \sim q}\left[\exp\left(\sum_{t=1}^{T} \ell^t(\pi; \pi^t, o^t)\right)\right].$$

The proof is then completed by the following basic fact of KL divergence: for any function $h : \Pi \to \mathbb{R}$,

$$\mathbb{E}_{\pi \sim q'}[h(\pi)] \leq \log \mathbb{E}_{\pi \sim q} \exp(h(\pi)) + D_{\mathrm{KL}}(q' \| q).$$

$\square$

### G.4 Application: Structured bandits

We now instantiate our general results to give tighter guarantees for structured bandits, improving the upper bounds in Section 4.2.

**DEC for structured bandits.** We consider the same structured bandit protocol as in Section 4.2; recall that $\mathcal{H}$ denotes the reward function class and $\mathcal{M}_{\mathcal{H}}$ denotes the induced model class. In what follows, we simplify the results in Theorem 15 to be stated purely in terms of $\mathcal{H}$. For a reference value function $\bar{h} : \mathcal{C} \times \mathcal{A} \to [0, 1]$, we define

$$\mathsf{r\text{-}dec}_\varepsilon^{\mathsf{c}}(\mathcal{H}, \bar{h}) := \inf_{p \in \Delta(\Pi)} \sup_{h \in \mathcal{H}} \left\{ \mathbb{E}_{\pi \sim p}\left[h(\pi_h) - h(\pi)\right] \,\Big|\, \mathbb{E}_{\pi \sim p}(h(\pi) - \bar{h}(\pi))^2 \leq \varepsilon^2 \right\},$$

where we recall that $\pi_h := \max_{\pi \in \Pi} h(\pi)$. We then define the DEC for $\mathcal{H}$ as

$$\mathsf{r\text{-}dec}_\varepsilon^{\mathsf{c}}(\mathcal{H}) = \sup_{\bar{h} \in \mathrm{co}(\mathcal{H})} \mathsf{r\text{-}dec}_\varepsilon^{\mathsf{c}}(\mathcal{H} \cup \{\bar{h}\}, \bar{h}).$$

As a corollary of Theorem G.3, the $\mathsf{r\text{-}dec}_\varepsilon^{\mathsf{c}}(\mathcal{H})$ and $\log \mathsf{N}_{\mathrm{frac}}(\mathcal{H}, \Delta)$ together provide an upper bound for structured bandits with $\mathcal{H}$.

**Theorem G.5.** *Let $\mathcal{H}$ be given. Suppose that $\Pi$ is finite, and that $\varepsilon \mapsto \mathsf{r\text{-}dec}_\varepsilon^{\mathsf{c}}(\mathrm{co}(\mathcal{H}))$ satisfies moderate decay as a function of $\varepsilon$ (Assumption 3) with constant $c_{\mathrm{reg}}$. Let $\bar{\varepsilon}(T) \asymp \sqrt{\log \mathsf{N}_{\mathrm{frac}}(\mathcal{H}, \Delta)/T}$. The Algorithm 1 ensures that high probability,*

$$\mathbf{Reg}_{\mathsf{DM}} \leq T \cdot \Delta + O(c_{\mathrm{reg}} T \sqrt{\log T}) \cdot \mathsf{r\text{-}dec}_{\bar{\varepsilon}(T)}^{\mathsf{c}}(\mathrm{co}(\mathcal{H})).$$

*As a corollary, the minimax sample complexity of structured bandit learning with $\mathcal{H}$ is bounded as*

$$\max\left\{T^{\mathsf{DEC}}(\mathcal{H}, \Delta), \log \mathsf{N}_{\mathrm{frac}}(\mathcal{H}, 2\Delta)\right\} \lesssim T^\star(\mathcal{M}_{\mathcal{H}}, \Delta) \lesssim T^{\mathsf{DEC}}(\mathrm{co}(\mathcal{H}), \Delta) \cdot \log \mathsf{N}_{\mathrm{frac}}(\mathcal{H}, \Delta/2),$$

(44)

*where we denote $T^{\mathsf{DEC}}(\mathcal{H}, \Delta) = \inf_{\varepsilon \in (0,1)} \{\varepsilon^{-2} : \mathsf{r\text{-}dec}_\varepsilon^{\mathsf{c}}(\mathcal{H}) \leq \Delta\}$ (following Eq. (15)) and omit logarithmic factors and dependence on the constant $c_{\mathrm{reg}}$.*

There are many standard structured bandit problems where the value function class $\mathcal{H}$ is convex, including multi-armed bandits, linear bandits, and non-parametric bandits (with smoothness [72], or concavity [54], or sub-modularity [65], or etc.). For these problem classes, the complexity of no-regret learning is completely characterized by the DEC of $\mathcal{H}$ and the fractional covering number $\mathsf{N}_{\mathrm{frac}}(\mathcal{H}, \Delta)$ (up to a quadratic factor).

We also note that the lower bound of Eq. (44) is proven for Gaussian noise, while our upper bound applies to a much more general class of reward distributions (with bounded variance).

### G.5 Application: Contextual bandits with general function approximation

Next, we instantiate our general results for stochastic contextual bandits with general function approximation, generalizing the structured bandit problem. We consider the stochastic contextual bandit problem with context space $\mathcal{C}$, action space $\mathcal{A}$, and a reward function class $\mathcal{H} \subseteq (\mathcal{C} \times \mathcal{A} \to [0,1])$. This problem is a special case of the DMSO setting with decision space $\Pi = (\mathcal{C} \to \mathcal{A})$, and the environment is specified by a tuple $(h_\star \in \mathcal{H}, \nu_\star \in \Delta(\mathcal{C}))$. The protocol is as follows: For each round $t$, the environment draws $c^t \sim \nu$, and the learner takes action $a^t = \pi^t(c^t)$ based on the decision $\pi^t : \mathcal{C} \to \mathcal{A}$, and receives a reward $r^t \sim \mathcal{N}(h_\star(c^t, a^t), 1)$.

We can formulate the model class as follows. For a reward function $h \in \mathcal{H}$ and context distribution $\nu \in \Delta(\mathcal{C})$, the corresponding model $M_{h,\nu}$ is specified as

$$(c, a, r) \sim M_{h,\nu}(\pi): \quad c \sim \nu, a = \pi(a), r \sim \mathcal{N}(h(c,a), 1).$$

Let $\mathcal{M}_{\mathcal{H}} = \{M_{h,\nu} : h \in \mathcal{H}, \nu \in \Delta(\mathcal{C})\}$ be the induced model class of contextual bandits. Following Appendix G.4, we instantiate Theorem 15 to provide characterization of learning $\mathcal{M}_{\mathcal{H}}$.

**DEC for contextual bandits.** For any context $c \in \mathcal{C}$, the value function class $\mathcal{H}$ induces a restricted value function class $\mathcal{H}|_c = \{h(c, \cdot) : h \in \mathcal{H}\}$, which corresponds to a (non-contextual) bandit function class. We define the following variant of the DEC

$$\mathsf{r\text{-}dec}_\varepsilon^{\mathsf{c}}(\mathcal{H}) := \sup_{c \in \mathcal{C}} \mathsf{r\text{-}dec}_\varepsilon^{\mathsf{c}}(\mathcal{H}|_c),$$

which corresponds to the maximum of the *per-context* DEC over all contexts. We also define $T^{\mathsf{DEC}}(\mathcal{H}, \Delta) = \inf_{\varepsilon \in (0,1)} \{\varepsilon^{-2} : \mathsf{r\text{-}dec}_\varepsilon^{\mathsf{c}}(\mathcal{H}) \le \Delta\}$, following Eq. (15).

**Fractional covering number for contextual bandits.** Specializing the fractional covering number to contextual bandits, we define

$$\mathsf{N}_{\mathrm{frac}}(\mathcal{H}, \Delta) := \inf_{p \in \Delta(\Pi)} \sup_{h \in \mathcal{H}, \nu \in \Delta(\mathcal{C})} \frac{1}{p(\pi : \mathbb{E}_{c \sim \nu}[h(c, \pi_h(c)) - h(c, \pi(c))] \le \Delta)}, \qquad (45)$$

where $\pi_h \in \Pi$ is defined via $\pi_h(c) := \arg\max_{a \in \mathcal{A}} h(c, a)$ for $c \in \mathcal{C}$.

Intuitively, the value of the fractional covering number $\log \mathsf{N}_{\mathrm{frac}}(\mathcal{H}, \Delta)$ for contextual bandits captures the difficulty of estimating optimal actions, but also the difficulty of generalizing across contexts. For example, when we consider the *unstructured* contextual bandit problems (i.e., $\mathcal{H} = (\mathcal{C} \times \mathcal{A} \to [0,1])$), it holds that $\log \mathsf{N}_{\mathrm{frac}}(\mathcal{H}, \Delta) = |\mathcal{C}| \log |\mathcal{A}|$, but in general we can have $\log \mathsf{N}_{\mathrm{frac}}(\mathcal{H}, \Delta) \ll \log |\Pi| = |\mathcal{C}| \log |\mathcal{A}|$.

As a corollary of Theorem 15, we derive the following upper and lower bounds on the complexity of contextual bandit learning with $\mathcal{H}$.

**Theorem G.6.** *Let $\mathcal{H}$ be given. Suppose that both the context space $\mathcal{C}$ and the action space $\mathcal{A}$ are finite, and that $\varepsilon \mapsto \mathsf{r\text{-}dec}_\varepsilon^{\mathsf{c}}(\mathrm{co}(\mathcal{H}))$ satisfies moderate decay as a function of $\varepsilon$ (Assumption 3) with constant $c_{\mathrm{reg}}$. Let $\bar{\varepsilon}(T) \asymp \sqrt{\log \mathsf{N}_{\mathrm{frac}}(\mathcal{H}, \Delta)/T}$. Then Algorithm 1 ensures that with high probability, ;*

$$\mathbf{Reg}_{\mathsf{DM}} \le T \cdot \Delta + O(c_{\mathrm{reg}} T \sqrt{\log T}) \cdot \mathsf{r\text{-}dec}_{\bar{\varepsilon}(T)}^{\mathsf{c}}(\mathrm{co}(\mathcal{H})).$$

*As a corollary, the complexity of learning $\mathcal{M}_{\mathcal{H}}$ is bounded by*

$$\max\left\{ T^{\mathsf{DEC}}(\mathcal{H}, \Delta), \frac{\log \mathsf{N}_{\mathrm{frac}}(\mathcal{H}, 2\Delta)}{\log |\mathcal{C}|} \right\} \lesssim T^\star(\mathcal{M}_{\mathcal{H}}, \Delta) \lesssim T^{\mathsf{DEC}}(\mathrm{co}(\mathcal{H}), \Delta) \cdot \log \mathsf{N}_{\mathrm{frac}}(\mathcal{H}, \Delta/2),$$

(46)

*omitting dependence on $c_{\mathrm{reg}}$ and logarithmic factors.*

By definition, we have $\mathsf{r\text{-}dec}^{\mathsf{c}}_{\varepsilon}(\mathrm{co}(\mathcal{H})) = \mathsf{r\text{-}dec}^{\mathsf{c}}_{\varepsilon}(\mathcal{H})$ if the *per-context* value function class $\mathcal{H}|_c$ is convex for every context $c \in \mathcal{C}$. Natural settings in which $\mathcal{H}|_c$ is convex include contextual linear bandits [28], contextual non-parametric bandits [22], contextual concave bandits [54], etc. For these problem classes, the complexity of no-regret learning is completely characterized by the DEC of $\mathcal{H}$ and the newly proposed $\mathsf{N}_{\mathrm{frac}}(\mathcal{H}, \Delta)$ (up to a quadratic factor and a factor of $\log |\mathcal{C}|$).

As a concrete example. we can derive upper bounds based on the fractional covering number for finite-action contextual bandits as follows.

**Corollary G.7.** *For any value function class* $\mathcal{H}$, *Algorithm 1 ensures the following regret bound with high probability.*

$$\mathbf{Reg}_{\mathsf{DM}}(T) \leq T \cdot \Delta + O\left(\sqrt{T|\mathcal{A}| \cdot \log \mathsf{N}_{\mathrm{frac}}(\mathcal{H}, \Delta)}\right).$$

Compared to the well-known regret bound of $O(\sqrt{T|\mathcal{A}| \cdot \log |\mathcal{H}|})$ for learning any with any finite contextual bandit class $\mathcal{H}$ [38, 77], this result above always provides a tighter upper bound, as $\log \mathsf{N}_{\mathrm{frac}}(\mathcal{H}, \Delta) \leq \log |\mathcal{H}|$. For certain (very simple) function classes $\mathcal{H}$, the quantity $\log \mathsf{N}_{\mathrm{frac}}(\mathcal{H}, \Delta)$ can be much smaller than $\log |\mathcal{H}|$ (for details, see Example 14). More importantly, $\log \mathsf{N}_{\mathrm{frac}}(\mathcal{H}, \Delta)$ leads to lower bounds for *any* contextual bandit function class (Theorem G.6). By contrast, lower bounds for structured contextual bandits in prior work have been proven in a case-by-case fashion (for specific value function classes $\mathcal{H}$).

# H    Proofs from Section 4 and Appendix G

In this section, we mainly focus on no-regret learning, and we present the regret upper and lower bounds in terms of DEC and $\log \mathsf{N}_{\mathrm{frac}}(\mathcal{M}, \Delta)$. The results can be generalized immediately to PAC learning.

## H.1    Proof of Theorem 10

Fix an arbitrary reference model $\overline{M} \in (\Pi \to \Delta(\mathcal{O}))$ such that Assumption 2 holds. We remark that $\overline{M}$ is not necessarily in $\mathcal{M}$ or $\mathrm{co}(\mathcal{M})$.

We only need to prove the following fact.

**Fact.** If $T < \frac{\log \mathsf{N}_{\mathrm{frac}}(\mathcal{M}, \Delta) - 2}{2 C_{\mathrm{KL}}}$, then for any $T$-round algorithm $\mathsf{ALG}$, there exists a model $M \in \mathcal{M}$ such that $\mathbf{Risk}_{\mathsf{DM}}(T) \geq \Delta$ with probability at least $\frac{1}{2}$ under $\mathbb{P}^{M, \mathsf{ALG}}$.

*Proof.* By the definition (17) of $\mathsf{N}_{\mathrm{frac}}(\mathcal{M}, \Delta)$, we know

$$\frac{1}{\mathsf{N}_{\mathrm{frac}}(\mathcal{M}, \Delta)} := \sup_{p \in \Delta(\Pi)} \inf_{M \in \mathcal{M}} p(\pi : L(M, \pi) \leq \Delta).$$

Therefore, we have

$$\inf_{M \in \mathcal{M}} p_{\overline{M}, \mathsf{ALG}}(\pi : L(M, \pi) \leq \Delta) \leq \frac{1}{\mathsf{N}_{\mathrm{frac}}(\mathcal{M}, \Delta)},$$

and hence there exists $M \in \mathcal{M}$ such that

$$T < \frac{\log\left(1/p_{\overline{M}, \mathsf{ALG}}(\pi : L(M, \pi) \leq \Delta)\right) - 2}{2 C_{\mathrm{KL}}}.$$

Notice that by the chain rule of KL divergence, we have

$$D_{\mathrm{KL}}(\mathbb{P}^{M, \mathsf{ALG}} \parallel \mathbb{P}^{\overline{M}, \mathsf{ALG}}) = \mathbb{E}^{M, \mathsf{ALG}}\left[\sum_{t=1}^{T} D_{\mathrm{KL}}(M(\pi^t) \parallel \overline{M}(\pi^t))\right] \leq T C_{\mathrm{KL}}.$$

Hence, using data-processing inequality,

$$D_{\mathrm{KL}}(p_{M, \mathsf{ALG}} \parallel p_{\overline{M}, \mathsf{ALG}}) < \frac{\log\left(1/p_{\overline{M}, \mathsf{ALG}}(\pi : L(M, \pi) \leq \Delta)\right) - 2}{2}$$

$$\leq D_{\mathrm{KL}}(1/2 \parallel p_{\overline{M}, \mathsf{ALG}}(\pi : L(M, \pi) \leq \Delta)).$$

This immediately implies $p_{M, \mathsf{ALG}}(\pi : L(M, \pi) \leq \Delta) < \frac{1}{2}$ by the monotonicity of KL divergence. $\qquad\square$

## H.2 Proof of Theorem 11

In this section, we present an algorithm based on reduction to multi-arm bandits (Algorithm 2) that achieves the desired upper bound. For the application to bandits with Gaussian rewards, we relax the assumption $R : \mathcal{O} \to [0,1]$ as follows.

**Assumption 5.** *For any $M \in \mathcal{M}$ and $\pi \in \Pi$, the random variable $R(o)$ is 1-sub-Gaussian under $o \sim M(\pi)$.*

Suppose that $\Delta > 0$ is given, and fix a distribution $p_\Delta^\star$ that attains the infimum of (17). Based on $p_\Delta^\star$, we consider a reduced decision space $\Pi_{\mathsf{sub}} \subset \Pi$, generated as

$$\Pi_{\mathsf{sub}} = \{\pi^{(1)}, \cdots, \pi^{(N)}\}, \qquad \pi^{(1)}, \cdots, \pi^{(N)} \sim p_\Delta^\star \text{ independently},$$

where we set $N = \mathsf{N}_{\mathrm{frac}}(\mathcal{M}, \Delta) \log(1/\delta)$. Then the space $\Pi_{\mathsf{sub}}$ is guaranteed to contain a near-optimal decision, as follows.

**Lemma H.1.** *With probability at least $1 - \delta$, there exists $\pi \in \Pi_{\mathsf{sub}}$ such that $L(M^\star, \pi) \leq \Delta$.*

Therefore, we can then regard $M^\star$ as a $N$-arm bandit instance with action space $\mathcal{A} = \Pi_{\mathsf{sub}}$, and for each pull of an arm $\pi \in \mathcal{A}$, the stochastic reward $r$ is generated as $r = R(o), o \sim M^\star(\pi)$. Then, we pick a standard bandit algorithm BanditALG, e.g. the UCB algorithm (see e.g. Lattimore and Szepesvári [57]), and apply it to the multi-arm bandit instance $M_{\mathsf{Bandit}}^\star$, and the guarantee of BanditALG yields

$$\sum_{t=1}^{T} \max_{\pi' \in \Pi_{\mathsf{sub}}} f^{M^\star}(\pi') - f^{M^\star}(\pi^t) \leq O\left(\sqrt{TN \log(T/\delta)}\right).$$

with probability at least $1 - \delta$. Therefore, we have

$$\mathbf{Reg}_{\mathsf{DM}}(T) \leq T \cdot (f^{M^\star}(\pi_{M^\star}) - \max_{\pi' \in \Pi_{\mathsf{sub}}} f^{M^\star}(\pi')) + O\left(\sqrt{TN \log(T/\delta)}\right)$$

$$\leq T \cdot \Delta + O\left(\sqrt{TN \log(T/\delta)}\right),$$

with probability at least $1 - 2\delta$. This gives the desired upper bound, and we summarize the full algorithm in Algorithm 2. $\qquad\square$

**Proof of Lemma H.1.** By definition,

$$\mathbb{P}\left(\forall i \in [N], L(M^\star, \pi^{(i)}) > \Delta\right) \leq p_\Delta^\star(\pi : L(M^\star, \pi) > \Delta)^N$$

$$\leq \left(1 - \frac{1}{\mathsf{N}_{\mathrm{frac}}(\mathcal{M}, \Delta)}\right)^N$$

$$\leq \exp\left(-\frac{N}{\mathsf{N}_{\mathrm{frac}}(\mathcal{M}, \Delta)}\right) \leq \delta.$$

$\qquad\square$

## H.3 Proof of Lemma G.1

**Proof of the upper bound.** Take a minimal $\Delta$-covering of $\Pi_\mathcal{M}$, i.e., a set $\{\pi^1, \cdots, \pi^n\} \subseteq \Pi$ such that for all $M \in \mathcal{M}$, there exists $i \in [n]$ such that $\rho(\pi_M, \pi^i) \leq \Delta$. Therefore, we may consider the distribution $p = \mathrm{Unif}(\{\pi^1, \cdots, \pi^n\})$, which guarantee

$$\mathsf{N}_{\mathrm{frac}}(\mathcal{M}, \Delta) \leq \sup_{M \in \mathcal{M}} \frac{1}{p(\pi : \rho(\pi_M, \pi) \leq \Delta)} \leq n = \mathsf{N}(\Pi_\mathcal{M}, \Delta).$$

**Proof of the lower bound.** Consider the maximal $2\Delta$-packing of $\Pi_\mathcal{M}$, i.e., let $\{\pi^1, \cdots, \pi^m\} \subseteq \Pi_\mathcal{M}$ be a maximal set such that $\rho(\pi^i, \pi^j) > 2\Delta$ for any $i \neq j$. Then, by the duality between packing and covering, the set $\{\pi^1, \cdots, \pi^m\}$ form a $2\Delta$-covering of $\Pi_\mathcal{M}$, and hence we have $m \geq \mathsf{N}(\Pi_\mathcal{M}, 2\Delta)$. On the other hand, the sets $\Pi^i := \{\pi : \rho(\pi, \pi^j) \leq \Delta\}$ are pairwise disjoint, and hence for any $p \in \Delta(\Pi)$, we have

$$m \cdot \inf_{M \in \mathcal{M}} p(\pi : \rho(\pi_M, \pi) \leq \Delta) \leq \sum_{i=1}^{m} p(\pi : \rho(\pi^i, \pi) \leq \Delta) \leq 1.$$

Therefore, it holds that $\mathsf{N}_{\mathrm{frac}}(\mathcal{M}, \Delta) \geq m \geq \mathsf{N}(\Pi_\mathcal{M}, 2\Delta)$. $\qquad\square$

---

**Algorithm 2** A reduction algorithm based on the fractional covering number

---

**Input:** Problem $(\mathcal{M}, \Pi)$, parameter $\Delta, \delta > 0, T \geq 1$, Algorithm BanditALG for multi-arm bandits.

1: Set

$$p_\Delta^\star = \arg \inf_{p \in \Delta(\Pi)} \sup_{M \in \mathcal{M}} \frac{1}{p(\pi : L(M, \pi) \leq \Delta)}. \tag{47}$$

2: Set $N = \mathsf{N}_{\mathrm{frac}}(\mathcal{M}, \Delta) \log(1/\delta)$ and sample the decision subspace $\Pi_{\mathsf{sub}} = \{\pi^{(1)}, \cdots, \pi^{(N)}\} \subset \Pi$ as

$$\pi^{(1)}, \cdots, \pi^{(N)} \sim p_\Delta^\star \text{ independently.}$$

3: Run the bandit algorithm BanditALG on the instance $M_{\mathsf{Bandit}}^\star$ for $T$ rounds.

---

### H.4 Proof of Example 12

It remains to prove Eq. (39). More generally, we prove the following lemma.

**Lemma H.2.** *For model class $\mathcal{M} = \bigcup_{i=1}^n \mathcal{M}_i$, it holds that*

$$\mathsf{N}_{\mathrm{frac}}(\mathcal{M}, \Delta) \leq \sum_{i=1}^n \mathsf{N}_{\mathrm{frac}}(\mathcal{M}_i, \Delta).$$

**Proof of Lemma H.2** For each $i \in [n]$, we define $\lambda_i = \frac{\mathsf{N}_{\mathrm{frac}}(\mathcal{M}_i, \Delta)}{\sum_{j=1}^n \mathsf{N}_{\mathrm{frac}}(\mathcal{M}_j, \Delta)}$.

Fix $p_1, \cdots, p_n \in \Delta(\Pi)$. Then, let us consider the distribution $p = \sum_{i=1}^n \lambda_i p_i \in \Delta(\Pi)$. For any model $M \in \mathcal{M}$, there exists $i \in \mathcal{M}_i$, and hence $p(\pi : L(M, \pi) \leq \Delta) \geq \lambda_i \min_{M_i \in \mathcal{M}_i} p_i(\pi : L(M_i, \pi) \leq \Delta)$. Therefore, it holds that

$$\inf_{M \in \mathcal{M}} p(\pi : L(M, \pi) \leq \Delta) \geq \min_{i \in [n]} \lambda_i \inf_{M \in \mathcal{M}_i} p_i(\pi : L(M, \pi) \leq \Delta).$$

In other words,

$$\sup_{M \in \mathcal{M}} \frac{1}{p(\pi : L(M, \pi) \leq \Delta)} \geq \max_{i \in [n]} \frac{1}{\lambda_i} \sup_{M \in \mathcal{M}_i} \frac{1}{p_i(\pi : L(M, \pi) \leq \Delta)}.$$

Taking infimum over $p_1, \cdots, p_n \in \Delta(\Pi)$ gives

$$\begin{aligned} \mathsf{N}_{\mathrm{frac}}(\mathcal{M}, \Delta) = \inf_{p \in \Delta(\Pi)} \sup_{M \in \mathcal{M}} \frac{1}{p(\pi : L(M, \pi) \leq \Delta)} &\geq \max_{i \in [n]} \frac{1}{\lambda_i} \sup_{M \in \mathcal{M}_i} \frac{1}{p_i(\pi : L(M, \pi) \leq \Delta)} \\ &\leq \inf_{p_1, \cdots, p_n \in \Delta(\Pi)} \max_{i \in [n]} \frac{1}{\lambda_i} \sup_{M \in \mathcal{M}_i} \frac{1}{p_i(\pi : L(M, \pi) \leq \Delta)} \\ &= \max_{i \in [n]} \frac{1}{\lambda_i} \inf_{p_i \in \Delta(\Pi)} \sup_{M \in \mathcal{M}_i} \frac{1}{p_i(\pi : L(M, \pi) \leq \Delta)} \\ &= \max_{i \in [n]} \frac{1}{\lambda_i} \cdot \mathsf{N}_{\mathrm{frac}}(\mathcal{M}_i, \Delta) = \sum_{i=1}^n \mathsf{N}_{\mathrm{frac}}(\mathcal{M}_i, \Delta), \end{aligned}$$

where the last line follows from the definition of $\lambda_1, \cdots, \lambda_n$. This is the desired result. $\qquad \square$

### H.5 Proof of Theorem 14

We first state the following more general result, and Theorem 14 is then a direct corollary (under Assumption 3). Analoguous guarantees also hold for PAC learning.

**Theorem H.3.** *Let $T \geq 1, \delta \in (0, 1)$. With suitably chosen prior $q \in \Delta(\Pi)$, ExO$^+$ (Algorithm 1) achieves with probability at least $1 - \delta$:*

$$\frac{1}{T} \mathsf{Reg}_{\mathsf{DM}} \leq \Delta + \mathsf{r\text{-}dec}_{\gamma/8}^{\mathrm{o}}(\mathrm{co}(\mathcal{M})) + \gamma \frac{\log \mathsf{N}_{\mathrm{frac}}(\mathcal{M}, \Delta) + \log(1/\delta)}{T}. \tag{48}$$

*In particular, when $\mathcal{M}$ is a reward-maximization problem class (Example 1), $\mathsf{ExO}^+$ achieves (with a suitable parameter $\gamma$) that with probability at least $1 - \delta$:*

$$\frac{1}{T}\mathbf{Reg}_{\mathsf{DM}} \leq \Delta + C\sqrt{\log(T)} \cdot \overline{\mathsf{r\text{-}dec}}_{\bar{\varepsilon}(T)}^{\mathrm{c}}(\mathrm{co}(\mathcal{M})), \tag{49}$$

*where $C$ is an absolute constant, $\bar{\varepsilon}(T) = \sqrt{\frac{\log \mathsf{N}_{\mathrm{frac}}(\mathcal{M},\Delta) + \log(1/\delta)}{T}}$, and the modified version of constrained regret-DEC is defined as*

$$\overline{\mathsf{r\text{-}dec}}_{\varepsilon}^{\mathrm{c}}(\mathrm{co}(\mathcal{M})) := \varepsilon \cdot \sup_{\varepsilon' \in [\varepsilon, 1]} \frac{\mathsf{r\text{-}dec}_{\varepsilon'}^{\mathrm{c}}(\mathrm{co}(\mathcal{M}))}{\varepsilon'}. \tag{50}$$

**Remark H.4** (Upper bound without regularity condition)**.** In Theorem 14 (and Eq. (49)), we assume that (1) $\mathcal{M}$ is a reward-maximization problem, and (2) the constrained regret-DEC of $\mathrm{co}(\mathcal{M})$ satisfies certain regularity condition (Assumption 3). We can relax these two assumptions and obtain a weaker upper bound. Specifically, we may only assume that $f^M : \Pi \to [0, 1]$ is affine with respect to $M \in \mathrm{co}(\mathcal{M})$ (cf. Theorem G.2). In this case, we can still bound the regret of $\mathsf{ExO}^+$ as

$$\frac{1}{T}\mathbf{Reg}_{\mathsf{DM}} \leq \Delta + \mathsf{r\text{-}dec}_{\sqrt{\gamma/8}}^{\mathrm{c}}(\mathrm{co}(\mathcal{M})) + \gamma\frac{\log \mathsf{N}_{\mathrm{frac}}(\mathcal{M}, \Delta) + \log(1/\delta)}{T}, \tag{51}$$

which follows from Eq. (48) and the fact that (by definition)

$$\mathsf{r\text{-}dec}_{\gamma}^{\mathrm{o}}(\mathrm{co}(\mathcal{M})) \leq \mathsf{r\text{-}dec}_{\sqrt{\gamma}}^{\mathrm{c}}(\mathrm{co}(\mathcal{M})), \qquad \forall \gamma > 0.$$

In particular, using Eq. (51) above, we can show that (omitting poly-logarithmic factors)

$$T^{\star}(\mathcal{M}, \Delta) \lesssim \frac{1}{\Delta} \cdot T^{\mathsf{DEC}}(\mathrm{co}(\mathcal{M}), \Delta/3) \cdot \log \mathsf{N}_{\mathrm{frac}}(\mathcal{M}, \Delta/2).$$

This is worse than the upper bound of Theorem 15 by (roughly) a factor of $\Delta^{-1}$.

**Proof of Theorem H.3.** By the definition (17) of $\mathsf{N}_{\mathrm{frac}}(\mathcal{M}, \Delta)$, we know

$$\frac{1}{\mathsf{N}_{\mathrm{frac}}(\mathcal{M}, \Delta)} := \sup_{p \in \Delta(\Pi)} \inf_{M \in \mathcal{M}} p(\pi : L(M, \pi) \leq \Delta).$$

Therefore, there exists $q \in \Delta(\Pi)$ such that

$$\inf_{M \in \mathcal{M}} q(\pi : L(M, \pi) \leq \Delta) \geq \frac{1}{\mathsf{N}_{\mathrm{frac}}(\mathcal{M}, \Delta)},$$

We then instantiate Algorithm 1 with such a prior $q$, and Eq. (48) follows immediately from Theorem G.3. To prove Eq. (49), we invoke the following structural result that relates offset DEC to constrained DEC.

**Theorem H.5.** *Suppose that Assumption 4 holds for the model class $\mathcal{M}$. Then for any $\varepsilon \in (0, 1]$, it holds that*

$$\inf_{\gamma > 0} \left( \mathsf{r\text{-}dec}_{\gamma}^{\mathrm{o}}(\mathcal{M}) + \gamma\varepsilon^2 \right) \leq \left( 3\sqrt{\lfloor \log_2(2/\varepsilon) \rfloor} + 2 \right) \cdot \left( \overline{\mathsf{r\text{-}dec}}_{\varepsilon}^{\mathrm{c}}(\mathcal{M}) + C_{\mathrm{r}}\varepsilon \right).$$

Under the assumption that $\varepsilon \mapsto \mathsf{r\text{-}dec}_{\varepsilon}^{\mathrm{c}}(\mathrm{co}(\mathcal{M}))$ is of moderate decay with a constant $c_{\mathrm{reg}}$, we have

$$\overline{\mathsf{r\text{-}dec}}_{\varepsilon}^{\mathrm{c}}(\mathrm{co}(\mathcal{M})) \leq c_{\mathrm{reg}}\mathsf{r\text{-}dec}^{\mathrm{c}}\varepsilon(\mathcal{M}), \qquad \forall \varepsilon \in (0, 1].$$

Hence, Eq. (49) follows from (48) as long as the parameter $\gamma$ is chosen according to Eq. (H.5). $\square$

### H.5.1 Proof of Theorem H.5

Fix a $\varepsilon \in (0, 1]$ and $\overline{M} \in \mathrm{co}(\mathcal{M})$. We only need to prove the following result:

**Claim.** Suppose that $\mathsf{r\text{-}dec}_{\varepsilon'}^{\mathrm{c}}(\mathcal{M}, \overline{M}) \leq D\varepsilon'$ for all $\varepsilon' \in [\varepsilon, 1]$. Then there exists $\gamma = \gamma(D, \varepsilon)$ such that

$$\mathsf{r\text{-}dec}_{\gamma}^{\mathrm{o}}(\mathcal{M}) + \gamma\varepsilon^2 \leq \left( 3\sqrt{\lfloor \log_2(2/\varepsilon) \rfloor} + 2 \right) \cdot (D + C_{\mathrm{r}})\varepsilon.$$

Set $K = \lfloor \log_2(1/\varepsilon) \rfloor + 1$ and fix a parameter $c = c(\varepsilon) \in (0, \frac{1}{2}]$ to be specified later in proof. Define $\varepsilon_i := 2^{-i}$ for $i = 0, \cdots, K-1$ and $\varepsilon_K = \varepsilon$. We also define $\lambda_i := c\varepsilon \cdot 2^i$ for $i = 0, \cdots, K-1$, and $\lambda_K = 1 - \sum_{i=0}^{K-1} \lambda_i \geq c$.

Define $\Delta_i = \mathsf{r\text{-}dec}^{\mathsf{c}}_{\varepsilon_i}(\mathcal{M} \cup \{\overline{M}\}, \overline{M})$, and let $p_i$ attains the $\inf_p$. In the following, we choose $\gamma = \gamma(D, \varepsilon) = \frac{9(D + C_{\mathrm{r}})}{8c\varepsilon}$.

By definition of $p_i$, it holds that

$$\mathbb{E}_{\pi \sim p_i}[L(M, \pi)] \leq \Delta_i, \qquad \forall M \in \mathcal{M} \cup \{\overline{M}\} : \mathbb{E}_{\pi \sim p_i} D_{\mathrm{H}}^2\left(M(\pi), \overline{M}(\pi)\right) \leq \varepsilon_i^2.$$

In particular, we may abbreviate $\mathcal{M}_i := \{M \in \mathcal{M} : \mathbb{E}_{\pi \sim p_i} D_{\mathrm{H}}^2\left(M(\pi), \overline{M}(\pi)\right) \leq \varepsilon_i^2\}$, and it holds

$$f^M(\pi_M) \leq f^{\overline{M}}(\pi_{\overline{M}}) + \Delta_i + C_{\mathrm{r}}\varepsilon_i, \qquad \forall M \in \mathcal{M}_i.$$

Next, we choose $p = \sum_{i=0}^K \lambda_i p_i \in \Delta(\Pi)$, and we know

$$\mathbb{E}_{\pi \sim p}[L(\overline{M}, \pi)] \leq \sum_{i=0}^K \lambda_i \mathbb{E}_{\pi \sim p}[L(\overline{M}, \pi)] \leq \sum_{i=0}^K \lambda_i \Delta_i =: \Delta.$$

Fix a $M \in \mathcal{M}$. Let $j \in \{0, \cdots, K\}$ be the maximum index such that $M \in \mathcal{M}_j$. Such a $j$ must exists because $\mathcal{M} = \mathcal{M}_0$. Now,

$$\mathbb{E}_{\pi \sim p}[L(M, \pi)] = f^M(\pi_M) - f^{\overline{M}}(\pi_{\overline{M}}) + \mathbb{E}_{\pi \sim p}[L(\overline{M}, \pi)] + \mathbb{E}_{\pi \sim p}[f^{\overline{M}}(\pi) - f^M(\pi)]$$
$$\leq \Delta_j + C_{\mathrm{r}}\varepsilon_j + \Delta + C_{\mathrm{r}}\mathbb{E}_{\pi \sim p} D_{\mathrm{H}}\left(M(\pi), \overline{M}(\pi)\right).$$

Case 1: $j = K$. Then, using AM-GM inequality, we have

$$\mathbb{E}_{\pi \sim p}[L(M, \pi)] - \gamma \mathbb{E}_{\pi \sim p} D_{\mathrm{H}}^2\left(M(\pi), \overline{M}(\pi)\right) \leq \Delta_K + \varepsilon_K + \Delta + \frac{C_{\mathrm{r}}^2}{4\gamma}.$$

Case 2: $j < K$. Then for each $i > j$, it holds that $\mathbb{E}_{\pi \sim p_j} D_{\mathrm{H}}^2\left(M(\pi), \overline{M}(\pi)\right) > \varepsilon_j^2$, and hence

$$\mathbb{E}_{\pi \sim p} D_{\mathrm{H}}^2\left(M(\pi), \overline{M}(\pi)\right) \geq \sum_{i=j+1}^K \lambda_j \mathbb{E}_{\pi \sim p_j} D_{\mathrm{H}}^2\left(M(\pi), \overline{M}(\pi)\right) \geq \sum_{i=j+1}^K \lambda_j \varepsilon_j^2 \geq \frac{c\varepsilon \cdot \varepsilon_j}{2}.$$

Therefore, using AM-GM inequality,

$$\mathbb{E}_{\pi \sim p}[L(M, \pi)] - \gamma \mathbb{E}_{\pi \sim p} D_{\mathrm{H}}^2\left(M(\pi), \overline{M}(\pi)\right)$$
$$\leq \Delta_j + C_{\mathrm{r}}\varepsilon_j + \Delta + \frac{9C_{\mathrm{r}}^2}{4\gamma} - \frac{8}{9}\gamma \mathbb{E}_{\pi \sim p} D_{\mathrm{H}}^2\left(M(\pi), \overline{M}(\pi)\right)$$
$$\leq \Delta_j + C_{\mathrm{r}}\varepsilon_j + \Delta + \frac{9C_{\mathrm{r}}^2}{4\gamma} - \frac{8c\gamma\varepsilon}{9}\varepsilon_j.$$

By our choice of $\gamma$, we have $\gamma\varepsilon \geq \frac{9}{8c}\left(\frac{\Delta_j}{\varepsilon_j} + C_{\mathrm{r}}\right)$, and hence in both cases, we have

$$\mathbb{E}_{\pi \sim p}[L(M, \pi)] - \gamma \mathbb{E}_{\pi \sim p} D_{\mathrm{H}}^2\left(M(\pi), \overline{M}(\pi)\right) \leq \Delta + (D + C_{\mathrm{r}})\varepsilon + \frac{9C_{\mathrm{r}}^2}{4\gamma}.$$

Note that by definition, we have $\Delta \leq (cK + 1)D\varepsilon$ and $\gamma(\varepsilon) \cdot \varepsilon = \frac{9}{8c}(D + C_{\mathrm{r}})$, and hence

$$\mathsf{r\text{-}dec}^{\mathrm{o}}_{\gamma(\varepsilon)}(\mathcal{M}, \overline{M}) \leq (2D + C_{\mathrm{r}} + cKD + 2cC_{\mathrm{r}})\varepsilon.$$

Thus,

$$\mathsf{r\text{-}dec}^{\mathrm{o}}_{\gamma(\varepsilon)}(\mathcal{M}, \overline{M}) + \gamma(\varepsilon)\varepsilon^2 \leq \left(2D + C_{\mathrm{r}} + cK(D + C_{\mathrm{r}}) + \frac{9(D + C_{\mathrm{r}})}{8c}\right)\varepsilon_K.$$

Balancing $c$ and re-arranging yields the desired result. $\qquad\qquad\square$

## H.6 Proof of Theorem 15

Note that the minimax-optimal sample complexity $T^\star(\mathcal{M}, \Delta)$ is just a way to better illustrate our minimax regret upper and lower bounds. By the definition of $T^\star(\mathcal{M}, \Delta)$, we have

$$\frac{1}{T}\mathbf{Reg}^\star(\mathcal{M}, T) = \sup\{\Delta : T^\star(\mathcal{M}, \Delta) \leq T\}.$$

Under Assumption 3, the regret upper bound in Theorem 14 implies (up to $c_{\mathrm{reg}}$, $C_{\mathrm{KL}}$ and logarithmic factors)

$$\frac{1}{T}\mathbf{Reg}^\star(\mathcal{M}, T) \lesssim \mathsf{r\text{-}dec}^{\mathsf{c}}_{\bar{\varepsilon}(T)}(\mathcal{M}).$$

And the regret lower bound Theorem E.1 implies (up to $c_{\mathrm{reg}}$ and logarithmic factors)

$$\mathsf{r\text{-}dec}^{\mathsf{c}}_{\underline{\varepsilon}(T)}(\mathcal{M}) \lesssim \frac{1}{T}\mathbf{Reg}^\star(\mathcal{M}, T).$$

By the definition of $T^\star(\mathcal{M}, \Delta)$ and $T^{\mathsf{DEC}}(\mathcal{M}, \Delta)$, we then have

$$T^{\mathsf{DEC}}(\mathcal{M}, \Delta) \lesssim T^\star(\mathcal{M}, \Delta) \lesssim T^{\mathsf{DEC}}(\mathrm{co}(\mathcal{M}), \Delta) \cdot \log \mathsf{N}_{\mathrm{frac}}(\mathcal{M}, \Delta/2).$$

Together with Theorem 10, we prove that

$$\max\left\{T^{\mathsf{DEC}}(\mathcal{M}, \Delta), \frac{\log \mathsf{N}_{\mathrm{frac}}(\mathcal{M}, \Delta)}{C_{\mathrm{KL}}}\right\} \lesssim T^\star(\mathcal{M}, \Delta) \lesssim T^{\mathsf{DEC}}(\mathrm{co}(\mathcal{M}), \Delta) \cdot \log \mathsf{N}_{\mathrm{frac}}(\mathcal{M}, \Delta/2).$$

$\square$

## H.7 Proof of Theorem G.5

For the upper bound, we work with more general noise structure (beyond Gaussian noises). We define $\mathcal{M}_{\mathcal{H}, \mathbb{V}}$ to be the class of all bandits models with mean reward function in $\mathcal{H}$ and variance bounded by 1. Specifically, for any $M \in \mathcal{M}_{\mathcal{H}, \mathbb{V}}$, it is associated with a value function $h^M \in \mathcal{H}$, such that for any decision $\pi \in \Pi$, the distribution $M(\pi)$ of the random reward $r$ has mean $h^M(\pi)$ and variance at most 1.

We also recall that the subclass $\mathcal{M}_{\mathcal{H}} \subseteq \mathcal{M}_{\mathcal{H}, \mathbb{V}}$ is the bandit problem class with the standard Gaussian noise.

**Proof of Theorem G.5: lower bound of (44).** The lower bound with $\log \mathsf{N}_{\mathrm{frac}}(\mathcal{H}, \Delta)$ is exactly Corollary 12.

To prove the lower bound with $T^{\mathsf{DEC}}(\mathcal{H}, \Delta)$, we need to lower bound the DEC of $\mathcal{M}_{\mathcal{H}}$ in terms of the DEC of $\mathcal{H}$, as follows.

**Lemma H.6.** *Consider $\mathcal{M}^+ = \mathcal{M}_{\mathrm{co}(\mathcal{H}), \mathbb{V}}$ as the class of all reference models (Appendix E). Then,*

$$\max_{\overline{M} \in \mathcal{M}^+} \mathsf{r\text{-}dec}^{\mathsf{c}}_{\varepsilon}(\mathcal{M}_{\mathcal{H}} \cup \{\overline{M}\}, \overline{M}) \geq \mathsf{r\text{-}dec}^{\mathsf{c}}_{2\sqrt{2}\varepsilon}(\mathcal{H}). \tag{52}$$

Notice that for $\mathcal{M}^+$, Assumption 4 holds with $C_{\mathrm{r}} = \sqrt{10}$ (by Lemma C.5). Therefore, as a corollary of Theorem E.3: for any $T$-round algorithm ALG, there exists $M^\star \in \mathcal{M}_{\mathcal{H}}$ such that

$$\mathbf{Reg}_{\mathsf{DM}}(T) \geq \frac{T}{2} \cdot (\mathsf{r\text{-}dec}^{\mathsf{c}}_{\underline{\varepsilon}(T)}(\mathcal{H}) - 5\underline{\varepsilon}(T)) - 1 \tag{53}$$

with probability at least 0.01 under $\mathbb{P}^{M^\star, \mathrm{ALG}}$, where $\underline{\varepsilon}(T) = \frac{1}{50\sqrt{T}}$. Therefore, the lower bound in terms of $T^{\mathsf{DEC}}(\mathcal{H}, \Delta)$ follows immediately (using regularity condition Assumption 3).

Combining both lower bounds completes the proof. $\square$

**Proof of Theorem G.5: upper bound.** We apply Theorem H.3 similar to the proof of Theorem 15 (in Appendix H.2).

Using Theorem H.3, we know that $\mathsf{ExO}^+$ can be suitably instantiated on the model class $\mathcal{M}_{\mathcal{H},\mathbb{V}}$ so that with probability at least $1-\delta$,

$$\frac{1}{T}\mathbf{Reg}_{\mathsf{DM}} \leq \Delta + C\sqrt{\log(T)} \cdot \overline{\mathsf{r\text{-}dec}}^{\mathsf{c}}_{\bar{\varepsilon}(T)}(\mathrm{co}(\mathcal{M}_{\mathcal{H},\mathbb{V}})),$$

where $C$ is an absolute constant, $\bar{\varepsilon}(T) = \sqrt{\frac{\log \mathsf{N}_{\mathrm{frac}}(\mathcal{H},\Delta)+\log(1/\delta)}{T}}$. We only need to upper bound the $\overline{\mathsf{r\text{-}dec}}^{\mathsf{c}}_{\varepsilon}(\mathrm{co}(\mathcal{M}_{\mathcal{H},\mathbb{V}}))$ (defined in (50)) in terms of the DEC of $\mathrm{co}(\mathcal{H})$.

**Lemma H.7.** *For any $\varepsilon \geq 0$, it holds that*

$$\mathsf{r\text{-}dec}^{\mathsf{c}}_{\varepsilon}(\mathcal{M}_{\mathcal{H},\mathbb{V}}) \leq \mathsf{r\text{-}dec}^{\mathsf{c}}_{\sqrt{10}\varepsilon}(\mathcal{H})$$

We also note that $\mathrm{co}(\mathcal{M}_{\mathcal{H},\mathbb{V}}) \subseteq \mathcal{M}_{\mathrm{co}(\mathcal{H}),\mathbb{V}}$ because the model class $\mathcal{M}_{\mathrm{co}(\mathcal{H}),\mathbb{V}}$ is convex and it contains $\mathcal{M}_{\mathcal{H},\mathbb{V}}$. Therefore, we know

$$\mathsf{r\text{-}dec}^{\mathsf{c}}_{\varepsilon}(\mathrm{co}(\mathcal{M}_{\mathcal{H},\mathbb{V}})) \leq \mathsf{r\text{-}dec}^{\mathsf{c}}_{\varepsilon}(\mathcal{M}_{\mathrm{co}(\mathcal{H}),\mathbb{V}}) \leq \mathsf{r\text{-}dec}^{\mathsf{c}}_{\sqrt{10}\varepsilon}(\mathrm{co}(\mathcal{H})).$$

Using the regularity of $\varepsilon \mapsto \mathsf{r\text{-}dec}^{\mathsf{c}}_{\varepsilon}(\mathrm{co}(\mathcal{H}))$, we know

$$\overline{\mathsf{r\text{-}dec}}^{\mathsf{c}}_{\bar{\varepsilon}(T)}(\mathrm{co}(\mathcal{M}_{\mathcal{H},\mathbb{V}})) \leq c_{\mathrm{reg}} \cdot \mathsf{r\text{-}dec}^{\mathsf{c}}_{\sqrt{10}\varepsilon}(\mathrm{co}(\mathcal{H})).$$

This gives the desired upper bound. $\qquad\square$

### H.7.1 Proof of Lemma H.6

Fix a $\varepsilon \in [0,1]$, we denote $\varepsilon_1 = 2\sqrt{2}\varepsilon$ and take any $\Delta < \mathsf{r\text{-}dec}^{\mathsf{c}}_{\varepsilon_1}(\mathcal{H})$. We pick $\bar{h} \in \mathrm{co}(\mathcal{H})$ such that $\mathsf{r\text{-}dec}^{\mathsf{c}}_{\varepsilon_1}(\mathcal{H},\bar{h}) > \Delta$. Then, it holds that

$$\inf_{p\in\Delta(\Pi)} \sup_{h\in\mathcal{H}\cup\{\bar{h}\}} \left\{ \mathbb{E}_{\pi\sim p}[h(\pi_h) - h(a)] \,\middle|\, \mathbb{E}_{\pi\sim p}(h(a)-\bar{h}(a))^2 \leq \varepsilon_1^2 \right\} \geq \Delta.$$

Suppose that $\bar{h} \in \mathrm{co}(\mathcal{H})$ is given by $\bar{h} = \mathbb{E}_{h\sim\mu}[h]$ with $\mu \in \Delta(\mathcal{H})$. Then, consider the reference model $\overline{M} \in \mathcal{M}^+$ with mean reward function $\bar{h}$ and Gaussian noise, i.e. $\overline{M}(\pi) = \mathcal{N}(\bar{h}(\pi),1)$. Then, we know that for $\mathcal{M} = \mathcal{M}_{\mathcal{H}}$,

$$\begin{aligned}
&\mathsf{r\text{-}dec}^{\mathsf{c}}_{\varepsilon}(\mathcal{M}\cup\{\overline{M}\},\overline{M}) \\
&= \inf_{p\in\Delta(\Pi)} \sup_{M\in\mathcal{M}\cup\{\overline{M}\}} \left\{ \mathbb{E}_{\pi\sim p}[L(M,\pi)] \,\middle|\, \mathbb{E}_{\pi\sim p}D^2_{\mathrm{H}}\left(M(\pi),\overline{M}(\pi)\right) \leq \varepsilon^2 \right\} \\
&= \inf_{p\in\Delta(\Pi)} \sup_{h\in\mathcal{H}\cup\{\bar{h}\}} \left\{ \mathbb{E}_{\pi\sim p}[h(\pi_h) - h(\pi)] \,\middle|\, \mathbb{E}_{\pi\sim p}D^2_{\mathrm{H}}\left(\mathcal{N}(h(\pi),1),\mathcal{N}(\bar{h}(\pi),1)\right) \leq \varepsilon^2 \right\} \\
&\geq \inf_{p\in\Delta(\Pi)} \sup_{h\in\mathcal{H}\cup\{\bar{h}\}} \left\{ \mathbb{E}_{\pi\sim p}[h(\pi_h) - h(\pi)] \,\middle|\, \mathbb{E}_{\pi\sim p}(h(\pi)-\bar{h}(\pi))^2 \leq 8\varepsilon^2 \right\} \geq \Delta,
\end{aligned}$$

where the last line follows from Lemma C.5. Taking $\Delta \to \mathsf{r\text{-}dec}^{\mathsf{c}}_{\varepsilon_1}(\mathcal{H})$ completes the proof of (52). $\qquad\square$

### H.7.2 Proof of Lemma H.7

Fix a reference model $\overline{M} \in \mathrm{co}(\mathcal{M}_{\mathcal{H},\mathbb{V}})$. By definition, we know the mean reward function $h^{\overline{M}}$ of $\overline{M}$ belongs to $\mathrm{co}(\mathcal{H})$, i.e. $\overline{M} \in \mathcal{M}_{\mathrm{co}(\mathcal{H}),\mathbb{V}}$. Therefore, for any model $M \in \mathcal{M}_{\mathcal{H},\mathbb{V}}$ and decision $\pi \in \Pi$, by Lemma C.5,

$$D^2_{\mathrm{H}}\left(M(\pi),\overline{M}(\pi)\right) \geq \frac{1}{10}|h^M(\pi) - h^{\overline{M}}(\pi)|^2.$$

Therefore, for $\mathcal{M} = \mathcal{M}_{\mathcal{H},\mathbb{V}}$,

$$\begin{aligned}
&\mathsf{r\text{-}dec}^{\mathsf{c}}_{\varepsilon}(\mathcal{M}\cup\{\overline{M}\},\overline{M}) \\
&= \inf_{p\in\Delta(\Pi)} \sup_{M\in\mathcal{M}\cup\{\overline{M}\}} \left\{ \mathbb{E}_{\pi\sim p}[L(M,\pi)] \,\middle|\, \mathbb{E}_{\pi\sim p}D^2_{\mathrm{H}}\left(M(\pi),\overline{M}(\pi)\right) \leq \varepsilon^2 \right\}
\end{aligned}$$

$$\geq \inf_{p\in\Delta(\Pi)} \sup_{M\in\mathcal{M}\cup\{\bar{M}\}} \left\{ \mathbb{E}_{\pi\sim p}[L(M,\pi)] \mid \mathbb{E}_{\pi\sim p}|h^M(\pi) - h^{\bar{M}}(\pi)|^2 \leq 10\varepsilon^2 \right\}$$

$$= \inf_{p\in\Delta(\Pi)} \sup_{h\in\mathcal{H}\cup\{\bar{h}\}} \left\{ \mathbb{E}_{\pi\sim p}[h(\pi_h) - h(\pi)] \mid \mathbb{E}_{\pi\sim p}(h(\pi) - \bar{h}(\pi))^2 \leq 8\varepsilon^2 \right\}$$

$$= \mathsf{r\text{-}dec}^{\mathsf{c}}_{\sqrt{10}\varepsilon}(\mathcal{H}\cup\{\bar{h}\}, \bar{h}),$$

where the second equality follows from the fact that when $= h$, we have $L(M,\pi) = h(\pi_h) - h(\pi)$. Taking supremum over $\bar{M}$ completes the proof. $\qquad\square$

## H.8  Proof of Theorem G.6

Similar to Appendix H.7, we consider a larger model class $\mathcal{M}_{\mathcal{H},\mathbb{V}}$ of models with general noise structure. A model $M \in \mathcal{M}_{\mathcal{H},\mathbb{V}}$ is specified by a context distribution $\nu_M \in \Delta(\mathcal{C})$, a reward function $h^M \in \mathcal{H}$, and a reward distribution $\mathsf{R}^M(\cdot|\cdot,\cdot)$, such that for any $c \in \mathcal{C}, a \in \mathcal{A}, r \sim \mathsf{R}^M(\cdot|c,a)$ has mean $h^M(c,a)$ and variance at most 1. The model $M$ is then given by

$$(c,a,r) \sim M(\pi): \quad c \sim \nu_M, a = \pi(c), r \sim \mathsf{R}^M(\cdot|c,a).$$

The model class $\mathcal{M}_{\mathcal{H},\mathbb{V}}$ is defined to be the set of all possible models described above.

**Proof of Theorem G.6: lower bound.**  The lower bound with $\log \mathsf{N}_{\mathrm{frac}}(\mathcal{H}, \Delta)$ follows immediately by applying Theorem 10 to the class $\mathcal{M}_{\mathcal{H}}$, which admits $C_{\mathrm{KL}} = O(\log|\mathcal{C}|)$ in Assumption 2 (as shown in Example 9).

On the other hand, the lower bound with $T^{\mathsf{DEC}}(\mathcal{H}, \Delta)$ follows from the reduction to the *per-context* bandits problem. Specifically, for a fixed context $c \in \mathcal{C}$, $\mathcal{H}|_c$ corresponds to a structure bandits class $\mathcal{M}_{\mathcal{H}|_c}$. Notice that we can naturally regard $\mathcal{M}_{\mathcal{H}|_c} \subset \mathcal{M}_{\mathcal{H}}$ by viewing $\mathcal{M}_{\mathcal{H}|_c}$ as a contextual bandits class with the fixed context $c$. Therefore, by Theorem G.5 (specifically (53)):

$$\frac{1}{T}\mathbf{Reg}^\star(\mathcal{M}_{\mathcal{H}}, T) \geq \frac{1}{T}\mathbf{Reg}^\star(\mathcal{M}_{\mathcal{H}|_c}, T) \gtrsim \mathsf{r\text{-}dec}^{\mathsf{c}}_{\underline{\varepsilon}(T)}(\mathcal{H}|_c) - 6\underline{\varepsilon}(T), \qquad \underline{\varepsilon}(T) = \frac{1}{50\sqrt{T}}.$$

Taking maximum over $c \in \mathcal{C}$ yields

$$\frac{1}{T}\mathbf{Reg}^\star(\mathcal{M}_{\mathcal{H}}, T) \gtrsim \mathsf{r\text{-}dec}^{\mathsf{c}}_{\underline{\varepsilon}(T)}(\mathcal{H}) - 6\underline{\varepsilon}(T).$$

This gives the desired lower bound with $T^{\mathsf{DEC}}(\mathcal{H}, \Delta)$.

Combining both lower bounds completes the proof. $\qquad\square$

**Proof of Theorem G.6: upper bound.**  We follow the proof strategy of Appendix H.7. By Theorem H.3, $\mathsf{ExO}^+$ can be suitably instantiated on the problem class $\mathcal{M}_{\mathcal{H},\mathbb{V}}$ so that with probability at least $1 - \delta$:

$$\frac{1}{T}\mathbf{Reg}_{\mathsf{DM}} \leq \Delta + C \inf_{\gamma>0} \left( \mathsf{r\text{-}dec}^{\mathsf{o}}_{\gamma/8}(\mathrm{co}(\mathcal{M}_{\mathcal{H},\mathbb{V}})) + \gamma\frac{\log \mathsf{N}_{\mathrm{frac}}(\mathcal{M}, \Delta) + \log(1/\delta)}{T} \right).$$

We also note that $\mathrm{co}(\mathcal{M}_{\mathcal{H},\mathbb{V}}) \subseteq \mathcal{M}_{\mathrm{co}(\mathcal{H}),\mathbb{V}}$. Therefore, it remains to upper bound the offset DEC of $\mathcal{M}_{\mathrm{co}(\mathcal{H}),\mathbb{V}}$.

**Lemma H.8.** *For $\gamma > 0$, it holds that*

$$\mathsf{r\text{-}dec}^{\mathsf{o}}_{\gamma}(\mathcal{M}_{\mathcal{H},\mathbb{V}}) \leq \sup_{c\in\mathcal{C}} \mathsf{r\text{-}dec}^{\mathsf{o}}_{\gamma/2}(\mathcal{M}_{\mathcal{H}|_c,\mathbb{V}}).$$

Then, we can apply the result of Theorem H.5. From the proof of Theorem H.5, it is not hard to see that: for any $\varepsilon > 0$, there exists $\gamma = \gamma(\varepsilon)$ such that for any $c \in \mathcal{C}$,

$$\mathsf{r\text{-}dec}^{\mathsf{o}}_{\gamma/2}(\mathcal{M}_{\mathcal{H}|_c,\mathbb{V}}) + \gamma\varepsilon^2 \lesssim \sqrt{\log(2/\varepsilon)} \cdot (c_{\mathrm{reg}} \cdot \mathsf{r\text{-}dec}^{\mathsf{c}}_{\varepsilon}(\mathrm{co}(\mathcal{H})) + \varepsilon),$$

where we also use the regularity condition of $\varepsilon \mapsto \mathsf{r\text{-}dec}^{\mathsf{c}}_{\varepsilon}(\mathrm{co}(\mathcal{H}))$. This immediately gives

$$\mathbf{Reg}_{\mathsf{DM}} \leq T\Delta + \mathcal{O}(c_{\mathrm{reg}}T\sqrt{\log T}) \cdot \mathsf{r\text{-}dec}^{\mathsf{c}}_{\bar{\varepsilon}(T)}(\mathrm{co}(\mathcal{H})),$$

where $\bar{\varepsilon}(T) = \sqrt{\frac{\log \mathsf{N}_{\mathrm{frac}}(\mathcal{H},\Delta) + \log(1/\delta)}{T}}$. This is the desired upper bound. $\qquad\square$

### H.8.1 Proof of Lemma H.8

Fix a reference model $\overline{M} \in \text{co}(\mathcal{M}_{\mathcal{H},\mathbb{V}})$, and then $\overline{M} \in \mathcal{M}_{\text{co}(\mathcal{H}),\mathbb{V}}$ by definition. In particular, $\overline{M}$ has mean value function $h^{\overline{M}} \in \mathcal{H}$ and context distribution $\bar{\nu} \in \Delta(\mathcal{C})$. We also know that for each $c \in \mathcal{C}$, $h^{\overline{M}}(x, \cdot) \in \text{co}(\mathcal{H}|_c)$.

Then, by Lemma C.4, we also have

$$2D_{\text{H}}^2\left(M(\pi), \overline{M}(\pi)\right) \geq \mathbb{E}_{c \sim \nu_M, a = \pi(c)} D_{\text{H}}^2\left(\mathsf{R}^M(r = \cdot|c, a), \mathsf{R}^{\overline{M}}(r = \cdot|c, a)\right).$$

Thus, we adopt the following notations: For each $c \in \mathcal{C}$ and model $M \in \mathcal{M}_{\mathcal{H},\mathbb{V}}$, we define $M_c \in \mathcal{M}_{\mathcal{H}|_c,\mathbb{V}}$ to be a bandit model such that for every action $a \in \mathcal{A}$, $M_c(a) = \mathsf{R}^M(r = \cdot|c, a)$. Then by definition, it holds that

$$2D_{\text{H}}^2\left(M(\pi), \overline{M}(\pi)\right) \geq \mathbb{E}_{c \sim \nu_M, a = \pi(c)} D_{\text{H}}^2\left(M_c(a), \overline{M}_c(a)\right).$$

Now, combining the inequalities above, we have

$$\mathsf{r\text{-}dec}_{\gamma}^{\text{o}}(\mathcal{M}_{\mathcal{H},\mathbb{V}}, \overline{M})$$

$$= \inf_{p \in \Delta(\Pi)} \sup_{M \in \mathcal{M}_{\mathcal{H},\mathbb{V}}} \mathbb{E}_{\pi \sim p}[L(M, \pi)] - \gamma \mathbb{E}_{\pi \sim p} D_{\text{H}}^2\left(M(\pi), \overline{M}(\pi)\right)$$

$$\leq \inf_{p \in \Delta(\Pi)} \sup_{M \in \mathcal{M}_{\mathcal{H},\mathbb{V}}} \mathbb{E}_{\pi \sim p}\mathbb{E}_{c \sim \nu_M, a = \pi(c)}\left[h^M(c, \pi_M(c)) - h^M(c, a) - \frac{\gamma}{2}D_{\text{H}}^2\left(M_c(a), \overline{M}_c(a)\right)\right]$$

$$\overset{(1)}{=} \inf_{p = (p_c), p_c \in \Delta(\mathcal{A})} \sup_{M \in \mathcal{M}_{\mathcal{H},\mathbb{V}}} \mathbb{E}_{c \sim \nu_M, a \sim p_c}\left[h^M(c, \pi_M(c)) - h^M(c, a) - \frac{\gamma}{2}D_{\text{H}}^2\left(M_c(a), \overline{M}_c(a)\right)\right]$$

$$\overset{(2)}{\leq} \inf_{p = (p_c), p_c \in \Delta(\mathcal{A})} \sup_{M \in \mathcal{M}_{\mathcal{H},\mathbb{V}}} \sup_{c \in \mathcal{C}} \mathbb{E}_{a \sim p_c}\left[h^M(c, \pi_M(c)) - h^M(c, a) - \frac{\gamma}{2}D_{\text{H}}^2\left(M_c(a), \overline{M}_c(a)\right)\right]$$

$$\overset{(3)}{=} \inf_{p = (p_c), p_c \in \Delta(\mathcal{A})} \sup_{c \in \mathcal{C}} \sup_{M_c \in \mathcal{M}_{\mathcal{H}|_c,\mathbb{V}}} \mathbb{E}_{a \sim p_c}\left[h^{M_c}(\pi_{M_c}) - h^{M_c}(a) - \frac{\gamma}{2}D_{\text{H}}^2\left(M_c(a), \overline{M}_c(a)\right)\right]$$

$$\overset{(4)}{=} \sup_{c \in \mathcal{C}} \inf_{p_c \in \Delta(\mathcal{A})} \sup_{M_c \in \mathcal{M}_{\mathcal{H}|_c,\mathbb{V}}} \mathbb{E}_{a \sim p_c}\left[h^{M_c}(\pi_{M_c}) - h^{M_c}(a) - \frac{\gamma}{2}D_{\text{H}}^2\left(M_c(a), \overline{M}_c(a)\right)\right]$$

$$= \sup_{c \in \mathcal{C}} \mathsf{r\text{-}dec}_{\gamma/2}^{\text{o}}(\mathcal{M}_{\mathcal{H}|_c,\mathbb{V}}, \overline{M}_c) \leq \sup_{c \in \mathcal{C}} \mathsf{r\text{-}dec}_{\gamma/2}^{\text{o}}(\mathcal{M}_{\mathcal{H}|_c,\mathbb{V}}),$$

where the equality (1) is because for a sequence $(p_c \in \Delta(\mathcal{A}))_{c \in \mathcal{C}}$, there is a corresponding $p \in \Delta(\Pi)$ such that for $\pi \sim p$, we have $\pi(c) \sim p_c$ independently; in inequality (2) we bound the expectation over $c \sim \nu_M$ by the supremum $\sup_{c \in \mathcal{C}}$; the equality (3) follows from the fact that $M_c \in \mathcal{M}_{\mathcal{H}|_c,\mathbb{V}}$ is a bandit model with mean reward function $h^{M_c}(\cdot) = h^M(c, \cdot)$; and the equality (4) is because we can choose $p_c$ separately for every $c \in \mathcal{C}$. By the arbitrariness of $\overline{M} \in \text{co}(\mathcal{M})$, we now have

$$\mathsf{r\text{-}dec}_{\gamma}^{\text{o}}(\mathcal{M}_{\mathcal{H},\mathbb{V}}) \leq \sup_{c \in \mathcal{C}} \text{dec}_{\gamma/2}^{\text{o}}(\mathcal{M}_{\mathcal{H}|_c,\mathbb{V}}).$$

$\square$

### H.9 Proof of Corollary G.7

We follow the notations of Appendix H.8. By Lemma H.8, we have

$$\mathsf{r\text{-}dec}_{\gamma}^{\text{o}}(\mathcal{M}_{\mathcal{H},\mathbb{V}}) \leq \sup_{c \in \mathcal{C}} \mathsf{r\text{-}dec}_{\gamma/2}^{\text{o}}(\mathcal{M}_{\mathcal{H}|_c,\mathbb{V}}).$$

Notice that for each $c \in \mathcal{C}$, $\mathcal{M}_{\mathcal{H}|_c,\mathbb{V}}$ is a class of $|\mathcal{A}|$-arm bandits, and hence by Foster et al. [40, Proposition 5.1] and Lemma C.5, we have

$$\mathsf{r\text{-}dec}_{\gamma}^{\text{o}}(\mathcal{M}_{\mathcal{H}|_c,\mathbb{V}}) \leq \frac{8|\mathcal{A}|}{\gamma}.$$

Therefore, Theorem H.3 implies that $\mathsf{ExO}^+$ achieves with probability at least $1 - \delta$:

$$\frac{1}{T}\mathbf{Reg}_{\text{DM}} \leq \Delta + \frac{16|\mathcal{A}|}{\gamma} + \gamma\frac{\log \mathsf{N}_{\text{frac}}(\mathcal{H}, \Delta) + \log(1/\delta)}{T}.$$

Balancing $\gamma > 0$ gives the desired upper bound. $\square$

As a remark, we provide an example of function class $\mathcal{H}$ with $\log \mathsf{N}_{\text{frac}}(\mathcal{H}, \Delta) \ll \log |\mathcal{H}|$.

**Example 14.** Suppose that $\mathcal{A} = \{0, 1\}$, and the function class $\mathcal{H} = \{h_x\}_{x \in \mathcal{C}}$, where

$$h_x(c, 0) = \frac{1}{2}, \qquad h_x(c, 1) = \begin{cases} 1, & c = x, \\ 0, & c \neq x. \end{cases}$$

Clearly, we have $\log |\mathcal{H}| = \log |\mathcal{C}|$.

On the other hand, we consider a distribution $p$ over policies, such that $\pi \sim p$ is generated as $\pi(c) \sim \mathrm{Bern}(\varepsilon)$, independently over all $c \sim \mathcal{C}$. Then, for any $h = h_x \in \mathcal{H}$ and $\nu \in \Delta(\mathcal{C})$, we have

$$\mathbb{E}_{c \sim \nu}[h(c, \pi_h(c)) - h(c, \pi(c))] = \nu(x) \cdot \frac{1}{2}\mathbf{1}\{\pi(x) = 1\} + \frac{1}{2}\mathbb{E}_{c \sim \nu}[\mathbf{1}\{c \neq x, \pi(c) = 1\}].$$

Notice that $\pi(x) = 1$ with probability $\Delta$, and conditional on the event $\{\pi(x) = 1\}$,

$$\mathbb{E}_{\pi \sim p}[\mathbb{E}_{c \sim \nu}[\mathbf{1}\{c \neq x, \pi(c) = 1\}]|\pi(x) = 1] \leq \Delta.$$

Hence,

$$p(\pi : \mathbb{E}_{c \sim \nu}[h(c, \pi_h(c)) - h(c, \pi(c))] \leq \Delta) \geq \frac{\Delta}{2},$$

which implies $\log \mathsf{N}_{\mathrm{frac}}(\mathcal{H}, \Delta) \leq \log(2/\Delta)$.

Therefore, for unbounded context space $\mathcal{C}$, we have $\log \mathsf{N}_{\mathrm{frac}}(\mathcal{H}, \Delta) \ll \log |\mathcal{H}|$ for the function class $\mathcal{H}$ defined above.

