# OpenReview forum: "Assouad, Fano, and Le Cam with Interaction: A Unifying Lower Bound Framework and Characterization for Bandit Learnability"
_NeurIPS.cc/2024/Conference — NeurIPS 2024 spotlight_

### Official Review · Reviewer_zJ4c · 2024-06-26

**Soundness:** 2
**Presentation:** 3
**Contribution:** 3
**Rating:** 6
**Confidence:** 4

**Summary:**

This paper aims to provide a unified framework for deriving lower bounds of interactive decision-making problems using an extended technique adapted from Chen et al., 2016 [11], which was established only for non-interactive estimation problems. The paper offers a quite general formulation of the minimax value of interactive decision-making that encompasses both previously known and novel formulations. In particular, the paper demonstrates that such a lower bounding technique can be used to rederive the lower bounds from Foster et al., 2023 [25] and provide tighter lower and upper bounds via a new complexity named "decision coverage."

**Strengths:**

I find the general formulation and the technique based on percentiles (instead of the expected value used in prior literature such as [25]) for deriving lower bounds to be quite appealing. Although it appears to me that the lower bounding techniques are essentially "abstracted out" from prior literature, such as [11] and [23, 25], the paper presents it in a quite clean manner. The paper also provides several new results, such as interactive parameter estimation and tighter bounds for interactive reward maximization. I generally enjoyed reading this paper and find it worthy of publishing at NeurIPS. I expect the paper to have a broad impact on the community.

**Weaknesses:**

My main complaint about this paper is that it contains many hand-waving arguments. I had to fill in many gaps in the proofs myself to verify and understand the results.

I outline some specific comments below (some are minor, while others need additional justifications):

1. I'm not sure if it is appropriate to call Corollary 2 a "Generalized Fano’s inequality", as it is not clear to me that it implies the original Fano's inequality (though they are "qualitatively" equivalent, i.e., both require $I(M,X) \ll \log |\mathcal{M}|$ to arrive at a constant lower bound).
2. You claim in line 186 "Fano’s inequality, e.g., in the form of Corollary 2, cannot be used to prove Lemma 3." Is there an argument why this is true?
3. Can you explain how Theorem 5 is instantiated from Theorem 1? It appears to me that its proof is completely self-contained without invoking any parts of Theorem 1.
4. Can you comment on how prior technique for deriving Corollary 9 differs from your Fano's inequality-based argument?
5. Can you explain how $T^*(\mathcal{M},\Delta)$ characterizes the regret? Specifically, it does not appear to me that there is a simple way to express the regret in terms of $T^*(\mathcal{M},\Delta)$. Moreover, why does Theorem 4 imply the statement in line 282?
6. The proof of Theorem 10 is too sketchy. It is not clear to me that (12) shares the structure as Proposition 8. Are you applying the minimax theorem here?
7. Theorem 10 looks quite trivial to me, as it does not depend on any information-theoretical structure of the model class (except Assumption 2).
8. Given point 5, I'm not sure how to interpret your Theorem 12 and how to compare it with [25]. It appears you are claiming this as a main result; can you provide any specific examples that demonstrate the improvement (i.e., explicitly computing the "decision coverage" for some classes)?
9. You claim that "Theorem 12 provides (polynomially) matching lower and upper bounds for learning M." Isn't $\max\\{a,b\\}$ can be arbitrarily smaller than $a \cdot b$? How should the "matching" be interpreted?
10. It appears to me the upper bound of Theorem 11 essentially converts the dependency on the size of the model class to that of the decision space. Given the similarity of the proof with that in [24], can you comment on if the results from [24] recover your upper bound (with possible replacement of decision coverage with $\log|\Pi|$)?
11. Typos (there are many, but I only include what I still remember):
    - $D_f(a,b)$ was never properly defined, though one can guess it is for $D_f(\text{Bern}(a),\text{Bern}(b))$.
    - What is the $\alpha$ in line 873?

**Questions:**

See above.

---

> ### Author Rebuttal · Authors · 2024-08-06
>
> Thank you for your insightful feedback and for dedicating time to review our paper!
> > Q1. I'm not sure if it's appropriate to call Corollary 2 "Generalized Fano’s inequality"
>
> In our proof of Theorem 1 on page 15, we actually prove Theorem D.1, which holds for general quantile $\delta\in(0,1)$ rather than $½$. Similarily, Corollary 2 can be generalized to any $\delta$: when $I(M;X)+\log 2\leq (1-\delta)\cdot \log \frac{1}{\mu(M \in \mathcal{M} : L(M,x) < \Delta)}$, the Bayes risk $\geq\delta\Delta$. This precisely implies original Fano’s inequality when $\mu$ is uniform, $\mathcal{M}=\mathcal{X}, L(M,x)=\mathbb{1}(M\neq x), \Delta=1$. We will restate Corollary 2 for arbitrary $\delta$ and clearly explain the generalization.
> > Q2. You claim in line 186 "Fano’s inequality, e.g., in the form of Corollary 2, cannot be used to prove Lemma 3."
>
> Fano’s method and the “mixture vs mixture” method described in Lemma 3 are conceptually distinct due to the use of different divergences. In particular, Lemma 3 (and the mixture vs mixture method in general) use total variation distance, whereas Fano inequality uses KL-based mutual information. As a result, unless the distributions under consideration have favorable structure that allow one to bound the KL-based mutual information by total variation distance, it does not appear to be possible to recover Lemma 3 from Corollary 2.
>
> We mention in passing that we are not aware of any results which recover guarantees based on the mixture vs. mixture method from Fano method in prior work.
> > Q3. How Theorem 5 is instantiated from Theorem 1?
>
> The current proof of Theorem 5 is indeed self-contained (for completeness), but the idea is the same as Theorem 1, and it can indeed be proven as a corollary. To instantiate Theorem 1, we can choose the $f$-divergence to be the squared Hellinger distance, set $\mathbb{Q}=\mathbb{P}^{\overline{M},ALG}$, and choose $\mu$ to be the delta distribution on the model $M$. We will clarify this in the revision.
> > Q4. How prior technique for deriving Corollary 9 differs from your Fano's inequality-based argument?
>
> Prior work proves the regret lower bound in Corollary 9 using Assouad’s lemma (see e.g. Section 24 of [34]), which explicitly relies on the hypercube structure of the parameter space. By contrast, the proof of Corollary 9 is based on bounding the mutual information under a certain (uniform) prior and then applying Fano’s inequality. Conceptually, these two approaches appear to be fairly distinct (Assouad’s lemma often gives tight results when hypercube structure is available, while the Fano method is somewhat more general, but can require more effort to apply, particularly in interactive settings; therefore, it is interesting to see that both lead to the same result here).
> > Q5. How T∗(M,Δ) characterizes the regret? Why does Theorem 4 imply line 282?
>
> By the definition of $T^\star(\mathcal{M},\Delta)$, we have
> $$
> \frac{1}{T}\mathrm{Reg}^\star_T=\sup\{ \Delta: T^\star(\mathcal{M},\Delta)\leq T \}.
> $$
> For example, $T^\star(\mathcal{M},\Delta) \asymp \frac{C}{\Delta^2}$ implies that $\mathrm{Reg}^\star_T \asymp \sqrt{CT}$.
>
> Further, notice that Theorem 4 implies (under Assumption 3)
> $$
> \mathrm{dec}^c_{\underline{\epsilon}(T)}(\mathcal{M}) \lesssim\frac{1}{T}\mathrm{Reg}^\star_T\lesssim \mathrm{dec}^c_{\bar{\epsilon}(T)}(\mathcal{M})
> $$
>
> The statement in line 282 then follows from the definition of $T^\star(\mathcal{M},\Delta)$ and $T^{\rm DEC}(\mathcal{M},\Delta)$.
> > Q6. The proof of Theorem 10 is too sketchy. Are you applying the minimax theorem here?
>
> Thank you for pointing this out. The current presentation of the proof focuses on highlighting the connection to Proposition 8, which is somewhat subtle and requires applying the minimax theorem. We will be sure to include more details in the final revision (there is also a more direct proof, which we are happy to include for completeness).
> > Q7. Theorem 10 looks quite trivial...
>
> Indeed, the quantity DC measures certain inherent complexity of the decision space, and it does not depend on the information-theoretical structure of the model class. However, it does imply non-trivial lower bounds for various model classes:
>
> (1) Linear bandits: $\mathsf{DC}\gtrsim d$
>
> (2) Unstructured contextual bandits with context space $\mathcal{C}$: $\mathsf{DC}\gtrsim |\mathcal{C}|\log|\mathcal{A}|$
>
> (3) Tabular RL: $\mathsf{DC}\gtrsim |\mathcal{S}|$.
>
> Given that the proof of Theorem 10 only utilizes the structure of the decision space, DC should be regarded as a (non-trivial) complexity measure that is complementary to DEC (which captures the information-theoretical structure of the model class), in the sense that the lower bounds they provide are complementary, and together they provide also an upper bound (Theorem 11).
> > Q8. How to interpret your Theorem 12? Any specific examples?
> > Q9. Isn't max{a,b} can be arbitrarily smaller than a⋅b? How should the "matching" be interpreted?
>
> The main contribution of Theorem 12 is that the upper bound is at most the square of the lower bound (as $a\cdot b\leq \max(a,b)^2$). Therefore, for **any** convex model class, the sample complexity is completely determined by DEC and DC up to a square gap. Such a characterization of sample complexity for general decision making is new, and we believe it to be conceptually important.
> > Q10. Can you comment on if the results from [24] recover your upper bound (with possible replacement of decision coverage with log⁡|Π|)?
>
> Our proof of Theorem 11 is essentially an adaptation [24], with the goal of replacing $\log|\Pi|$ by the decision coverage in the upper bound. For general decision making problems, $\log|\Pi|$ can be arbitrarily larger than the decision coverage, meaning that our upper bound represents a significant improvement.
> > Typos
>
> Thank you for pointing out these typos. Indeed, $D_f(a,b)$ is the abbreviation of $D_f(Bern(a),Bern(b))$, and $\alpha$ in line 873 is a typo. We will clarify these in the revision.

---

> > ### Comment · Reviewer_zJ4c · 2024-08-07
> >
> > Thank you for the response. I maintain my current rating, favoring the acceptance of the paper.

---

### Official Review · Reviewer_uqGG · 2024-07-11

**Soundness:** 3
**Presentation:** 3
**Contribution:** 3
**Rating:** 6
**Confidence:** 2

**Summary:**

This paper develops the notion of Interactive Statistical Decision Making (ISDM), and a generic lower bound (Theorem 1) which can be instantiated to capture the standard Le Cam, Assaoud, Fano methods as well as recent lower bound results in interactive decision making. The authors further use Theorem 1 to derive new sample complexity bounds (Theorem 12, 13) on interactive decision making contexts (under some regularity conditions on the model class).

**Strengths:**

I think it's interesting to try to unify different lower bound approaches in order to gain new insights, so the premise of the paper is very intriguing to me. The submission further uses the general theorem to derive new bounds, based on the new notion of Decision Coverage, that tightens existing results in interaction decision making.

**Weaknesses:**

It may be due to my lack of expertise in interaction decision making literature, but I'm having trouble understanding and evaluating the new contributions, with intuition missing that I hoped the paper would give. Some of the contributions also seem a bit overclaimed. I hope the following constructive criticisms would help the authors improve the paper.

- "Addressing remaining gap" and "complete characterization": the authors say that the new results (Theorems 12/13) "completely characterize" the sample complexity of interactive decision making for convex model classes, but the left and right hand sides differ quadratically (and ignores log factors). Am I misinterpreting the results? They aren't even tight up to constants.

- Line 56: I don't quite understand why "unifying two-point vs mixture-vs-mixture methods" is a new contribution. Mixture-vs-mixture is clearly a generalization of singleton two-point methods, so why is the unification sold as a new contribution?

- Generally I find the bound in Theorem 1 challenging to interpret, as opposed to two-point or mixture-vs-mixture methods that make intuitive sense. I'm struggling to understand the insight gained by the Theorem 1 formulation through unifying two-point methods and Fano with interaction decision making. To me, ISDM reads like a very generic minimax game formulation, and then Theorem 1 tags on the reference distribution/ghost data $\mathbb{Q}$ in order to encompass existing techniques for interaction decision making. Lemma 3 then removes this extra component by declaring $\mathbb{Q}$ simply as the transcript of the algorithm, so what have we learned about standard statistical estimation through Theorem 1? In other words, my question is, why is Le Cam/Assaoud/Fano even part of the paper, instead of focusing only on the new bounds in interactive decision making? What am I missing?

- Theorem 12 is a bit too informal, hiding the log factors (especially without specifying log factors in what). It also wasn't actually proven -- I think the calculations using Assumptions 2 and 3 (and applied to Theorem 11) should be explicitly shown in the appendix.

- Relatedly, again on the topic of needing more interpretation, I wish the submission explained how DC is better than the $\log |\mathcal{M}|$ factor in Line 282.

Misc typos I spotted and other small comments:

- First page, should define Perf as cost, so that minimization is the correct direction.
- End of Line 123, should the asterisk in $M^\ast$ be removed?
- Line 177, $L:\Theta \times \mathcal{A} \rightarrow \mathbb{R}_+$ right? Instead of domain being $\Theta \times \Theta$? Just a consistency issue with the rest of the lemma.
- Line 594, I presume "ISDM" instead of "ASDM"?

**Questions:**

Please see weaknesses above.

- Could the authors also comment on how the techniques apply/adapt if we want to show lower bounds on high-probability loss instead of expected loss? Two-point methods are straightforward (and even simpler than Le Cam) to use in a high-probability setting.

**Limitations:**

Yes.

---

> ### Author Rebuttal · Authors · 2024-08-06
>
> Thank you for your careful review and constructive criticism! We will work on better presenting the intuition behind our results and revise our statements to make them more precise.
>
> > Regarding "addressing remaining gap" and "complete characterization": the lower and upper bounds differ quadratically ...They aren't even tight up to constants.
>
> Indeed, by using the term "complete characterization", we want to highlight that DC and DEC together provide a characterization of *polynomial learnability*. Although their lower and upper bounds may not offer a tight characterization of the T-round minimax risk, our one-round complexity measures, DC and DEC, elucidate what is necessary and sufficient for the estimation component (DC) and the estimation-to-decision-making component (DEC) in interactive decision making.
>
> DC is essential because it appears in both the upper and lower bounds (similar to DEC). In contrast, prior work [23][25] includes a $\log|\mathcal{M}|$ factor only in the upper bound, with no hope of it appearing in a lower bound. This discrepancy was identified as a major open question in previous studies [23][25]. To address this gap, we need to eliminate $\log|\mathcal{M}|$ and identify DC that appears in both upper and lower bounds.
>
> > Line 56: I don't quite understand why "unifying two-point vs mixture-vs-mixture methods" is a new contribution. Mixture-vs-mixture is clearly a generalization of singleton two-point methods, so why is the unification sold as a new contribution?
>
> Sorry for the confusion. Mixture-vs-mixture is indeed a generalization of the singleton two-point method, and we mean to say that our approach recovers both, rather than unifying them into one concept. Our intention is to convey that our method unifies mixture-vs-mixture (and thus the entire scope of two-point methods) with Fano's and Assoud's methods, where these three previously lacked a unified perspective. We will rewrite the sentence as “unify mixture-vs-mixture (and thus the entire scope of two point methods) as a special case of our general algorithmic lower bound” to convey the right message.
>
> > Generally I find the bound in Theorem 1 challenging to interpret, ...why is Le Cam/Assaoud/Fano even part of the paper, instead of focusing only on the new bounds in interactive decision making?
>
> Our goal is to integrate the Fano and Assouad methods (which provide dimensional insights but are typically challenging to apply in interactive settings and inherently follow a T-round analysis approach) with the DEC framework (which offers one-round complexity measures for interactive settings but previously lacked a dimensional factor/estimation component in the lower bound). Therefore, it is crucial to demonstrate that our framework can recover the non-interactive versions of these methods as a special case, serving as an important sanity check.
>
> Moreover, by utilizing the algorithmic Fano method (Proposition 8), we not only introduce the new complexity measure DC (Theorem 12), which includes the dimension in the lower bound, but also recover a tight lower bound for linear bandits (Corollary 9), which would otherwise not be captured by Theorem 12. This also illustrates the advantage of having a unified method for interactive decision making and Fano’s method.
>
> > Theorem 12 is a bit too informal, hiding the log factors (especially without specifying log factors in what). It also wasn't actually proven -- I think the calculations using Assumptions 2 and 3 (and applied to Theorem 11) should be explicitly shown in the appendix.
>
> Thanks! We will provide a detailed step-by-step explanation in the revised version. The minimax-optimal sample complexity $T^\star(\mathcal{M},\Delta)$ is just a way to better illustrate our minimax regret upper and lower bounds.
> Assumptions 2 and 3 are conditions to establish that the additional terms in the regret upper (Theorem 11) and lower bound (Theorem E.1) are of the same order as the dec terms. This ensures that one has
> $$
> \mathrm{dec}^c_{\underline{\epsilon}(T)}(\mathcal{M}) \lesssim\frac{1}{T}\mathrm{Reg}^\star_T\lesssim \mathrm{dec}^c_{\bar{\epsilon}(T)}(\mathcal{M})
> $$
> Theorem 12 is then proved by using the definitions of sample complexities $T^\star(\mathcal{M},\Delta)$ and $T^{\rm DEC}(\mathcal{M},\Delta)$.
>
> > Relatedly, again on the topic of needing more interpretation, I wish the submission explained how DC is better than the factor $\log|\mathcal{M}|$ in Line 282.
>
> DC is fundamentally different with $\log|\mathcal{M}|$ because it arise in both the upper and the lower bounds, while $\log|\mathcal{M}|$ only appears in the upper bound (and there is no hope for having it appear in a lower bound in general). Comparing Theorem 12 and Line 282, one can observe that DC together with DEC gives a complete characterization of the sample complexity up to a quadratic order. In contrast, $\log|\mathcal{M}|$ is clearly not a good candidate for the estimation component in the lower bound, as it is the coarsest measure in learning theory.
>
> Moreover, for examples like convex bandits, the $\log|\mathcal{M}|$ factor is obviously too loose, which will be $\exp(d)$ for the complete class of convex functions, while we know that $poly(d)$ regret is achievable for convex bandits [23][32].
>
> > Could the authors also comment on how the techniques apply/adapt if we want to show lower bounds on high-probability loss instead of expected loss? ...
>
> This is a great question! In our proof of Theorem 1, we essentially establish a lower bound for the quantile $\mathbb{P}_{M∼\mu, X\sim{\mathbb{P}^{M,ALG}}} (L(M, X) \geq \Delta)$ (line 578). This quantile lower bound (Theorem D.1) can be directly used to prove a high-probability lower bound in a straightforward manner (for any error $\delta$).
>
> > Misc typos I spotted and other small comments...
>
> Many thanks for pointing these out. We will be sure to clarify these issues in the final revision.

---

> > ### Comment · Reviewer_uqGG · 2024-08-07
> >
> > Thank you for your response. I have some more comments.
> >
> > **"Complete characterization"**: please rephrase it in the final paper.
> >
> > **Fano/Assouad**:
> > >Our goal is to integrate the Fano and Assouad methods (which provide dimensional insights but are typically challenging to apply in interactive settings and inherently follow a T-round analysis approach) with the DEC framework (which offers one-round complexity measures for interactive settings but previously lacked a dimensional factor/estimation component in the lower bound). Therefore, it is crucial to demonstrate that our framework can recover the non-interactive versions of these methods as a special case, serving as an important sanity check.
> >
> > I actually think this is a much more understandable and convincing message than the current version of the abstract and intro. "Goal is integrating Fano/Assouad, and we do a sanity check" does get around my concern that the ISDM problem and some lemmas looked artificial.
> >
> > **DC vs $\log|\mathcal{M}|$**: I was hoping for more quantitative intuition, actually comparing the two quantities. I understand that the difference is pretty analogous, in the qualitative sense, to something like VC dimension vs $\log |\mathcal{C}|$ in standard non-interactive PAC learning. I think a more quantitative discussion would strengthen the paper. But even absent that, emphasizing the qualitative difference more clearly would be useful.
> >
> > **High probability**: Thanks for confirming my guess. I wasn't sure about the quantile, since I didn't have time to read the proofs thoroughly. I think this would be a good remark to put in the paper.

---

> > > ### Author Response · Authors · 2024-08-09
> > >
> > > Thank you for the comments! In the revision, we will add a more detailed discussion regarding the "characterization," high probability version of the lower bounds, integrating Fano, and DC vs $\log|\mathcal{M}|$.
> > >
> > > >DC vs $\log|\mathcal{M}|$: I was hoping for more quantitative intuition, actually comparing the two quantities.
> > >
> > > Regarding "DC vs $\log |\mathcal{M}|$": Quantitatively, we always have $\mathsf{DC}_\Delta(\mathcal{M})\leq \log|\mathcal{M}|$.
> > >
> > > Specifically, $p_\Delta^\star=\sup_p\inf_M p(\pi:g^M(\pi)\leq \Delta)$ is lower bounded by $1/|\mathcal{M}|$ when $p$ is chosen as the induced distribution of $\pi_M$ for $M\sim \textup{uniform}(|\mathcal{M}|)$. Thus, $ DC_\Delta(\mathcal{M})= \log(1/p_\Delta^\star) \leq \log|\mathcal{M}|$.
> > >
> > > From this perspective, their relation is indeed analogous to VC dimension vs $\log|\mathcal{C}|$ for PAC learning. We will discuss this more thoroughly in the revision.

---

### Official Review · Reviewer_Fyj4 · 2024-07-13

**Soundness:** 3
**Presentation:** 3
**Contribution:** 4
**Rating:** 7
**Confidence:** 3

**Summary:**

This paper proposes a unified framework for lower-bound methods in statistical estimation and interactive decision-making. The authors integrate classical lower bound techniques (Fano's inequality, Le Cam's method, Assouad's lemma) with recent minimax lower bounds for interactive decision-making (Decision-Estimation Coefficient). The framework is based on a general algorithmic lower bound method and introduces a novel complexity measure, decision coverage. The paper also has a lot of other results, including the unification of classical and interactive methods, the generalization of Fano's separation condition, and the derivation of new lower bounds for interactive decision-making.

**Strengths:**

* The framework developed in the paper is valid for a very general class of problems and unifies classical and classical lower bound techniques, Fano's inequality, Le Cam's method, and Assouad's lemma, and provides a comprehensive framework for statistical estimation and decision making.
* Lots of other contributions: incorporating the previous work on DEC into the framework of this paper introduces a new complexity measure called decision coverage, which quantifies the  complements of the previous DEC lower bounds.
* Except for a few typos, the paper is well-written.
* Overall, it is a solid paper, and contributions are significant.

**Weaknesses:**

* I dont find any major technical weakness in this work. The work is good and a bit hard to follow for someone who is a non-expert in this area. Most of the points I highlight below are bunch of typos that I found and clarification that I think its good to include.
* Line 123, I think it should be $L(M,X)$ instead of $L(M^*,X)$
* The proof of corollary 2 is skipped. It involves 3-4 steps, and therefore I don't think it is trivial, especially for those who are not experts in dealing with these inequalities.

* There seem to be a lot of (minor) typos in Lemma 3, and it's proof.
1. Firstly, it's better to clarify whether sets $\theta_0, \theta_1$ are required to be distinct or not in the Lemma statement.
2. ASDM is not defined in the first line of proof.
3. I think the second equality in Line 602 will contain a factor of $1/2$. Can authors also clarify if the next step follows by data processing?
4. In line 603, $ d_{3/4}(\cdot,\cdot)$ is not defined. May be authors mean $ d_{f, 3/4}(\cdot,\cdot)$. Please also explain why that inequality follows, is it by choice of $\Delta$ and so that Theorem 1 can be applied?
5. There seems to be a typo in the subscripts of expectation in line 604, third inequality.

* I guess in Line 218 Eq. 9, $p_{out}$ is not defined.
* I don't think $\pi_{out}$  in Line 226 is defined before (in the statement of theorem 5).
* I would appreciate if the authors mention where the realizability assumption (Assumption 1) is needed and the issues that arise in the agnostic setting, i.e., when realizability does not hold. Also, please provide examples (or references) of well-posed model classes in Assumption 2.
* It is not clear to me why there exists $M$ in the model class, i.e., why is the supremum achieved here by some $M$ in proof of Theorem 5.
* Line 329, I don't think $\mathbb{V}$ is defined.

**Questions:**

See weaknesses.

**Limitations:**

Essentially no broader societal impacts.

---

> ### Author Rebuttal · Authors · 2024-08-06
>
> Thank you for your insightful feedback and for dedicating time to review our paper! We respond to the specific questions as follows:
>
> > 1. The proof of corollary 2 is skipped. It involves 3-4 steps, and therefore I don't think it is trivial, especially for those who are not experts in dealing with these inequalities.
>
> Thank you for the feedback. We will be sure to include a detailed proof for Corollary 2 for readers less familiar with these sorts of manipulations.
>
> > 2.  There seem to be a lot of (minor) typos in Lemma 3, and it's proof.
>
> >> Firstly, it's better to clarify whether sets $\theta_0,\theta_1$ are required to be distinct or not in the Lemma statement.
>
> In the most general case, $\Theta_0$ and $\Theta_1$ do not have to be distinct. It could be implied to be distinct if the loss function $L$ satisfies $L(\theta,\theta) = 0$ for all $\theta\in \Theta$.
>
> >> ASDM is not defined in the first line of proof.
> We apologize for the typo. This line is meant to say “ISDM”.
>
> >> I think the second equality in Line 602 will contain a factor of 1/2. Can authors also clarify if the next step follows by data processing?
>
> Yes, it will contain a factor of 1/2, and it is an inequality due to the convexity of TV distance. And yes, the next step follows by data-processing inequality.
>
> >> In line 603, $d_{3/4}(\cdot,\cdot)$ is not defined. May be authors mean $d_{d,3/4}(\cdot,\cdot)$. Please also explain why that inequality follows, is it by choice of $\Delta$ and so that Theorem 1 can be applied?
>
> In line 574, we defined $d_{f,\delta}$. In line 603, we mean $d_{|\cdot|, 1/4}$ where $f(x)=|x|$, which yields the TV distance. We then apply Theorem D.1 with $f(x)=|x|$ and $\delta=1/4$.
>
> >> There seems to be a typo in the subscripts of expectation in line 604, third inequality.
>
> We apologize for the typo. This should be $\mathbb{E}_{M\sim\mu}\mathbb{E}
> _{X\sim {\mathbb{P}^{M,\texttt{ALG}}}}[\ell(M,X)]$.
>
> > 3. I guess in Line 218 Eq. 9, $p_{out}$ is not defined. ...I don't think $\pi_{out}$ in Line 226 is defined before (in the statement of theorem 5).
>
> The output decision $\pi_{\text{out}}$ is the decision $\hat{\pi}$ in line 104, and the output policy $p_{\text{out}}$ is the distribution over the output decision $\pi_{\text{out}}$. We will make sure to add these explanations.
>
> > 4. I would appreciate if the authors mention where the realizability assumption (Assumption 1) is needed and the issues that arise in the agnostic setting, i.e., when realizability does not hold. Also, please provide examples (or references) of well-posed model classes in Assumption 2.
>
>
> Without the realizability assumption, the lower bound still holds since the model can be arbitrary. The realizability assumption is required for the upper bound, e.g., the last inequality of line 835 in Theorem F.2. When the realizability assumption is not satisfied, the algorithm of EXO+ is not known to adapt to misspecification, so no results can be established.
>
> Assumption 2 is a relatively mild condition on the model class $\mathcal{M}$. Examples:
>
> (1) **Bandits.** Suppose that $\mathcal{M}$ is a class of bandits with Gaussian rewards, i.e. $\Pi=\mathcal{A}$ is the action space, and for each $M\in \mathcal{M}$, $a\in\mathcal{A}$, the observation is generated as $o\sim N(f^M(a),1)$. Then, we can consider the reference model $\overline{M}$ with $f^{\overline{M}}(a)\equiv 0$, and $D_{KL}(M(a)||\overline{M}(a))=\frac{f^M(a)^2}{2}$. Therefore, $C_{KL}$ is bounded as long as the mean reward is uniformly bounded for models in the model class $\mathcal{M}$.
>
> (2) **Contextual bandits.** Suppose that $\mathcal{M}$ is a class of contextual bandits with context space $\mathcal{C}$ and Gaussian rewards. Then, similar to (1), we can bound $C_{KL}\leq \log|\mathcal{C}|+O(1)$.
>
> (3) **Problem classes with bounded observation $\mathcal{O}$.** For finite $\mathcal{O}$, we can always bound $C_{KL}\leq \log|\mathcal{O}|$.
>
> More generally, $C_{KL}$ is bounded as long as the models in $\mathcal{M}$ admit uniformly bounded density functions with respect to a common base model. This is indeed the case for most applications, including many control problems and RL problems.
>
>
> > 5. It is not clear to me why there exists $M$ in the model class, i.e., why is the supremum achieved here by some $M$ in proof of Theorem 5.
>
> We proved in Theorem 5 that $\sup_{M\in\mathcal{M}} \mathbb{P}^{M,\texttt{ALG}}\left( g^M(\pi)\geq \Delta \right)> \delta$. Notice the supremum is *strictly* larger. Thus, by the definition of supremum, there exists one model that is larger than $\delta$.
>
> > 6. Line 329, I don't think $\mathbb{V}$ is defined.
>
> Thank you for pointing this out. \mathbb{V} is defined as variance. We will be sure to clarify this in the final revision.

---

> > ### Comment · Reviewer_Fyj4 · 2024-08-12
> >
> > Thanks for the response. As my questions have been adequately addressed, and I did not identify any major flaws or concerns with the approach, I maintain my current rating, and I believe the paper meets the necessary standards for acceptance.

---

### Official Review · Reviewer_3RHd · 2024-07-13

**Soundness:** 4
**Presentation:** 4
**Contribution:** 4
**Rating:** 8
**Confidence:** 3

**Summary:**

This work provides a unified perspective of existing techniques for deriving lower bounds. This viewpoint covers techniques that are useful for traditional statistical estimation (e.g., Fano's inequality, Le Cam's method, and Assouad's approach) as well as the recently proposed approach using decision-estimation coefficients that concerns interactive decision-making. In addition, this work proposes a novel complexity measure called decision coverage. Using this novel measure, this work derives lower and a polynomially matching upper bound for learning convex model classes.

**Strengths:**

S1. The paper discusses a unified perspective of techniques to derive lower bounds. It integrates classical techniques (Fano’s inequality, Le Cam’s method, and Assouad’s lemma) with contemporary methods for interactive decision-making, based on the Decision-Estimation Coefficient (DEC).

S2. This work introduces a novel complexity measure called decision coverage. This measure facilitates the derivation of new lower bounds specifically tailored for interactive decision-making.

**Weaknesses:**

See questions

**Questions:**

Q1. How realistic is Assumption 2? It seems to require that we have a model $\bar{M}$ that is close to all models $M \in \mathcal{M}.$ It is unclear why should such an assumption be true for a finite $C_{KL}?$ Similarly, how realistic is Assumption 3? Can you give some application examples where Assumptions 2 and 3 hold?

Q2. What is $\mathbb{V}$ in line 329? Is it variance?

Q3. Why is Assumption 2 not needed in Theorem 13, even though the line above says that this result is a corollary to Theorem 12?

Q4. Can the ideas in this work be extended to the RL setup?

Q5. Can the ideas in this work be extended to the case of interactive decision-making with multiple agents?

I am happy to increase my score based on your response.

**Limitations:**

The work does not explicitly discuss the limitations of their work; at least, I could not find it anywhere.

---

> ### Author Rebuttal · Authors · 2024-08-06
>
> Thank you for your insightful feedback and for dedicating time to review our paper!
>
>
> > Q1. How realistic is Assumption 2? It seems to require that we have a model that is close to all models It is unclear why should such an assumption be true for a finite Similarly, how realistic is Assumption 3? Can you give some application examples where Assumptions 2 and 3 hold?
>
> A1: Assumption 2 is a relatively mild condition on the model class $\mathcal{M}$. Examples:
>
> (1) **Bandits.** Suppose that $\mathcal{M}$ is a class of bandits with Gaussian rewards, i.e. $\Pi=\mathcal{A}$ is the action space, and for each $M\in \mathcal{M}$, $a\in\mathcal{A}$, the observation is generated as $o\sim N(f^M(a),1)$. Then, we can consider the reference model $\overline{M}$ with $f^{\overline{M}}(a)\equiv 0$, and $D_{KL}(M(a)||\overline{M}(a))=\frac{f^M(a)^2}{2}$. Therefore, $C_{KL}$ is bounded as long as the mean reward is uniformly bounded for models in the model class $\mathcal{M}$.
>
> (2) **Contextual bandits.** Suppose that $\mathcal{M}$ is a class of contextual bandits with context space $\mathcal{C}$ and Gaussian rewards. Then, similar to (1), we can bound $C_{KL}\leq \log|\mathcal{C}|+O(1)$.
>
> (3) **Problem classes with bounded observation $\mathcal{O}$.** For finite $\mathcal{O}$, we can always bound $C_{KL}\leq \log|\mathcal{O}|$.
>
> More generally, $C_{KL}$ is bounded as long as the models in $\mathcal{M}$ admit uniformly bounded density functions with respect to a common base model. This is indeed the case for most applications, including many control problems and RL problems.
>
> Assumption 3 is a growth condition on the DEC. For a broad range of model classes, including bandits, contextual bandits and structured RL problems (see e.g. [25] and also [23, 9]), the DEC of $\mathcal{M}$ scales as $C\sqrt{\epsilon}$, where $C$ is a quantity depending on the complexity of model class $\mathcal{M}$, and hence the DEC (as a function of $\epsilon$) is indeed of moderate growth for such classes.
>
>
>
>
> > Q2. What is  in line 329? Is it variance?
>
> A2: Yes, it is variance. We will clarify this in the revision.
>
>
>
> > Q3: Why is Assumption 2 not needed in Theorem 13, even though the line above says that this result is a corollary to Theorem 12?
>
> A3: For contextual bandits problem, we can directly bound $C_{KL}$ by $\log|\mathcal{C}|$, which is a logarithmic factor and hence hidden in $\lesssim$. We will clarify this in the revision.
>
>
>
> > Q4. Can the ideas in this work be extended to the RL setup?
>
> A4: Yes, the RL setup is encompassed by the DMSO framework [9], so our results can be applied as-is. It has been shown in [25, 9] that many existing lower bounds for RL problems can be recovered by the DEC framework, and hence the ideas of this work can be directly applied to RL setup. Further, when applied to the RL setup, our framework can give more powerful results than the commonly used Fano method and DEC method, and hence we expect it to provide a unified perspective for proving lower bounds for RL problems.
>
>
>
> > Q5. Can the ideas in this work be extended to the case of interactive decision-making with multiple agents?
>
> A5: Yes, the problem of equilibrium computation in multi-agent decision making is encompassed by the DMSO framework [21], so our results can be applied as-is. In particular, the lower bounds in [21] can also be recovered by our framework through Theorem 6.

---

> ### Comment · Area_Chair_7cjx · 2024-08-12
> **Please interact with the authors**
>
> Dear reviewer,
>
> Thanks for your work on this submission. Can you please interact with authors at this stage? At a minimum this would require acknowledging their rebuttal and saying whether you intend to change your score.
>
> Best,
> Roberto

---

### Decision · Program_Chairs · 2024-09-25

**Decision:**

Accept (spotlight)

**Comment:**

This paper introduces a new methodology for proving lower bounds in interactive decision making. The framework is broad enough to apply to standard statistical settings with i.i.d. data, and can recover general versions of the inequalities by Fano, Le Cam and Assouad in this setting. In interactive settings, the new method in some senses improves upon recent work by Foster et al ([25] in the paper) and gives a new quantity in terms of which one can find polynomially matching upper and lower bounds for regret.

The paper received good scores from the four reviewers (6,6,7,8). I actually believe that stronger scores would have obtained from a slightly improved presentation with corrected typos, a clearer message as in the response to uqGG,  and some more examples, for instance the ones on bandits and contextual bandits in the response to Fyj4. The authors' responses make me confident that all of these issues can be adequately addressed.

Overall, I think the paper makes a significant contribution, and deserves to be widely known amongst theoreticians in the NeurIPS community.